# `Proc-to-Spec`: **A Functorial Map of Network Processes**

**Shanfeng Hu**                                                 *shanfeng2.hu@northumbria.ac.uk*
*School of Computer Science*
*Northumbria University*
*Newcastle-upon-Tyne*
*United Kingdom*

**Reviewed on OpenReview:** *https://openreview.net/forum?id=pT84Ii6igG*

## Abstract

The analysis of dynamic networks is central to understanding complex environmental systems in nature, yet traditional methods often focus on describing changing states rather than formalising the underlying processes of change. In this work, we introduce a category-theoretical framework, `Proc-to-Spec`, that provides a principled, functorial method for analysing the transformations that govern network evolution. We model *resource-constrained systems*, such as those commonly found in biology and ecology, within a source category `Proc`, where morphisms represent dissipative physical processes. We then construct a spectral functor, $\chi : \texttt{Proc} \to \texttt{Spec}$, that maps each process to a unique linear transformation between the eigenspaces of the network's symmetrised Laplacian. This framework allows us to establish a set of rigorous theorems. We prove that physical conservation laws in `Proc` correspond directly to spectral invariants in `Spec`, such as the conservation of the Laplacian's trace. We derive a spectral sensitivity theorem that formally links resource dissipation to network fragmentation via the Fiedler value. We also establish a stability-spectrum equivalence theorem, proving that a system's physical dynamics converge to a stable state if and only if its spectral geometry converges. We also derive an optimal `Spec-to-Func` projection to compress these transformations into interpretable, low-dimensional functional fingerprints. We validate our theory with numerical experiments and demonstrate its generality as a tool for scientific discovery across two comprehensive, contrasting case studies. **(1)** In a high-signal, high-noise, macro-timescale ecological case study of the Serengeti food web in northern Tanzania, we use a large collection of 1.2 million classified image sets of animal activity from 225 camera traps spread across 1,125 km$^2$ of the Serengeti National Park from 2010 to 2013 to show that our framework can detect the subtle, cyclical signature of seasonal change and identify the unique geometric fingerprint of the 2011 East Africa drought. **(2)** In a low-signal, high-noise, micro-timescale neuroscience case study, we show that our framework's functional fingerprints can detect and characterise subtle cognitive processes from human brain fMRI data, classifying 8 distinct task states with high, generalisable accuracy. Our work provides a different way of thinking about dynamic systems, shifting the focus from describing states to understanding the fundamental geometry of change. Code to reproduce all results in the paper is released at `https://github.com/shanfenghu/pts`

## 1 Introduction

The analysis of dynamic networks is fundamental to science, providing the mathematical language to describe systems of interacting components that evolve over time. In biology and ecology, this paradigm is essential for understanding the stability of food webs (Pimm, 1984), the function of gene-regulatory pathways (Barabasi & Oltvai, 2004), and the cascading failures that can lead to abrupt, system-wide critical transitions (Scheffer et al., 2012). These systems are not static; they are governed by a complex interplay of processes—such as predation, resource competition, and metabolic conversion—that continuously reshape their structure and

function. The ultimate goal of scientific discovery is to move beyond mere description of these changes and toward a predictive understanding of the underlying principles that govern them.

Current methods for analysing dynamic networks, while powerful, are predominantly descriptive. The standard approach treats a dynamic network as a discrete time-series of static snapshots, $G_1, G_2, \ldots, G_t$. Techniques from spectral graph theory (Chung, 1997) and temporal network analysis (Masuda & Lambiotte, 2016) are then applied to compute metrics for each snapshot and track their evolution. This yields valuable insights into changing properties like connectivity or community structure. However, this approach leaves a critical theoretical gap: *it analyses the states of the system but does not provide a formal language for the processes that transform one state into the next.* This limitation is not only conceptual; it extends to modern machine learning models for dynamic graphs. While models like temporal Graph Neural Networks (GNNs) can be powerful predictors for practical applications, they often function as heavily-parameterised 'black boxes' (Ying et al., 2019). More critically, as we will formally prove in §4.2, their core architectures are often non-functorial, meaning their representations can violate the basic compositional and stability guarantees required for rigorous scientific analysis (Seo et al., 2018; Rossi et al., 2020; Spivak, 2014). A principled mathematical framework that can map a specific, causal mechanism of change to its unique, global structural consequence remains less explored. This gap prevents us from moving from observing *that* a system's structure changed to proving *why* it changed in a particular way.

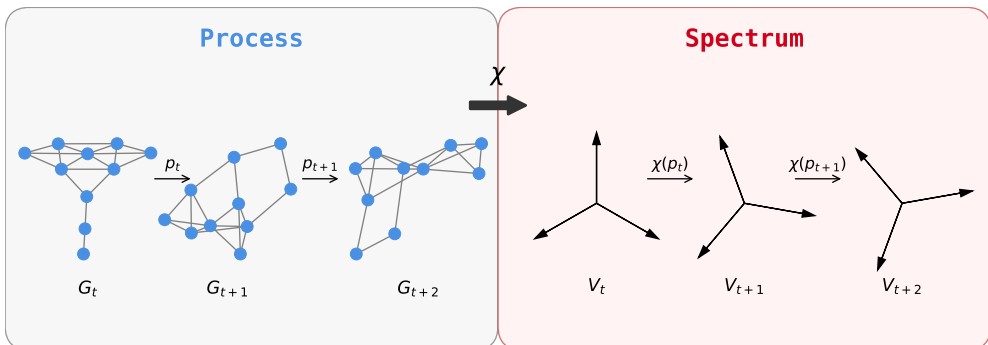

Figure 1: Conceptual illustration of our `Proc-to-Spec` framework. Our work establishes a formal mapping, the spectral functor ($\chi$), from the category of physical processes (`Proc`) to the category of their spectral representations (`Spec`). **Left**: A dynamic network evolves through a sequence of states ($G_t, G_{t+1}, G_{t+2}$). Each transformation is a physical process ($p_t, p_{t+1}$), which can represent any resource-constrained change, including perturbations to interaction strengths (edge weights) or topological modifications like the removal or addition of edges. **Right**: Each network state corresponds to a vector space ($V_t, V_{t+1}, V_{t+2}$) representing the eigenspace of the graph's symmetrised Laplacian. The functor $\chi$ maps each physical process $p_t$ to a unique linear transformation $\chi(p_t)$ that describes the change of basis between the corresponding eigenspaces. This framework allows us to analyse complex physical processes by studying their unique and well-behaved geometric signatures in the spectral domain.

Our work closes this gap by introducing a different way of thinking, `Proc-to-Spec`, as depicted in Figure 1. We deliberately focus on the foundational case of dynamic processes on simple, weighted graphs, which represent a vast class of real-world systems. We propose a conceptual shift from analysing system states to formalising the processes of transformation themselves. The core idea is that the effect of a physical process is most clearly understood not by tracking the state of individual nodes or edges, but by observing its impact on the network's holistic geometric structure. This structure is revealed by the network's Laplacian spectrum. We formalise this by building a direct, provable link between a process and its unique signature as a transformation in the spectral domain. This 'glass-box' approach provides a provably sound alternative to common predictive models (Seo et al., 2018; Rossi et al., 2020), grounding the analysis in a compositional framework rather than a 'black-box' architecture (Ying et al., 2019). This is analogous to moving from describing the scenes in a movie to analysing the script's rules that govern how one scene can lead to the next. While many methods for analysing dynamic networks focus on aggregating temporal information

or identifying recurring temporal motifs (Masuda & Lambiotte, 2016), our approach provides a different perspective by focusing on the geometric nature of the transformations between discrete network states.

We formalise this idea by constructing a categorical framework. We begin by defining a source category, `Proc` (for Process), where objects are weighted, directed graphs representing the state of a *resource-constrained* system. Crucially, the morphisms of `Proc` are not arbitrary graph edits, but are defined as dissipative physical processes that obey the fundamental constraint that resources cannot be created ex nihilo. This grounding in physical law is a key feature of our model. We then define a target category, `Spec` (for Spectrum), as the standard category of real vector spaces and linear transformations. The central contribution of this work is the construction and analysis of a spectral functor[1], $\chi : \texttt{Proc} \to \texttt{Spec}$, that serves as a structure-preserving map between these two worlds. This approach is inspired by the abstract and powerful language of Applied Category Theory (Spivak, 2014).

The functor $\chi$ maps a network state in `Proc` to the vector space spanned by the eigenvectors of its symmetrised Laplacian, providing a "spectral signature" of its structure. More importantly, $\chi$ maps a dissipative process to a unique linear transformation—a matrix—that describes the corresponding change of basis between the old and new eigenspaces. This formalism is not merely a bookkeeping device; it is a generative framework that allows us to derive a set of theorems. We prove that physical conservation laws in `Proc` correspond directly to spectral invariants in `Spec`, such as the conservation of the Laplacian's trace. We establish a spectral sensitivity theorem that formally links resource dissipation to network fragmentation via the Fiedler value. We also establish a stability-spectrum equivalence theorem, proving that a system's physical dynamics converge to a stable state if and only if its spectral geometry converges. Finally, to make the high-dimensional $n \times n$ functor map scientifically interpretable, we further derive an optimal, $k \times k$ `Spec-to-Func` projection that compresses the transformation into a low-dimensional, functional subspace (e.g., functional networks in brain network analysis).

We conduct a rigorous, multi-part experimental validation of our framework. First, we use a suite of synthetic experiments to systematically verify each lemma and theorem, confirming the mathematical correctness of our claims with numerical precision. Second, we demonstrate the framework's analytical power and generality across two comprehensive, real-world case studies from distinct scientific domains. **(1)** In a **high-signal, high-noise, macro-timescale, ecological** setting, we apply our framework to the Serengeti food web (Baskerville et al., 2011). We show that our theory can be applied to high-frequency, real-world data to generate both quantitative and qualitative insights into ecosystem dynamics. Using the large-scale "*Snapshot Serengeti*" camera trap dataset of animal activity in northern Tanzania (Swanson et al., 2015), we demonstrate that our framework is sensitive enough to detect the subtle, cyclical spectral signature of seasonal change and to characterise the unique geometric signature of a major, documented ecological shock—the 2011 East Africa drought. **(2)** In a **low-signal, high-noise, micro-timescale, neuroscience** setting, we analyse dynamic brain connectomes from the Human Connectome Project (HCP) (Van Essen et al., 2012). We show that the framework is sensitive enough to detect subtle geometric signatures of cognitive state change, revealing a 'lock-in' effect during tasks (He, 2013) and separating cognitive processes into distinct families based on their spectral and geometric properties. We then demonstrate that our `Spec-to-Func` projection compresses these signatures into unique, interpretable fingerprints that successfully classify subtle task states from brain fMRI data with high, generalisable accuracy. This validation confirms that our framework provides a different, practical, and powerful lens for understanding complex systems.

Our contributions are as follows:

- We introduce a categorical framework, `Proc-to-Spec`, for analysing dynamic networks by formalising the morphisms of change.

- We construct a spectral functor, $\chi : \texttt{Proc} \to \texttt{Spec}$, that maps dissipative physical processes to unique linear transformations in the spectral domain.

- We provide a formal critique of common temporal machine learning models, proving that they are non-functorial for rigorous scientific discovery.

---

[1]Our numerical validations show that functoriality holds up to a tiny, numerically tractable error in practice, while our proof in §A.1 details the specific conditions under which it is exact.

- We establish a set of rigorous theorems that provide a direct, provable link between physical properties (conservation, dissipation, stability) and spectral signatures.

- We derive a principled, optimal `Spec-to-Func` projection that compresses high-dimensional spectral transformations into interpretable, low-dimensional functional subspaces.

- We provide an experimental validation on synthetic data and demonstrate the framework's generality on two distinct real-world case studies: a high-signal, high-noise, macro-timescale ecological analysis of the Serengeti food web and a low-signal, high-noise, micro-timescale neuroscience analysis of cognitive processes in the human brain based on the Human Connectome Project.

The remainder of this paper is structured as follows. In §2, we review related work. In §3, we formally define our model. In §4, we present our theoretical analysis and prove our main theorems. In §5, we detail our synthetic experimental validation. In §6, we apply our framework to the Serengeti case study. In §7, we demonstrate its generality in a neuroscience case study of brain connectomics. We discuss implications and conclude in §8. Detailed proofs are provided in the §A.

## 2  Related Work

Our `Proc-to-Spec` framework represents a synthesis of ideas from several distinct but related fields. We begin in §2.1 by reviewing the foundational principles of Spectral Graph Theory, the core mathematical language we use for our structural representations. Next, in §2.2, we situate our model within the growing field of dynamic and temporal network analysis. We then ground our work in its target scientific domain in §2.3, discussing the rich history of network modelling in ecology and biology and the persistent challenge of linking local mechanisms to global system dynamics. Our formal approach is heavily inspired by the paradigm of Applied Category Theory, which we discuss in §2.4, clarifying how its principles of compositionality and functorial mappings provide the blueprint for our framework. Finally, in §2.5, we connect our framework to the contemporary landscape of machine learning on graphs.

### 2.1  Spectral Graph Theory and Its Applications

Spectral graph theory, which studies the properties of a graph via the eigenvalues and eigenvectors of its associated matrices, is a cornerstone of modern data analysis (Chung, 1997). The graph Laplacian, in particular, has found widespread application, with different versions (e.g., unnormalised, symmetric normalised, random walk) offering different perspectives on the graph's structure (Von Luxburg, 2007). Its properties are used for graph partitioning (Pothen et al., 1990), graph drawing (Koren, 2003), and non-linear dimensionality reduction through methods like Laplacian Eigenmaps and Diffusion Maps (Belkin & Niyogi, 2003). The entire field of Graph Signal Processing (GSP) is built on the idea of using the Laplacian eigenbasis as a Fourier basis for signals defined on graphs (Leus et al., 2023; Shuman et al., 2013). The most famous application is spectral clustering (Von Luxburg, 2007; Shi & Malik, 2000), which uses the eigenvectors of the Laplacian to identify community structure. The Fiedler value (the second-smallest eigenvalue) and its corresponding eigenvector are of particular importance, as they provide a measure of a graph's algebraic connectivity and identify its primary structural bottlenecks (Fiedler, 1973; Mohar et al., 1991). The analysis of how spectra change in response to dynamic processes is less formalised. Research has focused on spectral perturbation theory (Bhatia, 1992), which provides bounds on eigenvalue changes for small perturbations, and on tracking eigenvalue time-series to detect anomalies or regime shifts (Sandryhaila & Moura, 2014). Our work builds on this by providing a functorial framework that maps the processes of change themselves to explicit transformations in the spectral domain, a fundamentally different and more structured approach that aims to formalise the link between the cause and the spectral consequence of a change.

### 2.2  Dynamic and Temporal Networks

The study of networks that evolve over time is a mature field (Masuda & Lambiotte, 2016; Casteigts et al., 2012). Research has traditionally focused on developing metrics and models to characterise temporal interaction patterns, often distinguishing between representations as sequences of static snapshots, continuous-time

contact sequences, or more recently, stream graphs that explicitly model interactions as they occur (Masuda & Lambiotte, 2016; Latapy et al., 2018). Efforts to characterise these networks include identifying temporal motifs (Kovanen et al., 2011), analysing time-respecting reachability and pathfinding (Kempe et al., 2000), and developing temporal centrality measures to identify key nodes in dynamic processes (Lerman et al., 2010). A significant body of work has focused on modelling information diffusion and influence maximisation (Kempe et al., 2003), often using cascade models or threshold dynamics. Models for network evolution aim to capture the mechanisms driving change, such as preferential attachment for scale-free structures (Barabási & Albert, 1999), triadic closure for social clustering (Bianconi & Barabási, 2001), and a wide array of link prediction models that forecast future interactions (Liben-Nowell & Kleinberg, 2003; Hasan & Zaki, 2011). More advanced models have considered continuous-time dynamics using point processes like Hawkes processes (Nguyen et al., 2018), as well as the evolution of community structures over time (Palla et al., 2007; Tantipathananandh et al., 2007; Liu et al., 2020). While this body of work provides a rich vocabulary and powerful tools for *describing* and *predicting* network evolution, it generally lacks a formal, compositional language for the *processes* themselves. The focus remains on the sequence of states or the statistical properties of events, not on a formal algebra of the transformations that connect them, a gap our framework addresses directly.

## 2.3 Networks in Biological and Environmental Sciences

Network theory has become an indispensable tool in the sciences, providing a language to manage the immense complexity of biological and ecological systems. In ecology, food web analysis is used to study ecosystem stability, resilience, and the role of keystone species (Pimm, 1984; Montoya et al., 2006; Dunne et al., 2002; Williams & Martinez, 2000; Berlow et al., 2004). Network models are used to understand mutualistic interactions, whose nested structure is thought to increase biodiversity (Bascompte et al., 2003), as well as disease propagation (Newman, 2002) and the structure of metapopulations (Hanski, 1998). A key challenge in this field is the identification of early-warning signals for critical transitions, a problem our stability theorems directly address (Kéfi et al., 2014). In biology, networks are used to model protein-protein interactions (PPIs) to uncover functional modules and identify potential drug targets (Jeong et al., 2001; Barzel & Barabási, 2013), and gene-regulatory pathways that control cellular life (Davidson et al., 2002; Alon, 2007). Large-scale metabolic network reconstructions, such as the Recon project for human metabolism, are used in constraint-based modelling to predict metabolic fluxes (Rual et al., 2005). In neuroscience, analysing the brain's structural and functional connectomes is a central goal, with projects like the Human Connectome Project providing massive datasets (Sporns et al., 2005; Bullmore & Sporns, 2009; Honey et al., 2007; Van Essen et al., 2012). These applications demonstrate a clear and urgent need for models that can handle dynamic *processes* and provide insight into system-level properties like stability and resilience. Our work is directly motivated by this need, providing a formal language and a set of theoretical tools specifically designed to analyse the resource-constrained dynamics that are characteristic of these natural systems.

## 2.4 Categorical and Geometric Approaches to Systems

Our framework is inspired by a growing movement to apply principled, abstract mathematics to machine learning and systems modelling. Geometric Deep Learning seeks to unify graph-based models and beyond by focusing on underlying symmetries and invariances, building models that respect the geometry of their domain (Bronstein et al., 2017). Alongside spectral methods, tools from Topological Data Analysis (TDA) are also used to study the higher-order structure of networks through persistent homology (Petri et al., 2013). More abstractly, Applied Category Theory provides a formal language for compositionality, arguing that systems are best understood by how their components compose (Fong & Spivak, 2018). This approach has been used to model a wide range of systems, including databases (Guyot et al., 2022), electrical circuits (Takahashi, 2023), dynamical systems (Behrisch et al., 2017), compositional game theory (Ghani et al., 2018), and network protocols (Fong & Spivak, 2018). Other formalisms like operads are used to describe more general systems of composition (Baez & Stay, 2010). This research program, pioneered by figures like Baez & Stay (2010) and Spivak (2014), argues that the language of functors and morphisms is the natural way to describe complex, interacting systems. Our work contributes to this paradigm by providing a concrete instantiation of a functorial model. We use the categorical language not as an end in itself, but as the natural

grammar to build a specific, testable, and scientifically relevant theory that connects process to structure, thus grounding the abstract formalism in concrete, provable spectral consequences.

## 2.5 Machine Learning on Graphs

In recent years, machine learning on graph-structured data has been dominated by the success of Graph Neural Networks (GNNs), a broad class of models based on a neighbourhood aggregation or message-passing scheme (Scarselli et al., 2008; Gori et al., 2005; Gilmer et al., 2017). Architectures like Graph Convolutional Networks (GCNs) (Kipf, 2016), Graph Attention Networks (GATs) (Veličković et al., 2017), and GraphSAGE (Hamilton et al., 2017) have achieved state-of-the-art performance on tasks like node classification and link prediction. More recent developments include Graph Transformers, which aim to capture long-range dependencies (Shi et al., 2020). To handle dynamic networks, researchers have developed various temporal GNNs that integrate GNN principles with recurrent or attention-based sequence models (Seo et al., 2018; Rossi et al., 2020; Kazemi et al., 2020; Manessi et al., 2020). Other approaches include graph kernels (Shervashidze et al., 2011), and representation learning techniques like DeepWalk (Perozzi et al., 2014) and node2vec (Grover & Leskovec, 2016). A limitation of many of these powerful models, particularly for applications in science, is the trade-off between predictive accuracy and interpretability. While they provide high predictive accuracy, their heavily-parameterised, 'black-box' nature often lacks the desired level of scientific interpretability, and the community has invested significant effort in developing post-hoc explanation methods like GNNExplainer (Ying et al., 2019). More fundamentally, many temporal GNNs (e.g., Seo et al., 2018; Rossi et al., 2020) are what we formalise in §4.2 as *Entangled State-Update Models*. By their design, these models entangle the representation of a state with the representation of a process, often via a recurrent update function. As we prove in Theorems 1 and 2, this architectural choice makes them non-functorial: they are not guaranteed to preserve identity (i.e., they introduce artificial dynamics) and, more critically, they do not preserve the composition of morphisms. This makes their representations an artifact of the data's sampling rate, which is a critical flaw for rigorous, path-independent scientific analysis. Our `Proc-to-Spec` framework aims to provide a 'glass-box' alternative specifically designed to solve this limitation, offering a provably sound and path-independent approach where the link between process and outcome is explicit, provable, and scientifically interpretable, aiming for understanding over raw prediction.

## 3 The `Proc-to-Spec` Framework

In this section, we provide the formal mathematical construction of our `Proc-to-Spec` framework. The framework is built upon a single, central idea: *the existence of a structure-preserving map—a functor—that translates the dynamics of a physical system into the geometric language of linear algebra.* The resulting model is a logical consequence of defining this functor and its domain and codomain. We begin by defining our key notations, summarised in Table 1.

Table 1: Key symbols and notations used in the paper.

| Symbol | Description |
|---|---|
| $G = (V, E, W)$ | A weighted, directed graph representing a network state. |
| $V, E, W$ | The set of vertices, edges, and the edge weight function, respectively. |
| $\mathcal{R}(G)$ | The total resource in a network, defined as the sum of all edge weights. |
| $p : G \to G'$ | A process (morphism) that transforms one network state to another. |
| `Proc` | The category of resource-constrained dynamic networks. |
| $L_W$ | The weighted Laplacian of a graph with weight matrix $W$. |
| $L_{sym}$ | The symmetrised Laplacian, used for spectral analysis. |
| $\lambda_i, \mathbf{v}_i$ | The $i$-th eigenvalue and corresponding eigenvector of $L_{sym}$. |
| $\lambda_2$ | The Fiedler value, or algebraic connectivity of the graph. |
| `Spec` | The category of finite-dimensional real vector spaces. |
| $\chi$ | The spectral functor, mapping from `Proc` to `Spec`. |
| $\chi(p)$ | A linear transformation representing the process $p$ in the spectral domain. |

### 3.1 The Source Category: `Proc`

The source category, `Proc`, is designed to represent the physical reality of resource-constrained dynamic systems commonly found in biological and ecological sciences.

**Objects.** An object in `Proc` is a network state, formally defined as a weighted, directed graph $G = (V, E, W)$, where:

- $V$ is a finite set of $n$ vertices, representing the components of the system (e.g., species in an ecosystem, proteins in a cell).

- $E \subseteq V \times V$ is a set of directed edges, representing interactions between components.

- $W : E \to \mathbb{R}_0^+$ is a weight function that assigns a non-negative real value to each edge, representing the strength, capacity, or rate of flow of a resource (e.g., energy, biomass, information).

**Morphisms.** A morphism in `Proc`, denoted $p : G \to G'$, is a **dissipative process** that transforms an initial state $G$ into a final state $G'$. The defining characteristic of these systems is that they operate under resource limitation. We formalise this by defining the total resource of a network as the sum of all its edge weights:

$$\mathcal{R}(G) = \sum_{(u,v) \in E} W(u, v) \tag{1}$$

A process $p$ is defined as dissipative if the total resource in the final state is less than or equal to the total resource in the initial state:

$$\mathcal{R}(G') \leq \mathcal{R}(G) \tag{2}$$

This single constraint is fundamental. It reflects the second law of thermodynamics, where energy is lost in transfers between trophic levels (Odum, 1957), and the principle of resource competition that governs the dynamics of all biological populations (Tilman, 1982). This constraint ensures that our model is physically and biologically grounded for scientific discovery.

### 3.2 The Target Category: `Spec`

The target category, `Spec`, provides the abstract geometric space where the structural properties of our networks are analysed. It is the standard category of finite-dimensional real vector spaces.

**Objects.** An object is a finite-dimensional vector space $U$ over the field of real numbers, $\mathbb{R}$.

**Morphisms.** A morphism is a linear transformation $T : U \to V$ between two vector spaces.

### 3.3 The Spectral Functor $\chi : \text{Proc} \to \text{Spec}$

The spectral functor $\chi$ is the heart of our framework. It provides the formal, structure-preserving map from the physical world of processes to the geometric world of spectra.

**Action on Objects.** The functor $\chi$ maps a network object $G \in \text{Proc}$ to the $n$-dimensional vector space spanned by the eigenvectors of its symmetrised Laplacian. This construction is essential for ensuring a well-defined geometric representation. Given a graph $G$ with $n$ vertices and weight function $W$:

1. We first construct the $n \times n$ weighted adjacency matrix $A_W$, where $(A_W)_{ij} = W(i, j)$.

2. To guarantee a real spectrum and a complete orthonormal eigenbasis, we construct the symmetrised adjacency matrix, $A_{sym}$, where $(A_{sym})_{ij} = (W(i, j) + W(j, i))/2$. This captures the underlying "connectivity fabric" of the network, which is often the primary interest in resilience and stability studies (Mohar et al., 1991).

3. From this, we construct the symmetrised Laplacian:

$$L_{sym} = D_{sym} - A_{sym} \tag{3}$$

where $D_{sym}$ is the diagonal matrix of weighted degrees derived from $A_{sym}$.

4. This ensures that $L_{sym}$ is a real symmetric matrix and thus has a complete set of $n$ orthonormal eigenvectors $\{\mathbf{v}_1, \ldots, \mathbf{v}_n\}$.

5. The functor maps the graph object to the vector space these eigenvectors span:

$$\chi(G) = \mathrm{span}(\mathbf{v}_1, \ldots, \mathbf{v}_n) \cong \mathbb{R}^n \tag{4}$$

**Action on Morphisms.** The functor $\chi$ maps a process $p : G \to G'$ to the unique linear transformation $\chi(p) : \chi(G) \to \chi(G')$ that describes the change of basis between the respective eigenspaces. If $B = \{\mathbf{v}_i\}$ is the eigenbasis of $L_{sym}$ and $B' = \{\mathbf{v}'_i\}$ is the eigenbasis of $L'_{sym}$, then $\chi(p)$ is the unique linear map that represents this geometric transformation.

**Physical Intuition of the Functor.** Conceptually, the spectral functor $\chi$ can be understood as a mathematical "lens" that allows us to view the physical process through the language of geometry. The eigenbasis of a network's Laplacian represents its fundamental modes of variation—its "structural harmonics". A physical process, $p$, perturbs the network, creating a new set of structural harmonics. The transformation matrix $\chi(p)$ precisely quantifies how each of the old harmonics is distributed or "scattered" among the new ones (see Figure 12 for such scattering in a real-world ecological web). It is, in essence, a mathematical description of the structural reorganisation induced by the process. A simple process might only slightly alter the harmonics, resulting in a transformation close to the identity, while a complex process might completely scramble them, resulting in a highly complex rotational transformation.

### 3.4 Model Scope and Assumptions

Our framework is designed to be a model for rigorous scientific inquiry in environmental sciences, with carefully chosen scope and assumptions that we state explicitly here.

- **Deterministic Processes.** We model processes as deterministic, representing the expected outcome of potentially stochastic interactions. This is a standard and powerful simplification in the modelling of complex systems, providing a tractable first-order approximation of the system's dynamics (May, 2001).

- **Focus on Edge Weights.** We assume that the primary dynamics of the system can be captured by changes in the interaction strengths (edge weights). This is a well-justified focus in ecological network modelling, where fluctuations in interaction strength are a primary driver of system stability (Allesina & Tang, 2012). This naturally includes topological changes as a special case.

- **Symmetrised Laplacian.** The core operator used by the spectral functor is the symmetrised Laplacian, $L_{sym}$. We select this operator due to its desirable properties, namely a real spectrum and a complete orthonormal eigenbasis, which are essential for defining a consistent spectral geometry for our framework.

## 4 Theoretical Analysis

Having formally defined the `Proc-to-Spec` framework in §3, we now establish its key theoretical properties. Our analysis is structured to build a complete framework from the ground up, moving from foundational mathematical guarantees to the main scientific results. We begin in §4.1 by proving that our spectral map, $\chi$, is a valid functor, a foundational result that ensures the mathematical soundness and compositionality of our entire framework. With this foundational guarantee established, in §4.2, we provide a formal justification for

our categorical approach by contrasting it with common machine learning architectures. We define a broad class of temporal graph models as Entangled State-Update Models and prove that they are non-functorial and thus fundamentally ill-suited for rigorous, path-independent scientific analysis. Following this critique, in §4.3, we establish the core dictionary that translates the physical constraints of the `Proc` category into the language of the `Spec` category; here, we prove our key theorems on the spectral signatures of resource conservation and dissipation. Building on this, we present our main result in §4.4, the Stability-Spectrum Equivalence theorem, which provides a formal and testable link between the long-term dynamical stability of a system and the convergence of its spectral geometry. In §4.5, we derive a toolkit of more specialised theorems that provide concrete, interpretable matrix signatures for specific, common types of processes, such as local perturbations and node removals. While this toolkit is powerful for diagnosing simple events, interpreting the full $n \times n$ transformation matrix $\chi(p)$ of a complex, system-wide process remains a significant challenge. Therefore, in §4.6, we solve this problem by deriving an optimal projection that compresses the high-dimensional spectral transformation into a low-dimensional $k \times k$ functional matrix. Together, these results form a theoretical basis for using our framework as a tool for scientific reasoning.

## 4.1 Functoriality of the Spectral Map $\chi$

We begin by establishing the foundational mathematical property of our framework: that the map $\chi$ is a valid functor. This result is crucial as it guarantees that our spectral representation is a true and consistent reflection of the underlying system's dynamics, respecting both identity and the composition of processes.

**Lemma 1** (Functoriality of $\chi$). *The map $\chi : \mathtt{Proc} \to \mathtt{Spec}$ is a functor. It preserves identity morphisms and the composition of morphisms.*

*Proof Sketch.* To prove that $\chi$ is a functor, we must verify two conditions.

1. **Preservation of Identity:** The identity morphism in `Proc` is the process $\mathrm{id}_G : G \to G$, which leaves the network unchanged. This means the initial and final weight matrices are identical ($W' = W$), leading to identical symmetrised Laplacians ($L'_{sym} = L_{sym}$) and thus identical eigenbases. The linear transformation that maps an orthonormal basis to itself is the identity transformation, $\mathrm{id}_{\chi(G)}$. Thus, $\chi(\mathrm{id}_G) = \mathrm{id}_{\chi(G)}$.

2. **Preservation of Composition:** Consider two composable processes, $p_1 : G_1 \to G_2$ and $p_2 : G_2 \to G_3$. The map $\chi(p_1)$ is the change of basis matrix from the eigenbasis of $G_1$ to that of $G_2$, and $\chi(p_2)$ is the change of basis from $G_2$ to $G_3$. The composition of linear transformations, $\chi(p_2) \circ \chi(p_1)$, corresponds to the matrix product of these change of basis matrices. By the chain rule for change of basis, this product is precisely the matrix that transforms the basis of $G_1$ directly to the basis of $G_3$, which is by definition $\chi(p_2 \circ p_1)$. Thus, $\chi(p_2 \circ p_1) = \chi(p_2) \circ \chi(p_1)$.

The full proof is provided in §A.1. $\qquad\square$

**Interpretation.** This lemma provides the guarantee of our framework's logical consistency. It ensures that our spectral "lens" does not distort the structure of the system's dynamics. From a scientific perspective, this has two critical implications. First, the preservation of identity means that a system in a stable, unchanging state will have a stable, unchanging spectral representation; our method does not introduce artificial dynamics. Second, the preservation of composition means that our framework respects causality. For example, the overall structural impact of a drought followed by the introduction of an invasive species is precisely the composition of their individual spectral transformations. This allows scientists to model complex, multi-stage scenarios with the confidence that the resulting analysis is a faithful representation of the composite process, making our "spectral accounting" of change both rigorous and reliable.

## 4.2 Non-Functoriality of Entangled State-Update Models

The functorial property (Lemma 1) is not merely a mathematical convenience; it establishes a minimal and fundamental criterion for any model that claims to represent the *process of change* itself. It guarantees that

the model's representations are consistent, compositional, and path-independent. To illustrate the necessity of this property for the machine learning community, we now formalise a broad class of common temporal machine learning models (Seo et al., 2018; Rossi et al., 2020). We will show that these models, by their design, are non-functorial because they entangle the representation of a process with the representation of a state, leading to critical failures in preserving composition and identity.

First, we define this class of machine learning models, which we term Entangled State-Update Models.

**Definition 1** (Entangled State-Update Model). Let $\mathcal{G}$ be the space of graphs (network states) and $\mathcal{V}$ be a latent vector space (the embedding space). A common class of temporal graph models (e.g., GNN-RNNs (Seo et al., 2018)) are Entangled State-Update Models, defined by a pair of maps:

1. A state encoder, $\Phi : \mathcal{G} \to \mathcal{V}_{\text{state}}$ (e.g., a GNN that embeds a graph snapshot).

2. A stateful update function, $F : \mathcal{V}_{\text{state}} \times \mathcal{V}_{\text{history}} \to \mathcal{V}_{\text{history}}$ (e.g., an RNN cell).

The system's latent representation at time $t + 1$ is computed as:

$$\mathbf{z}_{t+1} = F(\Phi(G_{t+1}), \mathbf{z}_t) \tag{5}$$

where $\mathbf{z}_t$ is the history vector encoding the sequence $(G_0, \ldots, G_t)$. The update operation is a function of the new state $G_{t+1}$ and the previous history $\mathbf{z}_t$.

This class of models are powerful and predictive for machine learning applications. However, they have an explicit map for *states* ($\Phi(G)$) but no explicit map for *processes* ($p_t$). The process is implicit in the call to the update function $F$. We can now prove that this architectural choice makes it impossible for such a model to be a valid process model in our categorical sense.

These entangled models fail the most basic tests of a faithful representation. The first critical failure is the inability to preserve identity (Lemma 1, part 1). A model that cannot represent "no change" without introducing its own artificial dynamics cannot be trusted as a representation of the physical system.

**Theorem 1** (The Identity Failure of Entangled Models). *An Entangled State-Update Model (as defined in Definition 1) is non-functorial as it fails to preserve the identity morphism. The model's representation of an identity process $p_{id} : G_t \to G_t$ is not the identity transformation on the latent history space.*

*Proof Sketch.* The functoriality axiom for identity states $\chi(id_G) = id_{\chi(G)}$. In an Entangled State-Update Model, the transformation corresponding to a process $p : G_t \to G_{t+1}$ is the update operation $f_t(\cdot) = F(\Phi(G_{t+1}), \cdot)$. This map acts on the history space, $f_t : \mathcal{V}_{\text{history}} \to \mathcal{V}_{\text{history}}$.

Consider the identity process $p_{id} : G_t \to G_t$, where the system state is unchanged. The model's update operation for this process is $f_{id}(\cdot) = F(\Phi(G_t), \cdot)$. We apply this map to the current history vector $\mathbf{z}_t$:

$$\mathbf{z}_{t+1} = f_{id}(\mathbf{z}_t) = F(\Phi(G_t), \mathbf{z}_t) \tag{6}$$

For the model to preserve identity, the resulting history vector $\mathbf{z}_{t+1}$ must be identical to the input history vector $\mathbf{z}_t$. The map $f_{id}$ must be the identity map $id_{\mathcal{V}_{\text{history}}}$. This requires that:

$$\mathbf{z}_t = F(\Phi(G_t), \mathbf{z}_t) \quad \text{for all } \mathbf{z}_t \in \mathcal{V}_{\text{history}} \tag{7}$$

This condition is generally not met by the non-linear recurrent functions (e.g., LSTMs, GRUs) commonly used in these machine learning models. The update function $F$ is parameterised by trained weights and is designed to transform its input history vector $\mathbf{z}_t$ based on the new state embedding $\Phi(G_t)$. For a non-trivial RNN, $F(\mathbf{x}, \mathbf{h}) \neq \mathbf{h}$ in the general case. The model has no mechanism to guarantee that "re-processing" the same state $G_t$ will result in a null operation. Thus, $\mathbf{z}_{t+1} \neq \mathbf{z}_t$. The model fails to preserve identity. The full proof is provided in §A.2. □

**Interpretation.** This theorem proves that an Entangled State-Update Model introduces **artificial latent dynamics**. Even when the physical system is perfectly stable (the $p_{id}$ process), the model's internal representation $\mathbf{z}_t$ will continue to evolve. This makes it impossible to use the model's latent space to, for example, test for equilibrium (as in our Theorem 5). A non-functorial model's stability is a property of the model itself, not of the system it purports to represent.

The second critical failure of these models is the inability to preserve composition (Lemma 1, part 2). This failure manifests as a critical, practical problem: **path-dependence**.

**Theorem 2** (The Composition Failure of Entangled Models). *An Entangled State-Update Model (as defined in Definition 1) is non-functorial as it fails to preserve the composition of morphisms.*

*Proof Sketch.* The functoriality axiom for composition states $\chi(p_2 \circ p_1) = \chi(p_2) \circ \chi(p_1)$. We can test this by comparing the model's output for a "direct" process versus a "stepped" process. Let the system be in an initial state $G_1$ with a history vector $\mathbf{z}_1$. Consider a composite process $p_{1\to3} = p_2 \circ p_1$, which proceeds in two steps: $G_1 \xrightarrow{p_1} G_2 \xrightarrow{p_2} G_3$.

1. **The Stepped Path (Model's $\Psi(p_2) \circ \Psi(p_1)$):** An Entangled Model observes the full path. It first computes the effect of $p_1$, updating the history vector:

$$\mathbf{z}_2 = F(\Phi(G_2), \mathbf{z}_1) \tag{8}$$

    It then computes the effect of $p_2$, updating from the new history:

$$\mathbf{z}_3 = F(\Phi(G_3), \mathbf{z}_2) = F(\Phi(G_3), F(\Phi(G_2), \mathbf{z}_1)) \tag{9}$$

    This final vector, $\mathbf{z}_3$, represents the model's understanding of the full, two-step process.

2. **The Direct Path (Model's $\Psi(p_{1\to3})$):** Now consider an analyst who only has snapshots $G_1$ and $G_3$. They model the direct process $p_{1\to3}$. The Entangled Model, lacking the intermediate state $G_2$, computes the update in a single step based on the definition of its transformation:

$$\mathbf{z}'_3 = F(\Phi(G_3), \mathbf{z}_1) \tag{10}$$

3. **The Failure:** The model fails the composition test because the two resulting representations are not equal:

$$\mathbf{z}_3 \neq \mathbf{z}'_3 \tag{11}$$

$$F(\Phi(G_3), F(\Phi(G_2), \mathbf{z}_1)) \quad \neq \quad F(\Phi(G_3), \mathbf{z}_1) \tag{12}$$

    (This inequality holds for any non-trivial recurrent function $F$, as the inner term $F(\Phi(G_2), \mathbf{z}_1)$ is generally not equal to $\mathbf{z}_1$).

As a result, the model's representation of the change from $G_1$ to $G_3$ is fundamentally different depending on whether it observed the intermediate path through $G_2$. Its representation is **path-dependent**. In contrast, our `Proc-to-Spec` framework is **path-independent**. The map $\chi(p_{1\to3})$ (the direct change of basis from $G_1 \to G_3$) is *by definition* mathematically identical to the matrix product $\chi(p_2) \circ \chi(p_1)$. The full proof is provided in §A.3. □

**Interpretation.** This theorem essentially proves that a standard GNN-RNN's analysis of a system's change over a month is generally *not* the composition of its analyses of the 30 daily changes within that month. This failure is precisely what makes such models inadequate tools for scientific discovery, where data is often sparse or irregularly sampled. A model whose results are an artifact of the sampling rate (i.e., path-dependent) may not be able to reveal the underlying, invariant geometry of change. The functorial property, which our framework respects, is the formal guarantee of this crucial path-independence to ensure the consistency of scientific discovery.

**Category Theory for Model Soundness.** This sequence of theorems provides a formal justification for our `Proc-to-Spec` framework. We first defined a broad and common class of temporal machine learning architectures as Entangled State-Update Models (Definition 1), such as the GNN-RNN models common in the literature (Seo et al., 2018; Rossi et al., 2020). We then proved that this ubiquitous architecture is non-functorial, which is not a minor theoretical flaw but the source of a pair of critical, practical failures: they are **incapable of representing stability** (Theorem 1) and **path-dependent** (Theorem 2). This formalisation demonstrates that the language of category theory is not an abstract descriptor but a prescriptive benchmark for model soundness (Spivak, 2014; Fong & Spivak, 2018). While one could attempt to formalise these properties *ad hoc*, it is the language of functors that provides the established, minimal, and universally understood axioms—namely, the preservation of **identity** and **composition**—to test for them. Any alternative formalism that successfully captures these properties would, in essence, be a *re-invention* of these core categorical concepts.

The implication is of central relevance to machine learning, as it delineates a critical trade-off. While the non-functorial models we defined can be powerful *predictors*, their widespread adoption, often without consideration for these axiomatic failures, highlights a key gap in the pursuit of interpretable and robust AI (Ying et al., 2019). Our work proves that for the distinct and increasingly important goal of rigorous mechanistic analysis in machine learning applications, these architectures are fundamentally ill-suited. The fact that a model achieves high predictive accuracy does not automatically make it a sound and faithful representation of an underlying physical process. This makes it difficult to conduct the analysis we demonstrate in our case studies (§6 and §7)—such as distinguishing catastrophic critical transitions from seasonal changes (Scheffer et al., 2012), or formally assessing system stability (Pimm, 1984; Allesina & Tang, 2012)—especially on the sparse, real-world data common in science (Swanson et al., 2015).

### 4.3 The Physics-to-Spectra Dictionary

Having established the necessity, robustness, and canonical nature of our functorial approach, we now derive the core dictionary that translates the physical laws governing the `Proc` category into the geometric language of `Spec`. The following theorems show that fundamental physical constraints—namely, the conservation and dissipation of resources—induce specific, non-trivial, and observable signatures in the spectral domain.

**Theorem 3** (The Spectral Trace Conservation Law). *Let $p : G \to G'$ be a conservative process, where the total resource is unchanged ($\mathcal{R}(G) = \mathcal{R}(G')$). The trace of the symmetrised Laplacian is conserved, i.e., $\mathrm{Tr}(L'_{sym}) = \mathrm{Tr}(L_{sym})$. Consequently, the sum of the Laplacian eigenvalues is an invariant of the process.*

*Proof Sketch.* The key insight is to first establish a direct identity between the total resource of a network and the trace of its symmetrised Laplacian. The trace of $L_{sym}$ is the sum of its diagonal elements, which are the weighted degrees of the symmetrised graph. This sum is equivalent to the sum of all entries in the symmetrised adjacency matrix, $A_{sym}$. By substituting the definition of $A_{sym}$, we show that this sum is precisely equal to the total resource, $\mathcal{R}(G)$. Therefore, the identity $\mathrm{Tr}(L_{sym}) = \mathcal{R}(G)$ holds for any graph. For a conservative process, since $\mathcal{R}(G) = \mathcal{R}(G')$, it follows directly that $\mathrm{Tr}(L'_{sym}) = \mathrm{Tr}(L_{sym})$. Because the trace of a matrix is equal to the sum of its eigenvalues, the sum of the eigenvalues is also conserved. The full proof is provided in §A.4. □

**Interpretation.** This theorem provides our first concrete link between a physical law and a geometric invariant. It demonstrates that if a system is closed and only redistributes its internal resources (e.g., biomass transfer within a food web without external inputs or losses), the sum of its spectral eigenvalues remains constant. This spectral sum can be interpreted as a measure of the total "structural energy" or "information capacity" of the network. This result provides an integrity check for models of closed ecosystems, ensuring that the simulated dynamics correctly preserve this global spectral quantity.

**Theorem 4** (The Spectral Sensitivity of Algebraic Connectivity). *Let $p : G \to G'$ be a process that induces a sufficiently small change in the symmetrised Laplacian, $\Delta L_{sym} = L'_{sym} - L_{sym}$. If the process is structurally fragmenting, defined as satisfying the condition $\mathbf{v}_2^T(\Delta L_{sym})\mathbf{v}_2 < 0$, where $\mathbf{v}_2$ is the Fiedler eigenvector of the initial graph $G$, then the Fiedler value will decrease ($\lambda'_2 < \lambda_2$).*

*Proof Sketch.* The proof relies on first-order matrix perturbation theory, which states that the change in an eigenvalue, $\Delta\lambda_k$, can be approximated by the quadratic form $\mathbf{v}_k^T(\Delta L_{sym})\mathbf{v}_k$. By applying this principle to the Fiedler value ($k = 2$), we find that $\lambda_2' - \lambda_2 \approx \mathbf{v}_2^T(\Delta L_{sym})\mathbf{v}_2$. The theorem's premise is precisely that the term on the right-hand side is negative. Therefore, it follows directly that for small perturbations, $\lambda_2' - \lambda_2 < 0$. The full proof is provided in §A.5. □

**Interpretation.** This theorem provides a nuanced quantitative insight about resource loss in physical systems. It formalises the critical scientific idea that the location of a disturbance is as important as its magnitude. The Fiedler eigenvector, $\mathbf{v}_2$, identifies the network's primary structural vulnerability or "fault line". This theorem proves that a process, even a dissipative one, only harms the network's algebraic connectivity if it is "aligned" with this vulnerability—that is, if it preferentially weakens the crucial links that bridge the network's main communities. For example, a disease affecting a keystone predator that connects two sub-webs would be structurally fragmenting, causing a sharp drop in algebraic connectivity. In contrast, the loss of a peripheral species might have a negligible effect. This provides a valuable diagnostic tool for assessing the resilience of an ecosystem, allowing scientists to distinguish between benign and potentially catastrophic systemic changes.

### 4.4 The Stability-Spectrum Equivalence

Building upon the foundational link between physical processes and their spectral signatures, we now establish our main result. The following theorem provides a formal equivalence between the long-term dynamical stability of a network in the `Proc` category and the convergence of its geometric representation in the `Spec` category. This result allows our framework to function as a predictive tool, allowing the assessment of a system's stability through its observable algebraic properties.

**Theorem 5** (The Stability-Spectrum Equivalence). *A dynamic network sequence $(G_t)_{t=1}^{\infty}$ governed by dissipative processes converges to a stable state $G_\infty$ if and only if its corresponding sequence of spectral data (eigenvalues and eigenvectors of $L_{sym,t}$) converges to a stable limit.*

*Proof Sketch.* As this is an "if and only if" statement, we sketch the proof in two directions.

1. **Stability $\implies$ Spectral Convergence:** We first assume the system converges to a stable state, which means the sequence of weight matrices converges to a limit, $W_t \to W_\infty$. The map from a weight matrix $W$ to its symmetrised Laplacian $L_{sym}$ is continuous. Furthermore, the spectral decomposition of a symmetric matrix (its eigenvalues and eigenvectors) is a continuous function of the matrix entries. By the property of continuous functions, the convergence of the weight matrices ($W_t \to W_\infty$) implies the convergence of the Laplacians ($L_{sym,t} \to L_{sym,\infty}$), which in turn implies the convergence of their spectral data.

2. **Spectral Convergence $\implies$ Stability:** We now assume the full set of spectral data (all eigenvalues and eigenvectors) converges. A symmetric matrix is uniquely determined by its spectral decomposition. Therefore, the convergence of the spectral data implies the convergence of the sequence of symmetrised Laplacians, $L_{sym,t} \to L_{sym,\infty}$. The mapping from a weight matrix $W$ to $L_{sym}$ is injective for a given graph topology. Thus, the convergence of the Laplacians implies the convergence of the underlying weight matrices, $W_t \to W_\infty$, which is the definition of a stable state.

The full proof is provided in §A.6. □

**Interpretation.** This theorem establishes a rigorous and testable equivalence between a system's physical behaviour and its abstract structural properties. In practical terms, it means that the stability of a complex ecosystem or biological network can be definitively assessed by monitoring its "spectral signature". If the eigenvalues and eigenvectors of the system's Laplacian stop changing, the system has reached an equilibrium. This moves beyond correlation to a formal equivalence, providing a diagnostic tool. For example, ecologists can use time-series data to determine if a recovering ecosystem has truly stabilised or if it is still in a transient state, simply by observing whether its spectral representation has converged. This provides a non-invasive method for understanding the long-term trajectory and health of complex systems.

### 4.5 A Toolkit for Process Interpretation

The preceding theorems establish the foundational properties of our framework. We now derive a toolkit of more specialised results that provide concrete, interpretable matrix signatures for common types of processes. These theorems allow a scientist to move from observing a spectral transformation back to inferring the specific nature of the underlying physical process that caused it, providing a powerful method for causal inference and system diagnostics.

**Lemma 2** (The Change of Basis Formula). *Let $p : G \to G'$ be a process, with $\{\mathbf{v}_i\}$ and $\{\mathbf{v}'_j\}$ being the orthonormal eigenbases of the initial and final Laplacians, respectively. The entry $(i, j)$ of the matrix representation of the linear transformation $\chi(p)$ is given by the inner product of the respective basis vectors: $(\chi(p))_{ij} = \langle \mathbf{v}'_i, \mathbf{v}_j \rangle$.*

*Proof Sketch.* The matrix for the linear transformation $\chi(p)$ represents the change of coordinates from the initial basis $B = \{\mathbf{v}_i\}$ to the final basis $B' = \{\mathbf{v}'_j\}$. The $j$-th column of this matrix is the vector $\mathbf{v}_j$ expressed in the coordinates of the new basis. Since $B'$ is an orthonormal basis, the $i$-th coordinate is simply the projection of $\mathbf{v}_j$ onto $\mathbf{v}'_i$, which is given by their inner product. The full proof is provided in §A.7. $\square$

**Interpretation.** This lemma provides the direct algebraic formula for computing the transformation matrix. It gives a precise meaning to each entry: the magnitude of $(\chi(p))_{ij}$ quantifies the alignment or projection of the $j$-th original structural mode onto the $i$-th final structural mode. A large diagonal entry $(\chi(p))_{ii}$ signifies that a structural mode has been preserved, while a large off-diagonal entry $(\chi(p))_{ij}$ signifies a significant structural "rewiring" where the roles of two modes have become mixed.

**Theorem 6** (The Rank-One Update Signature). *Let $p$ be a simple process that only perturbs the weight of a single edge between nodes $a$ and $b$. The resulting change in the symmetrised Laplacian, $\Delta L_{sym}$, is a rank-one matrix. Consequently, the transformation matrix $\chi(p)$ is a low-rank perturbation of the identity matrix.*

*Proof Sketch.* A change in the weight of a single edge $(a, b)$ results in a change matrix $\Delta L_{sym}$ with non-zero entries only at positions $(a, a), (b, b), (a, b)$, and $(b, a)$. As shown in the full proof, this matrix can be expressed as a scalar multiple of a single outer product, $\delta(\mathbf{e}_a - \mathbf{e}_b)(\mathbf{e}_a - \mathbf{e}_b)^T$, making it a rank-one matrix. By matrix perturbation theory, a low-rank update to a matrix results in a correspondingly simple, low-rank perturbation to its spectral decomposition. Therefore, the change of basis matrix $\chi(p)$ will be close to the identity, differing only by a low-rank update related to the eigenvector components at the perturbed nodes. The full proof is provided in §A.8. $\square$

**Interpretation.** This theorem proves that a local cause has a local signature. It formalises the intuitive idea that a small, isolated event in an ecosystem (e.g., the weakening of a single predator-prey relationship) should not cause a catastrophic, chaotic rewiring of the entire system's structure. It provides a diagnostic tool: if an observed transformation matrix $\chi(p)$ is well-approximated by a low-rank perturbation of the identity, one can infer that the underlying physical cause was a simple, localised process.

**Theorem 7** (The Structural Inertia Theorem). *Let $p$ be a process that induces a small perturbation $\Delta L_{sym}$. The resulting transformation matrix $\chi(p)$ is diagonally dominant. The magnitude of its off-diagonal entries is bounded by the norm of the perturbation and the spectral gaps of the original graph.*

*Proof Sketch.* The proof relies on the Davis-Kahan theorem from matrix perturbation theory. The entries of $\chi(p)$ are the inner products $\langle \mathbf{v}'_i, \mathbf{v}_j \rangle$. For $i = j$, this value is close to 1. For $i \neq j$, the Davis-Kahan theorem provides a bound on the sine of the angle between the old eigenvector $\mathbf{v}_j$ and the new one $\mathbf{v}'_i$, showing that this angle remains close to orthogonal. This deviation from orthogonality, which determines the magnitude of the off-diagonal entries, is bounded by $\|\Delta L_{sym}\|_2 / |\lambda_i - \lambda_j|$. Thus, for a small perturbation, the off-diagonal entries are small, and the matrix is diagonally dominant. The full proof is provided in §A.9. $\square$

**Interpretation.** This theorem formalises the concept of structural robustness. It proves that complex systems possess a form of inertia; their fundamental organisational modes (the eigenvectors) are resistant to small, arbitrary changes. A small process cannot cause a catastrophic re-shuffling of all the primary structural modes. A transformation matrix $\chi(p)$ with large off-diagonal entries is therefore a clear signature of a major, non-perturbative structural reorganisation of the network, rather than a simple fluctuation.

We now consider processes that alter the network's topology by removing a node. While such processes fall outside our primary functorial definition, which maps between spaces of the same dimension, our geometric approach still provides a unique and powerful characterisation. As we prove in §A.10, node removal is uniquely identifiable as a projection, whose kernel corresponds directly to the removed node, providing a distinct signature for this class of major topological changes.

**Theorem 8** (The Node Removal Signature). *Let $p$ be a process that removes a node $k$ from a network $G$ with $n$ nodes. The resulting transformation $\chi(p)$ maps the original $n$-dimensional eigenspace to the new $(n-1)$-dimensional eigenspace and is a projection operator.*

*Proof Sketch.* The process of node removal transforms the original vector space $\chi(G) \cong \mathbb{R}^n$ to a lower-dimensional space $\chi(G') \cong \mathbb{R}^{n-1}$. Such a transformation is a projection. The kernel of this projection (the part of the space mapped to zero) is the one-dimensional subspace corresponding to the removed node. The Cauchy Interlacing Theorem guarantees a predictable relationship between the old and new eigenvalues, ensuring the transformation is well-behaved. The full proof is provided in §A.10. □

**Interpretation.** This theorem provides a unique, identifiable signature for a major topological event: the complete failure or removal of a system component. Unlike the subtle changes from weight perturbations, a node removal causes a change in the very dimension of the state space. If an observed transformation matrix is found to be a projection of rank $n-1$, one can infer with high confidence that the underlying physical process was the removal of a single node. This is a simple tool for diagnosing critical failures in a system, such as the extinction of a species from an ecosystem.

**Theorem 9** (The Signal Transport Theorem). *Let $\mathbf{f}$ be a vector representing a "signal" on the nodes of a network $G$. After a process $p$, the signal $\mathbf{f}'$ on the new network $G'$ that maintains the same coordinates with respect to the new eigenbasis is given by the transformation $\mathbf{f}' = T_{transport}\mathbf{f}$, where the transport matrix is $T_{transport} = V'V^T$, with $V$ and $V'$ being the matrices of eigenvectors for $G$ and $G'$, respectively.*

*Proof Sketch.* We define the signal's spectral coordinates on $G$ as $\mathbf{a} = V^T\mathbf{f}$, where $V$ is the matrix of eigenvectors. The transported signal on $G'$ is defined as $\mathbf{f}' = V'\mathbf{a}$. Substituting the expression for $\mathbf{a}$ gives $\mathbf{f}' = (V'V^T)\mathbf{f}$, so $T_{transport} = V'V^T$. The full proof is provided in §A.11. □

**Interpretation.** This theorem provides a concrete, practical tool for prediction. It answers the question: "If the network structure changes from $G$ to $G'$, how would a data pattern defined on the original structure be expressed on the new structure?" For instance, it allows scientists to predict how a specific pattern of gene expression would be re-distributed across a cell if the underlying gene-regulatory network is rewired by a mutation. This moves the framework beyond structural analysis to a tool for predicting the evolution of functional patterns on the network.

### 4.6 From `Spec` to `Func`: Projecting the Spectral Transformation to a Functional Subspace

The theoretical toolkit developed in §4.5 provides rigorous, interpretable signatures for elementary processes. Nevertheless, the primary object of our analysis, the spectral transformation $\chi(p)$ itself, remains a high-dimensional $n \times n$ matrix. For complex, real-world systems where $n$ is large (e.g., $n = 161$ in our ecological study in §6 or $n = 379$ in our neuroscience study §7), this spectral-to-spectral map is mathematically complete but can be scientifically difficult to interpret. A scientist is typically not concerned with the raw $n^2$ interactions between abstract spectral modes, but with the dynamics between a much smaller number, $k$, of pre-defined, physically-meaningful partitions (e.g., a small number of functional networks in a human brain or a few trophic levels in an ecosystem). To bridge this interpretation gap, we introduce a principled method to compress the $n \times n$ spectral transformation $\chi(p)$ into an interpretable, $k \times k$ functional transformation

matrix, $M(p)$. This $M(p)$ provides a "functional fingerprint" of the process, revealing the average spectral transformation between the pre-defined groups.

This projection is built by relating the $n$-dimensional node space (of brain regions or species) to the $k$-dimensional functional space (of network partitions).

- **The Partition Matrix ($P$):** We first define the known functional structure using a static partition matrix $P \in \mathbb{R}^{n \times k}$, where $P_{ik} = 1$ if node $i$ (e.g., a brain region) belongs to functional group $k$ (e.g., the Default Mode Network), and 0 otherwise. This matrix $P$ acts as an "embedding" map from the $k$-dimensional functional space to the $n$-dimensional node space.

- **The Functional Projector ($\rho$):** We define the corresponding projection map $\rho \in \mathbb{R}^{k \times n}$ that performs the reverse operation: mapping from the $n$-dimensional node space down to the $k$-dimensional functional space. For this, we use the pseudo-inverse of $P$:

$$\rho = P^\dagger = (P^T P)^{-1} P^T \tag{13}$$

The term $(P^T P)$ is a simple $k \times k$ diagonal matrix whose entries $N_k$ are the number of nodes in each group, making its inverse trivial to compute. The resulting projector $\rho$ effectively performs a "group averaging" operation.

While our transformation $\chi(p)$ operates in the $n$-dimensional spectral space, we can use the node-space projectors $P$ and $\rho$ to construct its $k$-dimensional functional-space equivalent. We do this by assuming a meaningful correspondence between the $n$-dimensional node basis and the $n$-dimensional spectral basis. We then apply a **Galerkin Projection**, a standard and optimal method for model reduction, to compute the $k \times k$ matrix $M(p)$ that best represents the action of the $n \times n$ operator $\chi(p)$ relative to this functional subspace.

**Theorem 10** (Direct Spectral-to-Functional Projection). *Let $\chi(p) \in \mathbb{R}^{n \times n}$ be the spectral transformation matrix. Let $P \in \mathbb{R}^{n \times k}$ be the partition matrix mapping $k$ functional groups to the $n$ nodes, and let $\rho \in \mathbb{R}^{k \times n}$ be its pseudo-inverse projector. The $k \times k$ functional transformation matrix $M(p)$ is the unique solution that minimises the Frobenius norm of the approximation error $||\chi(p) - PM(p)\rho||_F^2$, and is given by the Galerkin projection:*

$$M(p) = \rho \chi(p) P \tag{14}$$

*Proof Sketch.* The proof is a constructive application of the Galerkin projection method. The matrix $P$ defines the "input" basis vectors for the $k$-dimensional functional subspace (embedding $k$ functional groups into the $n$-dimensional space of input modes), and $\rho$ defines the "output" basis vectors (projecting the $n$-dimensional space of output modes back to $k$ functional groups). The formula $M(p) = \rho \chi(p) P$ is the standard computation for finding the $k \times k$ linear operator $M$ that best approximates the $n \times n$ operator $\chi$ with respect to these input and output bases. It effectively compresses the $n \times n$ operator by viewing it only through the $k$-dimensional lens of the functional groups. The full derivation is provided in §A.12. □

**Interpretation.** This theorem provides a principled and computationally efficient 'glass-box' tool for model reduction and interpretation. It proves that the functional transformation matrix, $M(p)$, is not an ad-hoc visualisation but is the optimal $k \times k$ representation of the full $n \times n$ spectral transformation $\chi(p)$, as viewed from the perspective of a pre-defined physical partition. This has broad implications. Interpreting an $n \times n$ $\chi(p)$ matrix is often intractable. This theorem provides a formal method to compress it into a much smaller $k \times k$ matrix $M(p)$ that is both quantitatively optimal and scientifically interpretable. The entry $M(p)_{ab}$ has a clear, physical meaning: it is the **average spectral influence** that the original structural modes associated with functional group $b$ (e.g., Visual Network) exert on the new structural modes associated with functional group $a$ (e.g., Dorsal Attention Network). This provides a rigorous foundation for systems-level analysis, allowing for the creation of low-dimensional fingerprints of complex processes, as we demonstrate in our neuroscience case study (§7, Figure 18).

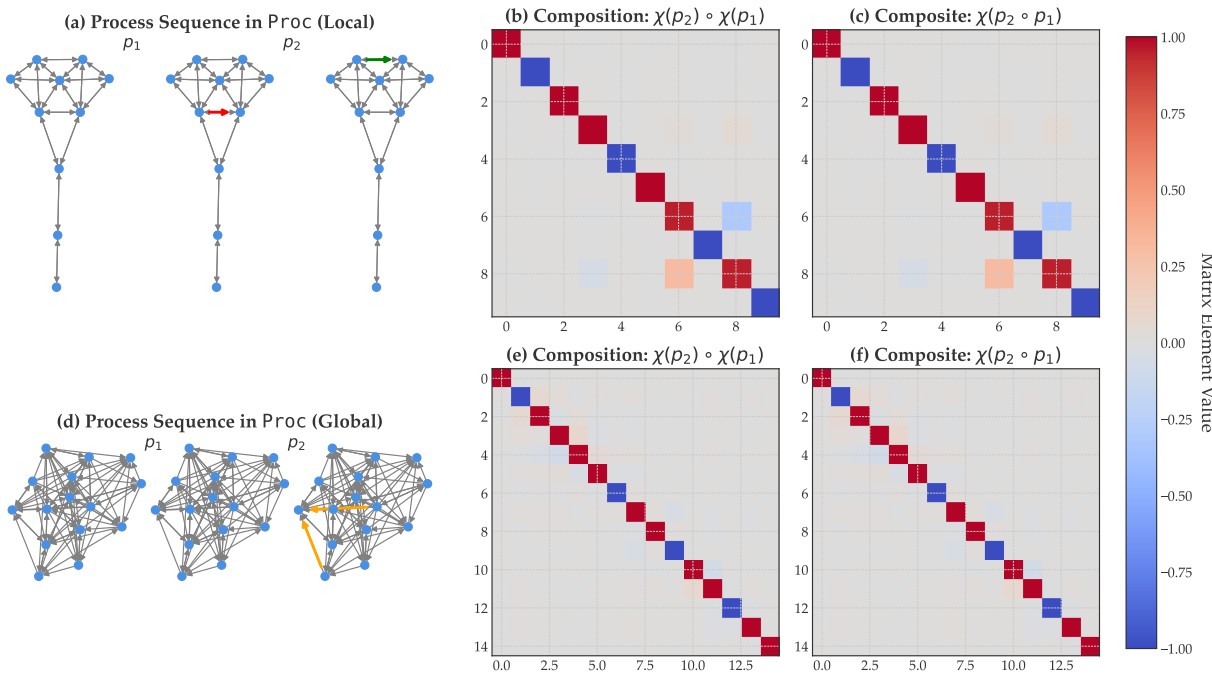

Figure 2: **Numerical validation of the functoriality of $\chi$ (Lemma 1).** This figure provides visual and quantitative proof that our framework respects the composition of processes for both simple, local changes (top row) and complex, global changes (bottom row). **(a)** A sequence of three network states in Proc connected by two simple processes, $p_1$ and $p_2$, each a single edge perturbation. The modified edge in each step is highlighted (red for decreased weight, green for increased). **(b)** The resulting transformation matrix for the composition of the two processes, computed as the matrix product $\chi(p_2) \circ \chi(p_1)$. **(c)** The transformation matrix for the single composite process, $\chi(p_2 \circ p_1)$. This matrix is visually and numerically identical to the one in (b). **(d)** A sequence of network states connected by two complex, global processes: a dissipative process ($p_1$) that modifies all edges, followed by a conservative one ($p_2$). **(e)** The matrix for the composition of the complex processes, showing a non-trivial, global transformation. **(f)** The matrix for the single composite complex process, which is again identical to its compositional counterpart in (e). The numerical difference between the composed and composite matrices is negligible in both cases (Frobenius norm for the simple case: $3.3 \times 10^{-15}$; for the complex case: $4.2 \times 10^{-15}$).

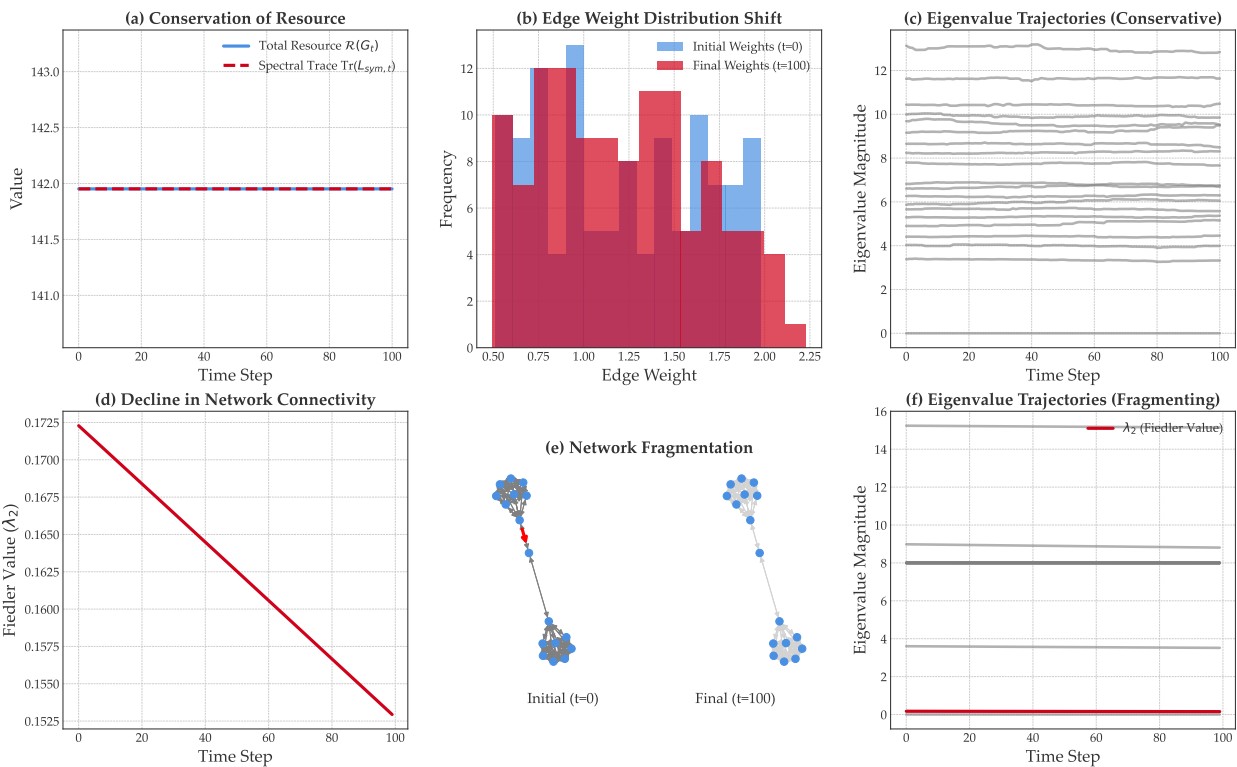

Figure 3: **Numerical validation of the physics-to-spectra dictionary (Theorems 3 and 4).** This figure demonstrates that the physical constraints defined in `Proc` have direct, predictable consequences in `Spec`. **Top Row**: A network was subjected to a purely conservative process for 100 steps. **(a)** The total resource $\mathcal{R}(G_t)$ and the spectral trace $\text{Tr}(L_{sym,t})$ are plotted over time. The lines are perfectly flat and coincident, numerically verifying the identity $\text{Tr}(L_{sym}) = \mathcal{R}(G)$ and proving that the spectral trace is a conserved quantity under conservative dynamics. **(b)** Histograms of the initial and final edge weights show that while the process has significantly redistributed the individual resource values, the total resource is unchanged. **(c)** The trajectories of all eigenvalues are shown. While individual structural modes fluctuate, their sum is perfectly constant, as proven in (a). **Bottom Row**: A barbell graph was subjected to a structurally fragmenting process for 100 steps. **(d)** The Fiedler value ($\lambda_2$), a measure of algebraic connectivity, shows a clear and monotonic decline, confirming that the process successfully fragments the network. **(e)** Visualisation of the initial network with its critical bridge edge highlighted (left) and the final network (right), where the weakening of the bridge has caused the two communities to drift apart. **(f)** The trajectories of all eigenvalues, with the Fiedler value highlighted. The plot shows $\lambda_2$ "peeling away" from the rest of the spectrum and dropping towards zero, a classic signature of network fragmentation.

# 5 Numerical Validation

In this section, we present a suite of controlled experiments on synthetic data to provide a rigorous validation of our theoretical claims from §4. We begin by numerically verifying the functoriality of our spectral map, confirming that it respects the composition of processes (§5.1). We then validate the core physics-to-spectra dictionary, providing evidence for both the Spectral Trace Conservation Law and the Spectral Sensitivity of connectivity (§5.2). Building on this, we provide a multi-faceted validation of our main result, the Stability-Spectrum Equivalence theorem (§5.3). Further, we demonstrate the diagnostic power of our framework by verifying the unique spectral signatures predicted by our process interpretation toolkit (§5.4). Finally, we validate the predictive power of our framework using signal transport (§5.5).

## 5.1 Verification of Functoriality

The cornerstone of our entire framework is the claim that the spectral map $\chi$ is a valid functor (Lemma 1). To verify this, we must show that it respects the composition of processes. We test this property in two scenarios: one with simple, local processes and one with complex, global processes. As shown in Figure 2, in both the simple case (Panel a-c) and the complex case (Panel d-f), the transformation matrix computed for the single composite process is numerically (almost) identical to the matrix product of the individual transformations. This provides clear, strong evidence that our framework is mathematically sound and that our "spectral accounting" of change is consistent and reliable.

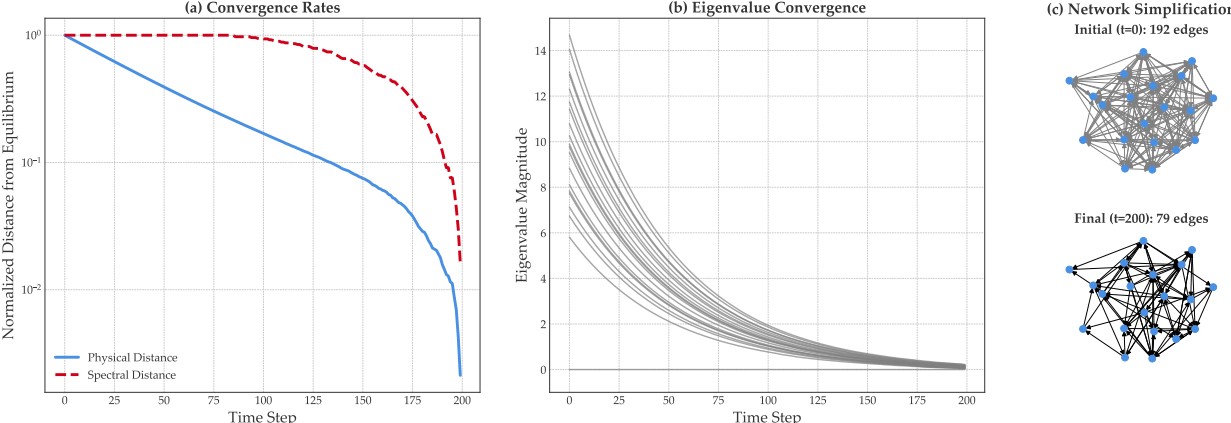

Figure 4: **Numerical validation of the Stability-Spectrum Equivalence (Theorem 5).** This figure provides a multi-faceted demonstration of the equivalence between a system's physical convergence to equilibrium and the stabilisation of its spectral representation. A randomly generated network was subjected to a dissipative process with edge pruning for 200 time steps. **(a)** The convergence rates of the physical and spectral representations. The log-linear plot shows that both the physical distance from equilibrium (blue, solid) and the spectral distance (red, dashed) decay to zero, confirming that the system stabilises as its spectral representation does. The differing decay profiles reveal a non-trivial insight: the spectral representation is less sensitive to initial fluctuations but provides a sharper signal of the final convergence to a stable state. **(b)** The convergence of the full Laplacian spectrum. The plot shows the trajectories of all eigenvalues over time. The initially chaotic dynamics smoothly resolve into a set of stable, horizontal lines, providing a holistic visualisation of the system's entire geometric structure settling into its final, equilibrium configuration. **(c)** The physical process of network simplification. The visualisation shows the initial, dense network state at $t = 0$ (left) and the final, sparse equilibrium state at $t = 200$ (right). The dissipative and pruning processes have driven the system to shed non-essential connections, converging from 192 edges to a stable "backbone" of 79 edges.

## 5.2 The Spectral Signatures of Physical Constraints

Next, we validate our "physics-to-spectra dictionary", which links the physical constraints of the `Proc` category to specific signatures in the `Spec` category. Figure 3 shows the results of two simulations designed to test these links.

The top row validates our Trace Conservation Law (Theorem 3). We subject a network to a purely conservative process that redistributes resources internally. As predicted, the total resource and the spectral trace remain perfectly constant and coincident throughout the simulation (Panel a), even as the individual edge weights are shuffled (Panel b) and the individual eigenvalues fluctuate (Panel c).

The bottom row validates our Spectral Sensitivity theorem (Theorem 4). We subject a barbell graph, which has a clear structural bottleneck, to a structurally fragmenting process. As predicted, the Fiedler value ($\lambda_2$), a measure of algebraic connectivity, shows a clear and monotonic decline (Panel d). The physical meaning of this is made clear in Panel (e), which shows the network visually breaking apart. The full spectrum dynamics in Panel (f) provide a deeper insight, showing the Fiedler value "peeling away" from the rest of the spectrum, a classic signature of fragmentation.

## 5.3 Equivalence of Physical and Spectral Stability

Another theoretical result of our framework is the equivalence between the physical stability of a system and the stability of its spectral representation. To validate this, we simulate a network's long-term evolution under a dissipative process with edge pruning, allowing it to converge to a stable equilibrium. The results, shown in Figure 4, provide a multi-faceted confirmation of the theorem.

Panel (a) is the core quantitative proof. It shows that the physical distance from equilibrium (blue, solid) and the spectral distance (red, dashed) both decay to zero, confirming that the system stabilises if and only if its spectral representation does. The differing decay profiles reveal a non-trivial insight: the spectral representation is a more sensitive indicator of the final convergence. Panel (b) provides a holistic view, showing the entire spectrum of the system converging to stable, horizontal lines. Finally, Panel (c) provides the intuitive physical implication, showing the network simplifying from a dense, chaotic initial state to a sparse, stable "backbone" structure.

## 5.4 Diagnostic Power of the Toolkit

A key claim of our work is that elementary processes have unique, identifiable spectral signatures. Figure 5 validates this diagnostic power. The top row shows that a simple, local edge perturbation results in a transformation matrix that is strongly diagonally dominant (Panel b) , supporting Theorem 7. We quantify this using the Diagonal Dominance Ratio, defined as the sum of the absolute values of the diagonal entries divided by the sum of the absolute values of all entries:

$$R_{DD}(\mathbf{M}) = \frac{\sum_i |M_{ii}|}{\sum_{i,j} |M_{ij}|} \tag{15}$$

For the matrix $\chi(p_{pert})$ in Panel (b), this ratio is 0.9641, indicating that over 96% of the matrix's "mass" is concentrated on the diagonal. The matrix also represents a sparse perturbation of the identity matrix (Panel c). While the underlying Laplacian change ($\Delta L_{sym}$) is rank-1 (Theorem 6), the spectral perturbation matrix $\chi(p_{pert}) - I$ shown in Panel (c) has a numerical rank of 5. Its top singular values are [2.0000, 2.0000, 2.0000, 1.9957, 1.9957, 0.0000], indicating the change is concentrated in a low-dimensional subspace, consistent with a localised physical process. This confirms that local changes have a simple, structured spectral signature. In contrast, the bottom row shows that a major topological change—a node removal— produces a completely different signature. The transformation is a projection operator (Panel e), which is confirmed by its eigenvalue spectrum of only 0s and 1s (Panel f). This demonstrates that our framework can unambiguously distinguish between different classes of physical events.

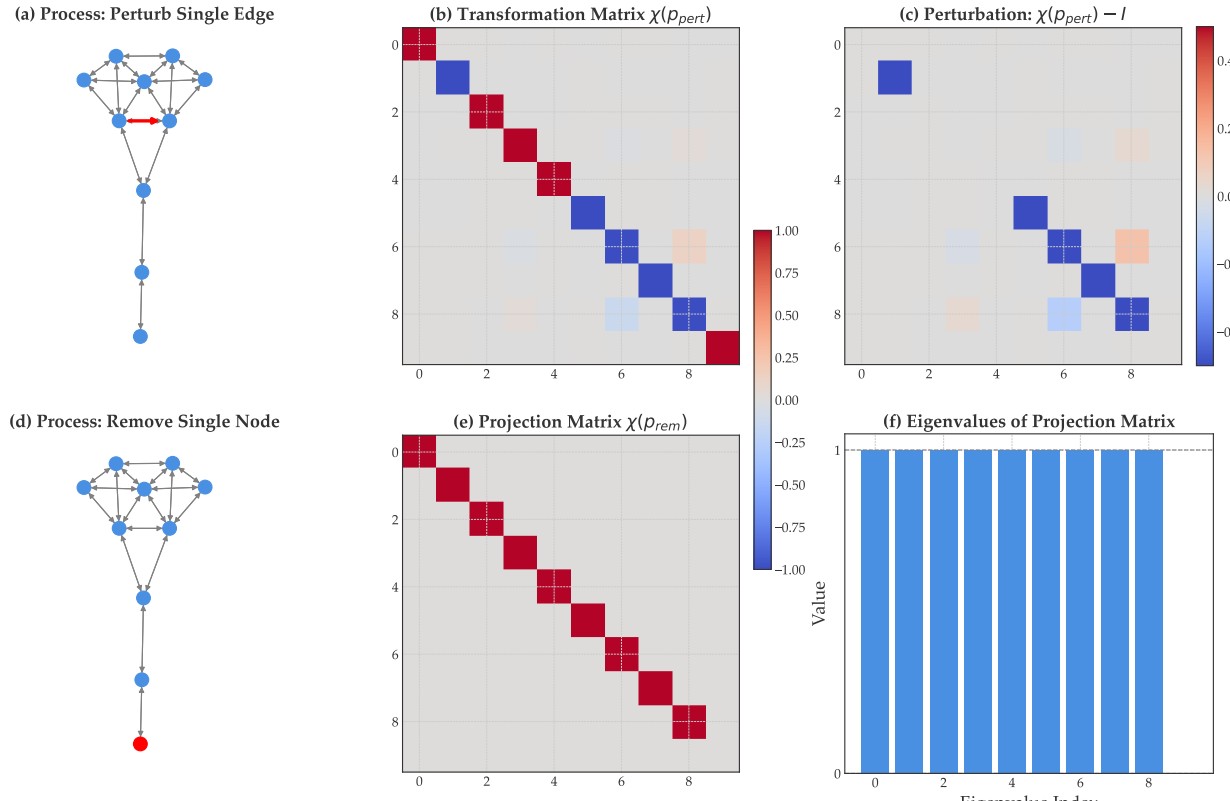

Figure 5: **Numerical validation of the process interpretation toolkit (Theorems 6, 7, and 8).** This figure demonstrates that different elementary processes have unique, identifiable spectral signatures, confirming the diagnostic power of our framework. **Top Row**: Signature of a Local Perturbation. **(a)** The physical process: a single edge (highlighted in red) on a network is perturbed. **(b)** The resulting transformation matrix $\chi(p_{pert})$. The matrix is strongly diagonally dominant, quantified by a Diagonal Dominance Ratio of 0.9641, visually confirming the principle of Structural Inertia (Theorem 7). **(c)** The perturbation matrix, $\chi(p_{pert}) - I$. While the underlying Laplacian change $\Delta L_{sym}$ is rank-1 (Theorem 6), the spectral perturbation shown here has a numerical rank of 5. Its top singular values are [2.0000, 2.0000, 2.0000, 1.9957, 1.9957, 0.0000], indicating the change is concentrated in a low-dimensional subspace, consistent with a localised physical process. **Bottom Row**: Signature of Node Removal. **(d)** The physical process: a single node (highlighted in red) is removed from the network. **(e)** The matrix of the resulting projection operator. Its signature—an identity matrix with a single zero on the diagonal—is visually distinct from the perturbation signature in (b). **(f)** The eigenvalues of the projection matrix. The bar plot provides a quantitative validation of the Node Removal Signature (Theorem 8), showing exactly one eigenvalue at 0 (for the removed dimension) and all others at 1.

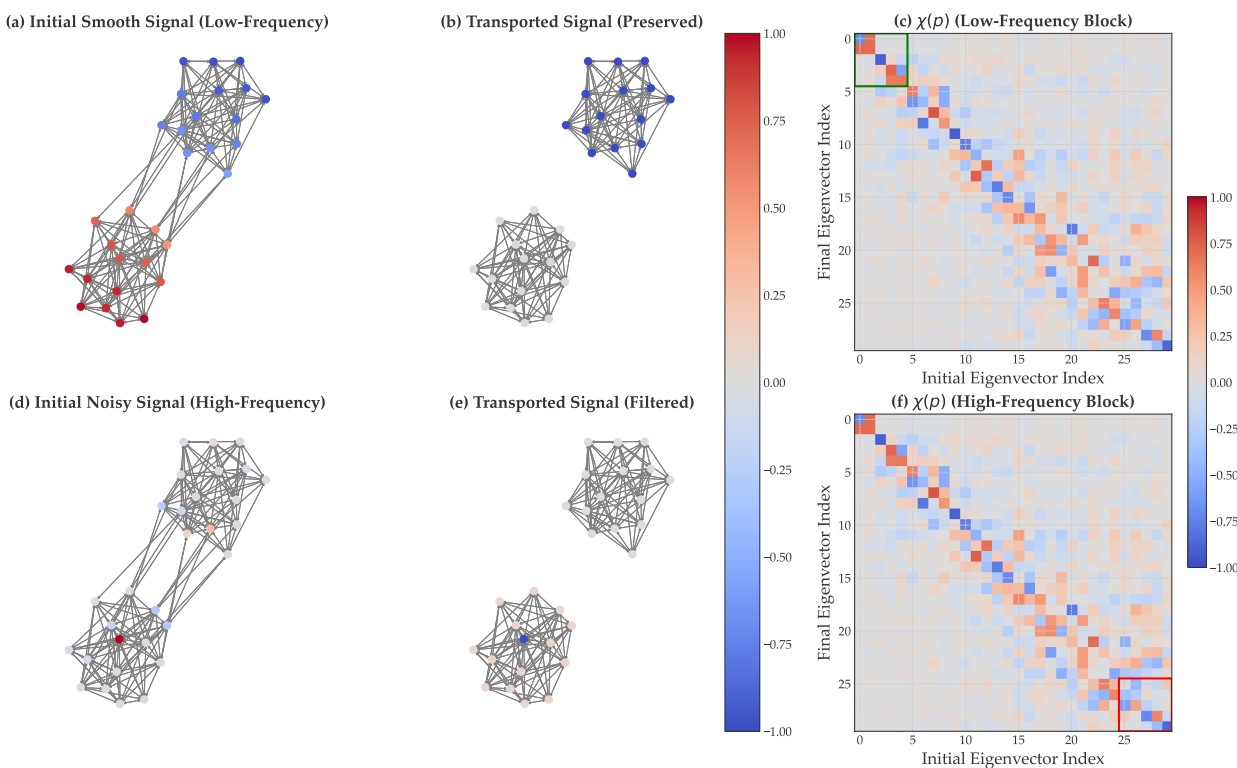

Figure 6: **Demonstration of Signal Transport as a Spectral Filter (Theorem 9).** This experiment reveals that a major structural change acts as a low-pass filter on network patterns, preserving smooth, low-frequency signals while destroying noisy, high-frequency ones. The process in both scenarios is the severing of all bridge edges between the two communities of a Stochastic Block Model graph. **Top Row**: Preservation of a Low-Frequency Signal. **(a)** The initial signal is a smooth, low-frequency pattern (the Fiedler eigenvector of the initial graph), creating a clear gradient between the two communities. **(b)** After transport to the new, disconnected graph, the smooth pattern is closely preserved within each community (Pearson correlation between initial and transported signal: $r = 0.9846$, $p < 10^{-22}$). **(c)** The transformation matrix $\chi(p)$ provides the mechanism. The highlighted top-left (low-frequency) block is strongly diagonal (Block Diagonal Dominance Ratio = 0.5251), proving that low-frequency modes of the initial graph map cleanly to the low-frequency modes of the final graph, ensuring pattern preservation. **Bottom Row**: Filtering of a High-Frequency Signal. **(d)** The initial signal is a noisy, high-frequency pattern (the last eigenvector of the initial graph). **(e)** After transport, the noisy pattern is destroyed and radically altered (Pearson correlation between initial and transported signal: $r = -0.8724$, $p < 10^{-9}$). **(f)** The transformation matrix $\chi(p)$ again reveals the mechanism. The highlighted bottom-right (high-frequency) block is scattered and non-diagonal (Block Diagonal Dominance Ratio = 0.3250). This visually demonstrates that high-frequency modes of the initial graph do not map cleanly to high-frequency modes of the final graph; their energy is scattered, destroying the original pattern.

### 5.5 Signal Transport as a Spectral Filter

Our final synthetic experiment demonstrates that our framework can predict the fate of patterns on a changing network. The results in Figure 6 show that a major structural change acts as a spectral low-pass filter. The top row shows the result of pattern preservation. A smooth, low-frequency signal (Panel a) is successfully transported to the new network with its structure closely preserved (Panel b)(Pearson correlation between initial and transported signal: $r = 0.9846$, $p < 10^{-22}$). The transformation matrix $\chi(p)$ (Panel c) reveals the mechanism: its top-left, low-frequency block is strongly diagonal, showing that the geometric language for smooth patterns is robust to the change, quantified by applying the Diagonal Dominance Ratio definition (Equation 15) to this $5 \times 5$ block, yielding a value of 0.5251. The bottom row visualises pattern destruction. A noisy, high-frequency signal (Panel d) is almost completely destroyed by the transport, collapsing to a closely uniform, low-energy state (Panel e); the destruction is quantified by a strong negative correlation between the initial and transported signals ($r = -0.8724$, $p < 10^{-9}$). The transformation matrix (Panel f) again reveals why: its bottom-right, high-frequency block is scattered and non-diagonal, proving that the geometric language for noisy patterns has been fundamentally broken by the process, reflected in a low Block Diagonal Dominance Ratio (calculated via Equation 15 on this $5 \times 5$ block) of 0.3250. This experiment confirms that our framework provides a tool for predicting not just *that* a pattern will change, but *how* it will change, based on its alignment with the network's underlying spectral geometry.

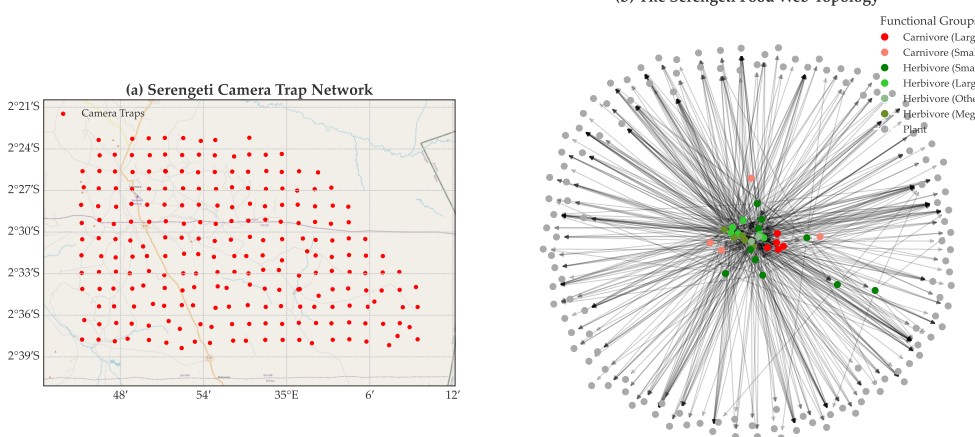

Figure 7: **The Serengeti Case Study Area.** This figure provides an overview of the two primary datasets used in our real-world case study. **(a)** A map of the Serengeti National Park showing the geographic distribution of the 225 camera traps from the Snapshot Serengeti project (Swanson et al., 2015). The camera grid covers an area of 1,125 km$^2$, and each point represents a unique camera site, plotted using its UTM coordinates. The basemap provides topographical context for the study area. **(b)** The full topological structure of the Serengeti food web, based on the data from Baskerville et al. (2011). The network consists of 161 species (nodes) and 592 directed feeding links (edges). Nodes are colour-coded by their functional group to illustrate the ecosystem's trophic structure: carnivores are shown in reds, herbivores in greens, and the broad plant base in grey. The high proportion of plant nodes reflects the high taxonomic resolution of the primary producer data, a key feature of this food web.

## 6 Real-World Case Study 1: Serengeti Ecosystem Dynamics and Drought Impacts

Having established the theoretical foundations of our `Proc-to-Spec` framework in §4 and validated its core properties with numerical experiments in §5, we now apply it to complex, real-world systems. To demonstrate the generality and robustness of our framework, we selected two comprehensive case studies from fundamentally different scientific domains, each presenting a distinct set of analytical challenges.

The first, presented in this section, is an analysis of the Serengeti food web (Baskerville et al., 2011). This case study serves as a test in a **high-signal, macro-timescale, and physically-grounded** setting. The objective here is to move from mathematical verification to scientific discovery, demonstrating our framework's power to interpret a known, massive event from high-resolution, inherently noisy ecological data. The core challenge in this domain is not signal *detection*, but signal *interpretation*: we must prove our framework is robust to the significant data sparsity of camera traps (Swanson et al., 2015) while successfully providing a novel, geometric characterisation of a system-wide shock. This analysis will serve as a direct methodological contrast to our second case study in §7, which will test the framework's sensitivity in a high-speed, low-signal, abstract cognitive system.

Table 2: Summary of the Serengeti 'Snapshot' and Food Web Datasets.

| Metric | Value | Source |
|---|---:|---|
| *Snapshot Serengeti Camera Trap Data* | | |
| Study Area | 1,125 km$^2$ | Swanson et al. (2015) |
| Number of Camera Traps | 225 | Swanson et al. (2015) |
| Time Period | 2010–2013 | Swanson et al. (2015) |
| Total Image Sets Classified | 1.2 million | Swanson et al. (2015) |
| *Food Web Topology Data* | | |
| Total Species (Nodes) | 161 | Baskerville et al. (2011) |
| Feeding Links (Edges) | 592 | Baskerville et al. (2011) |
| *Functional Groups* | | |
| Carnivores | 9 | Baskerville et al. (2011) |
| Herbivores | 23 | Baskerville et al. (2011) |
| Plants | 129 | Baskerville et al. (2011) |

## 6.1 The Serengeti Food Web Ecosystem

We have chosen the Serengeti ecosystem in northern Tanzania to be the testbed for our framework. It is a well-studied, resource-constrained system characterised by strong seasonal dynamics and occasional environmental shocks. Its complex food web and the availability of rich, modern datasets provide an valuable opportunity to test our framework's ability to analyse real-world network dynamics.

Our analysis is built upon two public datasets. The first is the "*Snapshot Serengeti*" dataset, a large collection of 1.2 million classified camera trap image sets from 225 camera traps spread across 1,125 km$^2$ of the Serengeti National Park from 2010 to 2013 (Swanson et al., 2015). This dataset provides a high-frequency, event-level record of animal activity. The second is a high-resolution food web topology, which details 592 predator-prey interactions among 161 species, notable for its high taxonomic resolution at the plant level (Baskerville et al., 2011). An overview of these datasets is provided in Figure 7 and Table 2.

## 6.2 Data Processing and Dynamic Network Construction

To apply our framework, we developed a rigorous data processing pipeline to transform the raw, event-level camera trap data (Swanson et al., 2015) into a time-series of dynamic networks in the `Proc` category. This pipeline consists of the following steps:

**Step 1: Event Aggregation and Cleaning** The foundational dataset, the *Snapshot Serengeti* consensus data, presents several data processing challenges. First, it is an event-level log, not an aggregated time-series. Second, and most critically, the 'Count' field is non-numeric, recorded as strings to represent uncertainty (e.g., '1', '2', '11-50', '51+').

As a foundational cleaning step, we implemented a robust parsing methodology to convert this raw log into a quantitative time-series. First, we ensured data integrity by filtering out any records missing essential fields (a valid timestamp, species identifier, or count string). Next, we defined a deterministic rule to convert

the count strings to integers: simple numeric strings were cast directly, ranges were mapped to their lower bound (e.g., '11-50' → 11), and open-ended ranges were mapped to their base value (e.g., '51+' → 51). Any unparseable entries were conservatively treated as zero.

With a clean, numeric count for each event, we then performed temporal aggregation. We parsed the 'DateTime' field of each event to extract its 'Year' and 'Month'. Finally, we grouped the entire dataset by 'Year', 'Month', and 'Species' and summed the numeric counts to produce a single, clean time-series: the total observed count for each species for each month of the study period.

**Step 2: Species Unification and Network Weighting**   The aggregated count time-series (from Step 1) must be mapped onto a static food web topology. This presented a significant data unification challenge: the camera trap counts use common species names (e.g., 'lionFemale', 'gazelleThomsons'), whereas the food web topology dataset (Baskerville et al., 2011) uses scientific names (e.g., 'Panthera leo', 'Eudorcas thomsonii'). A critical pre-processing step was to meticulously map all species from the count data to their corresponding scientific names in the topology, ensuring the two datasets could be algorithmically joined.

With the datasets unified, we constructed a dynamic network $G_t$ for each month $t$. The static node set (161 species) and edge set (592 feeding links) were taken from Baskerville et al. (2011). The dynamic edge weights $W_t(u, v)$ for a predator $u$ and prey $v$ were then computed using a well-established mass-action model in the literature (Murray, 2007):

$$W_t(u, v) = C \cdot \text{count}_t(u) \cdot \text{count}_t(v) \tag{16}$$

where $\text{count}_t(u)$ and $\text{count}_t(v)$ are the aggregated monthly counts for the predator and prey, respectively. Any interaction where either species had a count of zero resulted in a weight of zero. The scaling constant $C$ was set to $10^{-3}$ to scale the interaction strengths appropriately. This process resulted in a time-series of dynamic networks, each representing a monthly snapshot of the food web's effective interaction strengths.

**Step 3: Defining the Active Subgraph**   The resulting monthly networks $G_t$ are highly sparse, as a zero count for either predator or prey in a given month results in a $W_t(u, v) = 0$. To analyse the dynamics of the system, we must first isolate the components that are active in each snapshot. We defined the 'active subgraph' $G'_t$ as the subgraph containing only nodes and edges that participated in an interaction with a positive weight (i.e., where both predator and prey were observed that month).

This active subgraph $G'_t$ is often still disconnected. To focus on the core of the interacting system, we identified its main component by finding the Largest Connected Component (LCC). It is crucial to note that this LCC was identified from the undirected representation of $G'_t$, thus capturing the primary cluster of interacting species regardless of the direction of resource flow. This standard approach (Newman, 2018) ensures our analysis is robust to the transient disappearance of peripheral species and isolates the main, interacting component of the ecosystem for each month.

**Step 4: Ensuring a Stable Node Set for Transformation Analysis**   A key requirement of our framework for computing the transformation $\chi(p_t) : \chi(G_t) \to \chi(G_{t+1})$ is that both network states are defined over the same set of $n$ nodes, ensuring the transformation is a map between vector spaces of equal dimension. In a real-world dataset, data sparsity can cause peripheral nodes to 'blink' in and out of the active LCC (as defined in Step 3) from one month to the next. Analysing these changing node sets directly would introduce high-dimensional noise and artifacts.

To ensure a robust and stable analysis, we therefore computed all transformations on the largest common active component. This component is formally defined as the subgraph induced by the intersection of the node sets of the two consecutive LCCs: $V_{common} = V(\text{LCC}_t) \cap V(\text{LCC}_{t+1})$. All spectral analyses and transformation matrices were subsequently computed on this stable, common subgraph. This approach ensures our analysis is robust to observation noise and isolates the geometric changes of the core system.

To quantitatively assess the impact and robustness of this LCC-based approach, we analysed the coverage and temporal stability of the LCC throughout the study period (Figure 10). The results show that, for this dataset, the LCC consistently contained 100% of all nodes and resources involved in active interactions each month (Figure 10a), indicating that no interacting components were excluded by focusing on the LCC.

Furthermore, the node membership of the LCC was generally highly stable month-to-month, reflected in high overlap metrics (mean Jaccard Index = 0.939, Figure 10b) and low turnover rates (mean < 4%, Figure 10c). Significant LCC turnover (up to 40%) occurred primarily during the 2011 drought period, confirming the LCC's sensitivity to major ecological events while validating its use as a stable analytical core for most of the time series.

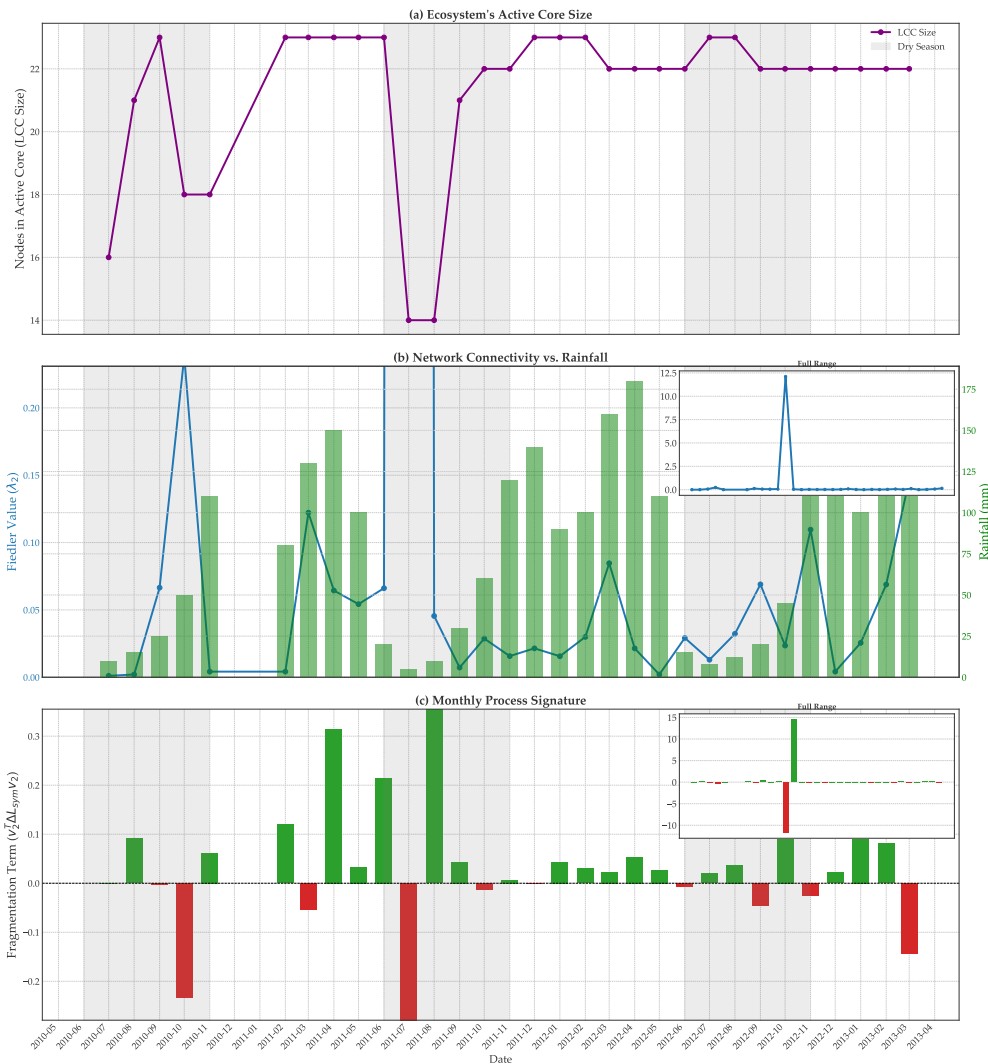

Figure 8: **Spectral Dynamics of the Serengeti Food Web (2010-2013).** The figure presents three metrics derived from the monthly network time-series, revealing a roughly consistent seasonal cycle and a major anomalous event. **(a) Active Core Size:** The number of species in the Largest Connected Component (LCC) shows a seasonal "pulse", vibrating between dry (shaded) and wet seasons. **(b) Network's Algebraic Connectivity:** The Fiedler value ($\lambda_2$) of the LCC is approximately anti-correlated with the dry seasons. The inset shows the full range, capturing a massive connectivity spike of the shrunken core during the July 2011 drought. **(c) Process Signature:** The fragmentation term reveals fragmenting processes (red bars) at the onset of dry seasons and consolidating processes (green bars) during recovery. The inset highlights the extreme nature of the 2011 event. Collectively, these results show that the framework is sensitive enough to detect both the subtle, recurring seasonal cycle and the multi-faceted signature of a major, documented drought event.

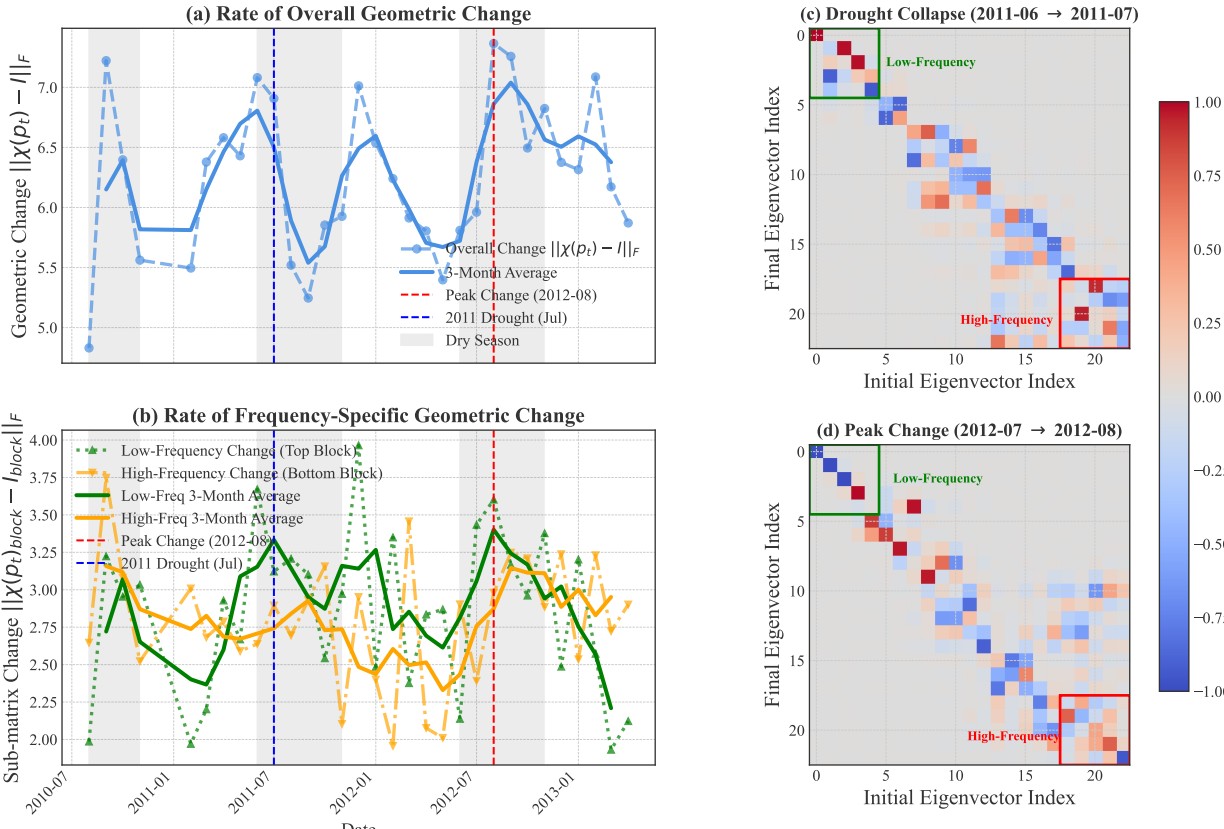

Figure 9: **Rate of Geometric Change in the Serengeti Food Web (2010-2013).** This figure quantifies the month-to-month geometric instability of the network's spectral structure. **(a)** The rate of overall geometric change, measured as the Frobenius distance $||\chi(p_t) - I||_F$ between consecutive months. Both the raw monthly value (dashed line) and a 3-month running average (solid line) are shown. The highest rate of change occurred in August 2012, while elevated rates are also observed during the 2011 drought period. Shaded regions indicate dry seasons. **(b)** The rate of geometric change decomposed into low-frequency (top 5x5 block of $\chi(p_t) - I$, green lines) and high-frequency (bottom 5x5 block, orange lines) spectral modes. High-frequency changes exhibit greater month-to-month volatility, while low-frequency changes show larger, slower swings that drive the major peaks. **(c, d)** Comparison of the $\chi(p)$ transformation matrices during two periods of high geometric instability: the Drought Collapse (June $\rightarrow$ July 2011) and the Peak Change period (July $\rightarrow$ Aug 2012). Both matrices are highly off-diagonal, indicating significant structural reorganisations, but exhibit qualitatively different patterns. Highlighted blocks correspond to the low-frequency and high-frequency components analysed in panel (b).

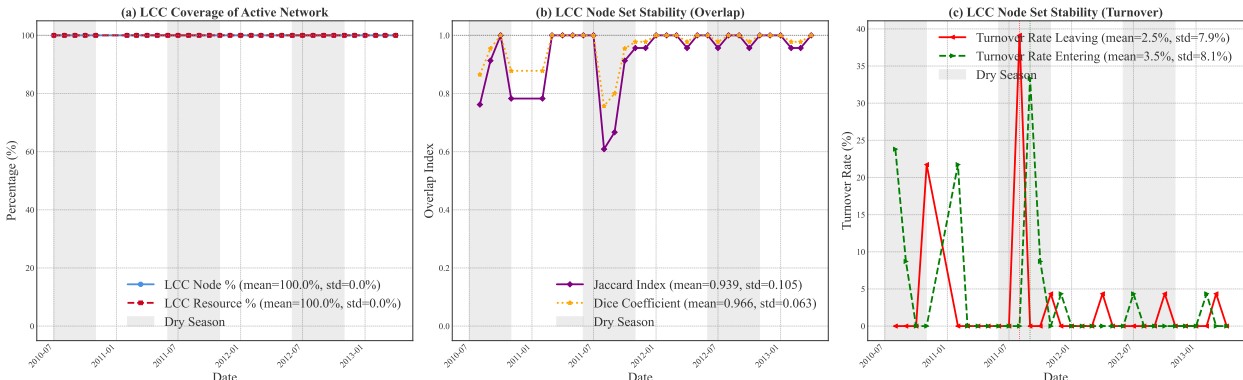

Figure 10: **LCC Coverage and Stability Analysis for the Serengeti Network (2010-2013).** This figure validates the use of the Largest Connected Component (LCC) by quantifying its coverage and temporal stability. **(a) LCC Coverage:** Plots the percentage of total active nodes (nodes with weight $> 0$) and total active resource (sum of weights $> 0$) contained within the LCC each month. In this dataset, the LCC consistently encompasses 100% of the active network components. Shaded regions indicate dry seasons. **(b) LCC Node Set Stability (Overlap):** Shows the month-to-month stability of the LCC node set using the Jaccard Index (mean=0.939, std=0.105) and Dice Coefficient (mean=0.966, std=0.063). Values near 1.0 indicate high stability, while dips (e.g., mid-2011) correspond to periods of greater structural flux. **(c) LCC Node Set Stability (Turnover):** Shows the percentage of nodes leaving the LCC (mean=2.5%, std=7.9%) and entering the LCC (mean=3.5%, std=8.1%) each month. While generally low, turnover rates peak significantly during the 2011 drought period, quantifying the disruption.

### 6.3 Experimental Results

Using the constructed monthly networks, here we conducted a series of experiments with each designed to validate a different core theorem of our framework.

**Detecting the Ecosystem's Pulse and a Major Ecological Shock.** Our first experiment tests the framework's sensitivity to environmental change. As shown in Figure 8, the analysis reveals a clear, cyclical pattern in the ecosystem's structure corresponding to the wet and dry seasons. The size of the active core (Panel a) and the network's algebraic connectivity (Panel b, Fiedler value) both consistently contract during the resource-scarce dry seasons. The framework also proves highly sensitive to extreme events, detecting a major anomaly in July 2011 that corresponds to a severe, well-documented regional drought. The framework provides a multi-faceted signature of this crisis: a collapse of the active core (Panel a), preceded by the most significant structural fragmentation event in the time-series (Panel c), which resulted in a small, hyper-connected remnant network (Panel b).

To gain insight into the network's structural reorganisation dynamics, we computed the rate of geometric change between consecutive months, measured as the Frobenius distance $||\chi(p_t) - I||_F$ between the transformation matrix and the identity matrix (Figure 9). This metric quantifies the magnitude of month-to-month instability in the network's spectral geometry (Figure 9a). While elevated rates of change occurred during the 2011 drought period (marked by the blue dashed line), the absolute peak occurred later, in August 2012 (marked by the red dashed line), suggesting a significant lagged reorganisation or response to other factors not captured by simple connectivity metrics. Decomposing this change by spectral frequency (Figure 9b), using 3-month running averages to visualise trends, reveals that high-frequency modes (representing finer structural details, orange lines) exhibit greater month-to-month volatility but fluctuate around a relatively stable mean. In contrast, low-frequency modes (representing core structure, green lines) undergo slower but wider swings that appear to drive the major peaks in overall geometric change. A comparison of the $\chi(p)$ matrices during the 2011 drought collapse (Figure 9c) and the 2012 peak change period (Figure 9d) confirms both were major, highly off-diagonal reorganisations, albeit with different geometric signatures. The

observed month-to-month fluctuations throughout the time series (dashed/dotted lines in Fig 9a, b) also highlight the inherent variability and potential noise present in real-world ecological data.

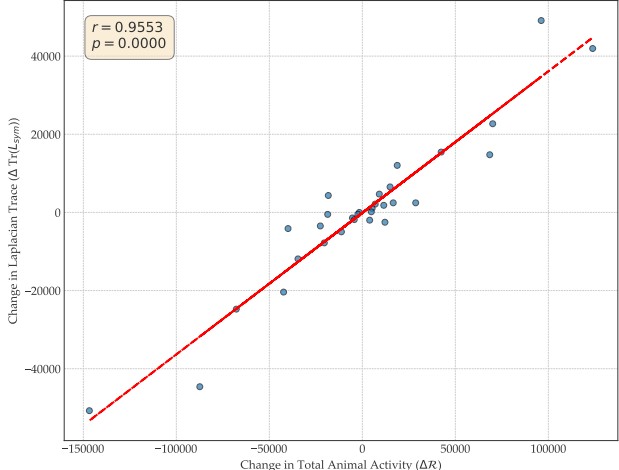

Figure 11: **Real-world Validation of the Spectral Trace Conservation Law (Theorem 3) on the Serengeti Food Web (2010-2013).** This figure provides a direct, quantitative test of the predicted relationship between physical resource dynamics and their spectral counterparts. Each point represents a month-to-month transition in the Serengeti ecosystem. The x-axis shows the change in the total observed animal count, our proxy for the total system resource ($\Delta\mathcal{R}$). The y-axis shows the corresponding change in the trace of the symmetrised Laplacian ($\Delta\mathrm{Tr}(L_{sym})$). The data reveals a strong, positive, and linear correlation between the physical and spectral quantities, as predicted by our framework. This relationship is quantified by a Pearson correlation coefficient of $r = 0.9553$ ($p < 0.0001$), providing compelling, real-world evidence that the physical conservation of resources within the system is directly reflected by this spectral invariant.

**Verifying Physical Conservation Laws in the Spectral Domain.**   To validate the framework's physical grounding, we tested the Spectral Trace Conservation Law. Figure 11 plots the change in total observed animal activity (our proxy for the system's total resource, $\Delta\mathcal{R}$) against the change in the trace of the symmetrised Laplacian ($\Delta\mathrm{Tr}(L_{sym})$) for each month-to-month transition. As predicted by Theorem 3, the two quantities exhibit a near-perfect linear relationship ($r = 0.9553, p < 0.0001$). It is important to note that this strong correlation is *not* a noisy, empirical biological finding, but rather a direct validation of the mathematical consistency of our framework. The result confirms that the abstract spectral quantity we defined (the Laplacian trace) behaves exactly as predicted by the physical quantity it is designed to represent (the total system resource, here modelled by camera-recorded animal activity). This grounds our functorial framework in the physical reality of the system.

**The Geometric Signature of a Crisis.**   We then leveraged the Process Interpretation Toolkit to move beyond detecting change to characterising its fundamental nature. Figure 12 compares the geometric signature ($\chi(p)$ matrix) of the 2011 drought collapse to that of a typical seasonal transition. The results are visually striking. A typical seasonal change (Panel c) is a minor perturbation, with a signature close to the identity matrix, although still significantly off-diagonal (Diagonal Dominance Ratio, Eq. 15, is 0.2884), and its core community structure is preserved (Panel f); this preservation is quantified by the very low Shannon entropy (0.0353 bits) calculated from the squared components shown. In contrast, the drought collapse (Panel a) has a complex, highly off-diagonal signature(Diagonal Dominance Ratio = 0.1890, even lower during recovery, Panel b, Ratio = 0.1566), corresponding to a topologically complex shattering of the Fiedler vector (Panel d), quantified by a significantly higher entropy (0.9582 bits, increasing further during recovery to 2.1167 bits, Panel e). The species-level analysis (Row 3) provides a concrete ecological interpretation. The pre-drought Fiedler vector (Panel g) is defined by the classic partition between migratory herds (e.g., *Connochaetes taurinus*) and their resident predators (e.g., *Crocuta crocuta*). The low Participation Ratios

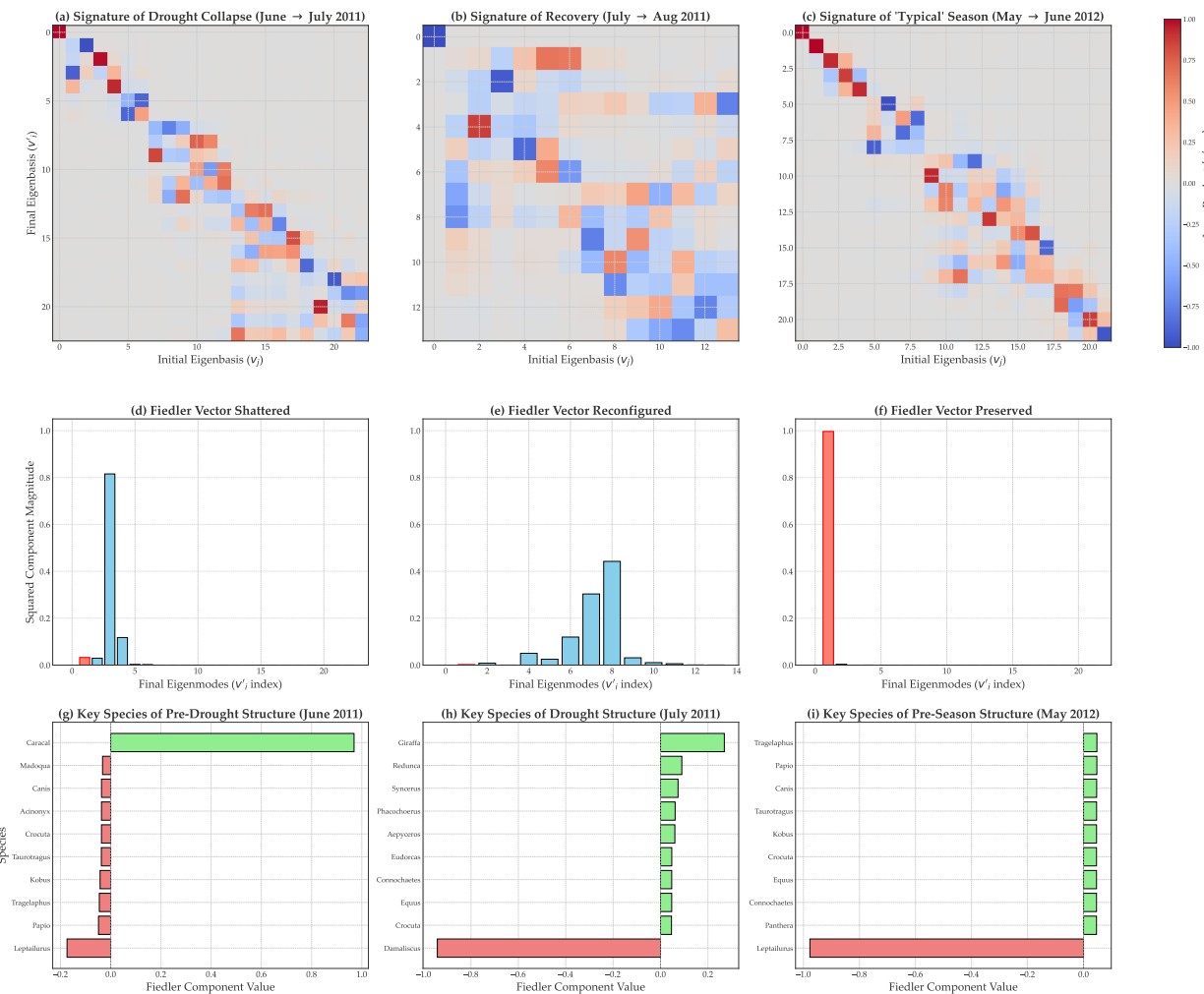

Figure 12: **The Geometric-Spectral Signature and Species-Level Impact of Ecological Processes of the Serengeti Food Web (2010-2013).** This figure provides a multi-layered validation of our framework's diagnostic power by comparing the geometric signature ($\chi(p)$ matrix) and structural impact of three distinct ecological processes. **Top Row (Panels a-c):** The heatmaps show the overall geometric signature of each process. A "typical" seasonal transition from wet to dry in 2012 (**c**) has a signature that is relatively more diagonal (Diagonal Dominance Ratio = 0.2884) compared to the highly off-diagonal signatures of the 2011 drought collapse (**a**) (Ratio = 0.1890) and subsequent recovery (**b**) (Ratio = 0.1566), quantitatively confirming the latter two as catastrophic structural reorganisations. **Middle Row (Panels d-f):** These panels reveal the mechanism behind the signatures by showing the transformation of the Fiedler vector ($v_2$), which represents the ecosystem's primary community structure. During the typical transition (**f**), the Fiedler vector is quantitatively preserved, mapping almost entirely onto the new Fiedler vector (Shannon Entropy of squared components = 0.0353 bits). During the drought collapse (**d**), this vector is quantitatively shattered, its energy scattered across many new structural modes (Entropy = 0.9582 bits). The recovery phase (**e**) shows the highest scattering (Entropy = 2.1167 bits). **Bottom Row (Panels g-i):** This provides a concrete, species-level interpretation. The low Participation Ratios of the initial Fiedler vectors ($\mathrm{PR}_g$=1.1319, $\mathrm{PR}_h$=1.2603, $\mathrm{PR}_i$=1.0974) indicate these structural modes were highly localised. The Fiedler vector of the pre-drought network (**g**) and the typical network (**i**) is defined by the classic ecological partition of the Serengeti: the large migratory herds (e.g., *Connochaetes taurinus*, *Equus quagga*) on one side, and their primary resident predators (e.g., *Crocuta crocuta*, *Panthera leo*) on the other. The shattering of this vector during the drought is the geometric signature of a well-documented ecological phenomenon: severe droughts force migratory herds to break their normal patterns in search of scarce resources, thus temporarily destroying the predictable spatial predator-prey dynamics that define the ecosystem's structure (Sinclair et al., 2007). The framework has detected and characterised this real-world crisis at the species level.

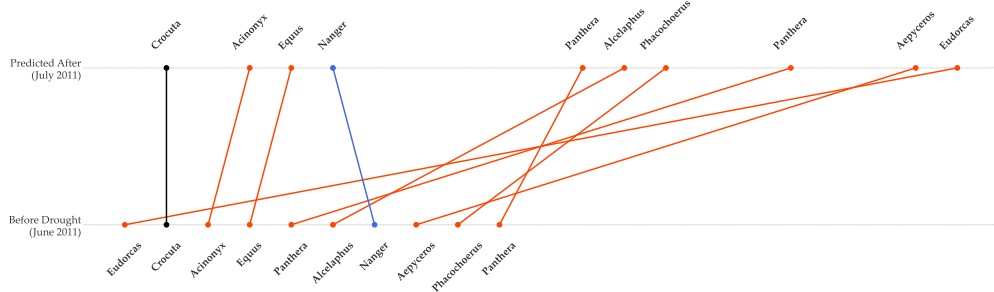

Figure 13: **A Falsifiable Prediction of Ecological "Winners" and "Losers" of the 2011 Drought using Signal Transport (Theorem 9).** This figure demonstrates the framework's predictive power. The signal is a species' importance, defined by its eigenvector centrality in the stable, pre-drought network of June 2011. This signal is then projected through the drought process using the transport matrix $T_{transport} = V'V^T$ to predict the new centrality of each species in the drought-stricken state of July 2011. The slope chart visualises the predicted change in rank for the top 10 most central species before the drought versus after applying the Signal Transport Theorem. The prediction indicates a major disruption to the species importance hierarchy, quantified by a large Mean Absolute Rank Change of 6.1 positions. Consistent with a significant reshuffling, the correlation between the initial and predicted ranks is weak and not statistically significant (Spearman's $\rho = 0.20$, $p = 0.58$). Despite the rank disruption, the framework predicts key ecological shifts reported in the literature (Sinclair et al., 2007): large grazers, such as Thomson's Gazelle (*Eudorcas*) and Hartebeest (*Alcelaphus*), and their specialist apex predator, the Lion (*Panthera*), are correctly identified as primary "losers", suffering a significant drop in relative importance. Conversely, adaptable, opportunistic carnivores like the Spotted Hyena (*Crocuta*) are predicted to be primary "winners", maintaining their central role. These results validate that the Signal Transport Theorem can quantitatively anticipate the significant reorganisation of ecological roles under major perturbations such as drought.

of these initial Fiedler vectors ($PR_g$=1.1319, $PR_h$=1.2603, $PR_i$=1.0974) confirm that this structure was highly localised, dominated by relatively few key species on either side of the partition. The shattering of this vector is the geometric signature of a well-documented ecological phenomenon: severe droughts force migratory herds to break their normal patterns, temporarily destroying the predictable spatial predator-prey dynamics that define the ecosystem's structure (Sinclair et al., 2007).

**Predicting Ecological Winners and Losers.** Our final experiment demonstrates the framework's predictive power. We defined a signal of species' "importance" (eigenvector centrality) on the stable, pre-drought network. We then used the transport matrix ($T_{transport}$) derived from the drought process to generate a falsifiable prediction for the new importance of each species in the drought-stricken state. Figure 13 visualises these predictions for the top 10 species initially. The transport predicts a major disruption to the species importance hierarchy, quantified by a large Mean Absolute Rank Change of 6.1 positions among the top 10 species shown. Consistent with this significant reshuffling, the correlation between the initial and predicted ranks is weak and not statistically significant (Spearman's $\rho = 0.20$, $p = 0.58$). The framework predicts that large grazers (e.g., *Eudorcas thomsonii*) and their specialist predators (*Panthera leo*) will be the primary "losers", while more adaptable, opportunistic carnivores (e.g., *Crocuta crocuta*) will maintain their central role. These predictions are supported by the ecological literature. Severe droughts are known to disrupt the primary food source for large grazers and, consequently, the specialist predators that depend on them. For instance, drought conditions in the Serengeti are known to fundamentally alter the dynamics of the wildebeest-predator system, affecting migration, foraging, and predation patterns (Sinclair et al., 2007). Despite the low overall rank correlation, the framework correctly identifies key ecological shifts reported in the literature: large grazers and their specialist predators are predicted "losers", while adaptable carnivores like hyenas are predicted "winners" (Sinclair et al., 2007). These results validate that the Signal Transport Theorem can quantitatively anticipate the significant reorganisation of ecological roles under major perturbations, correctly identifying key functional shifts even amidst substantial rank instability.

# 7 Real-World Case Study 2: Detecting Cognitive Processes in Brain Connectomics

To demonstrate the versatility and robustness of the `Proc-to-Spec` framework, we now apply it to a second case study from a fundamentally different scientific domain. This new case study, drawn from neuroscience, is designed to be a direct methodological contrast to the Serengeti analysis. Where the Serengeti study (§6) tested our framework's ability to *diagnose and interpret* a **high-signal, macro-timescale, external** shock (a drought) in a sparse, physically-grounded network, this brain network study tests its *sensitivity* to **low-signal, micro-timescale, internal** events (cognitive state shifts) in a dense, abstract network.

The scientific challenge here is distinct and, in many ways, more demanding. In the ecological case, the process of change was a massive, system-wide perturbation with a clear signal. In this neuroscience case, the process is a fleeting, internal cognitive shift, and its signature is notoriously subtle, buried within the high-dimensional, high-noise environment of fMRI (functional Magnetic Resonance Imaging) data. The core question thus shifts from one of interpretation (characterising a known shock) to one of detection: is the `Proc-to-Spec` framework sensitive enough to identify and quantify the unique geometric signature of a rapid, cognitive process?

## 7.1 HCP Data Processing and Dynamic Network Construction

The HCP case study requires a processing pipeline that is substantially different from the Serengeti's, one adapted for the high-speed, high-noise, and abstract nature of fMRI data. Our pipeline, summarised in Table 4, is designed to translate this data into the language of our `Proc-to-Spec` framework. We selected 21 subjects from the HCP 100 Unrelated release (see Table 3) for this analysis.

Table 3: **HCP-Young Adult 2025 Subject Cohort (n=21) used in this study.** All subjects were selected from the latest Human Connectome Project (HCP) S1200 release (updated in August 2025, datasets available for download at `https://balsa.wustl.edu/project?project=HCP_YA`), which provides extensive neuroimaging and behavioural data from healthy young adults (Van Essen et al., 2012). This table details the demographic and data completion status for each subject. **Release** (e.g., S500, S900) indicates the data release cohort, and **Acquisition** (e.g., Q07) denotes the acquisition quarter. **3T_Full_MR_Compl** confirms completion of the full 3T MRI protocol, which includes the resting-state (rfMRI) and working-memory task (tfMRI_WM) data used in our analysis. The **7T_Full_MR_Compl** and **MEG_FullProt_Compl** columns are included for reference and indicate completion of other imaging protocols not used in this specific study. We specifically selected those subjects for whom the full 3T, 7T, and MEG (Magnetoencephalography) imaging protocols were completed.

| Subject ID | Release | Acquisition | Gender | Age | 3T_Full_MR_Compl | 7T_Full_MR_Compl | MEG_FullProt_Compl |
|---|---|---|---|---|---|---|---|
| 105923 | MEG2 | Q07 | F | 31-35 | true | true | true |
| 108323 | S500 | Q04 | F | 26-30 | true | true | true |
| 109123 | S500 | Q06 | M | 31-35 | true | true | true |
| 116726 | S900 | Q08 | M | 26-30 | true | true | true |
| 140117 | S500 | Q04 | F | 26-30 | true | true | true |
| 156334 | MEG2 | Q03 | F | 26-30 | true | true | true |
| 162935 | S900 | Q09 | M | 22-25 | true | true | true |
| 164636 | S900 | Q09 | M | 22-25 | true | true | true |
| 175237 | S900 | Q10 | F | 31-35 | true | true | true |
| 185442 | S500 | Q07 | M | 22-25 | true | true | true |
| 191033 | S500 | Q06 | F | 26-30 | true | true | true |
| 192641 | MEG2 | Q07 | F | 31-35 | true | true | true |
| 198653 | S900 | Q10 | M | 22-25 | true | true | true |
| 204521 | S500 | Q07 | F | 31-35 | true | true | true |
| 257845 | S900 | Q09 | M | 26-30 | true | true | true |
| 283543 | S900 | Q09 | M | 22-25 | true | true | true |
| 406836 | S900 | Q10 | F | 31-35 | true | true | true |
| 581450 | S900 | Q09 | M | 22-25 | true | true | true |
| 680957 | Q3 | Q03 | F | 26-30 | true | true | true |
| 725751 | S900 | Q05 | M | 26-30 | true | true | true |
| 898176 | S500 | Q06 | M | 31-35 | true | true | true |

Table 4: **HCP Case Study: Data and Processing Parameters.** This table details the full processing pipeline and parameters used to construct the dynamic Resource Connectome (dRC) time-series from the raw HCP fMRI data. The *Dataset* parameters define the source data from the HCP S1200 release (Van Essen et al., 2012), the 379-node Glasser parcellation (Glasser et al., 2016), and the canonical Cole-Anticevic mapping to functional networks (Ji et al., 2019). The *Temporal Parameters* are critical: we use a 30-second (42 TR) window to ensure temporal precision for brief cognitive events, and a 50% (21 TR) step size to ensure consecutive windows are sufficiently distinct, which is essential for capturing the process of change. Finally, the *Proc-to-Spec Model* parameters define our novel, biologically-grounded network representation: Node Resource is the BOLD signal variance (a proxy for local metabolic activity (Garrett et al., 2011; Grady & Garrett, 2014)), and the Edge Weight is defined by our gated mass-action model, $W = \max(0, r) \cdot \mathrm{var}(u) \cdot \mathrm{var}(v)$, which directly parallels the ecological model in §6.

| Parameter | Value |
|---|---|
| *Dataset* | |
| Source | Human Connectome Project (HCP) S1200 (Van Essen et al., 2012) at `https://balsa.wustl.edu/project?project=HCP_YA` |
| Subjects | 21 (See Table 3) |
| Runs | All 4 resting-state (rfMRI) & 7 tasks (tfMRI: WM, MOTOR, LANGUAGE, EMOTION, GAMBLING, RELATIONAL, SOCIAL) |
| Atlas | Glasser et al. (2016) Multi-Modal Parcellation (Glasser et al., 2016) |
| Network Mapping | Cole-Anticevic Brain-wide Network Partition (Ji et al., 2019) at `https://github.com/ColeLab/ColeAnticevicNetPartition` |
| Nodes (Regions) | 379 (360 Cortical + 19 Subcortical) |
| *Temporal Parameters* | |
| fMRI Signal | BOLD (Blood Oxygenation Level Dependent) |
| Repetition Time (TR) | 0.720 seconds |
| Window Duration | 30 seconds (42 TRs) |
| Window Step Size | 21 TRs (∼15.1s, 50% Overlap) |
| *Proc-to-Spec Model* | |
| Network Type | Dynamic Resource Connectome (dRC) |
| Node Resource Metric | Signal Variance: $\mathrm{var}(\mathrm{BOLD}_t)$, a proxy for local metabolic activity (Garrett et al., 2011; Grady & Garrett, 2014) |
| Edge Weight Metric | Gated Mass-Action: $W = \max(0, r) \cdot \mathrm{var}(u) \cdot \mathrm{var}(v)$ |

**Step 1: BOLD Time-Series Extraction and Definition of Network Nodes.** Our analysis begins with the publicly available Human Connectome Project (HCP) S1200 release (Van Essen et al., 2012), from which we selected 21 subjects (listed in Table 3). The HCP provides extensive neuroimaging data for each subject, acquired using functional Magnetic Resonance Imaging (fMRI). This technology does not measure neural firing directly; instead, it measures the Blood Oxygenation Level Dependent (BOLD) signal (an example of which is shown in Figure 14b). The BOLD signal is a complex, indirect proxy for neural activity, reflecting the localised *changes* in blood flow and oxygenation that occur when a brain region becomes metabolically active (He, 2013).

This raw BOLD signal is exceptionally noisy, as it is contaminated by non-neural artifacts from subject head motion, breathing, and heartbeat. To address this, we used the HCP's pre-processed **'ICA-FIX' denoised** dataset. This dataset has been rigorously cleaned using Independent Component Analysis (ICA) to identify and regress out these known noise components, providing a much cleaner signal that is more representative of underlying neural dynamics.

To model the brain as a network, we must first parcellate it into a set of discrete nodes. We employed the Glasser et al. (2016) Multi-Modal Parcellation (Glasser et al., 2016), a state-of-the-art atlas that defines 379 distinct brain regions (360 cortical, 19 subcortical) based on their architecture, function, and connectivity (visualised in Figure 14a).

For each subject, we analysed all available fMRI scans, which fall into two categories:

- **Resting-state fMRI (rfMRI):** Four separate runs (two with 'LR' and two with 'RL' phase-encoding) where subjects simply rest. These provide a baseline of the brain's "idle" or spontaneous activity.

- **Task fMRI (tfMRI):** All seven distinct cognitive tasks, each with two runs (LR and RL encoding), for a total of 14 task runs. These tasks (Working Memory, Motor, Language, etc., as listed in Table 4) present specific cognitive challenges that actively reorganise the brain's functional networks.

The average BOLD time-series was extracted from each of the 379 Glasser regions for all of these fMRI runs. The final output of this step is a (timepoints × 379 regions) data matrix for each subject and run, which serves as the raw input for constructing our dynamic resource connectomes.

**Step 2: Defining the `Proc` Object (The dRC Matrix).** A significant challenge in applying our framework is that brain networks are not directly observed resource flows, but are instead abstract correlation matrices. Standard functional connectivity (FC) often uses the Pearson correlation coefficient $r$, which includes negative values (anti-correlations) and is bounded, making it incompatible with our non-negative, resource-constrained `Proc` category. To bridge this gap, we defined a new metric, the dynamic Resource Connectome (dRC), that is theoretically consistent with our framework and parallels the mass-action model from the Serengeti study. This construction involves two steps:

First, we define a node's resource, $\mathcal{R}(u, t)$. The raw BOLD signal's mean is an arbitrary scanner unit, but its **variance** (or standard deviation) within a time window $t$ is a well-established, meaningful signal. The literature has long demonstrated that BOLD signal variability is not mere "noise" but is a powerful, non-negative proxy for local metabolic "power", correlating with age, cognitive performance, and functional integrity (Garrett et al., 2011; Grady & Garrett, 2014). We thus define our node resource as:

$$\mathcal{R}(u, t) = \text{var}(\text{BOLD}_{u,t}) \tag{17}$$

This provides a non-negative scalar value for each node's metabolic resource consumption at time $t$.

Second, we define the edge weight $W(u, v)$ using a **gated mass-action model**:

$$W_t(u, v) = \max(0, r_{uv}) \cdot \mathcal{R}(u, t) \cdot \mathcal{R}(v, t) \tag{18}$$

This model is a direct analogue to the Serengeti's $C \cdot \text{count}(u) \cdot \text{count}(v)$. The model has two components:

- **The Cooperation Gate (**$\max(0, r_{uv})$**):** The term $r_{uv}$ is the Pearson correlation between regions $u$ and $v$ within the window $t$. The $\max(0, r)$ function acts as a gate, filtering for only positive, in-phase synchrony. This creates a graph that explicitly models the brain's "cooperative" resource network, while treating anti-correlations (a mechanistically different process) as a "no connection" state.

- **The Mass-Action Component (**$\mathcal{R}(u, t) \cdot \mathcal{R}(v, t)$**):** This term models the interaction strength as the product of the resources each node brings to the connection. A strong interaction requires *both* high synchrony (high $r$) and high metabolic activity (high variance) from *both* participating nodes.

This results in a non-negative, symmetric matrix $W_t$ that is a valid object in our `Proc` category. This variance-based metric is also sensitive to complex neural phenomena, such as the task-induced *reduction* in variability reported in the literature (He, 2013), making it a powerful tool for capturing non-obvious network reorganisations.

**Step 3: Defining the `Proc` Morphism (The Process $p_t$).** We create a time-series of these $G_t = (V, W_t)$ matrices to model the brain's dynamic processes, $p_t : G_t \to G_{t+1}$. The choice of sliding window parameters is a critical and non-trivial methodological step in all dynamic connectivity studies. The parameters must be carefully chosen to balance two competing factors: *statistical stability* (which favours longer windows to get

a reliable matrix) and *temporal precision* (which favours shorter windows to detect brief cognitive events) (Leonardi & Van De Ville, 2015; Zalesky & Breakspear, 2015). For this study, we selected a **30-second (42-TR) window**. This is a principled choice, as the cognitive blocks in the HCP task-fMRI (tfMRI) runs are themselves brief (e.g., the Working Memory task blocks are 25 seconds). A 30-second window is therefore short enough to achieve high temporal precision, creating $W_t$ matrices that are more likely to represent a single, "pure" cognitive state (either 'Rest' or 'Task') rather than averaging or "smearing" them together, which would be an unavoidable artifact of a longer (e.g., 60s or 90s) window.

Crucially, we used a **50% overlap (21-TR step size)**. This ~15s step is a critical methodological choice. Our initial experiments using a 1-TR (0.72s) step, which creates a 99% overlap between consecutive windows, failed to find a signal. This is because the extreme overlap mathematically "averaged out" the signal of change, as $G_t$ and $G_{t+1}$ were nearly identical. The 50% overlap is a standard signal-processing technique that ensures consecutive networks are substantially distinct, allowing our framework to robustly detect the process of change between them. The resulting `Proc` objects ($G_{rest}$, $G_{task}$) and the morphism ($G_{task} - G_{rest}$) are visualised in Figure 15 (a-c).

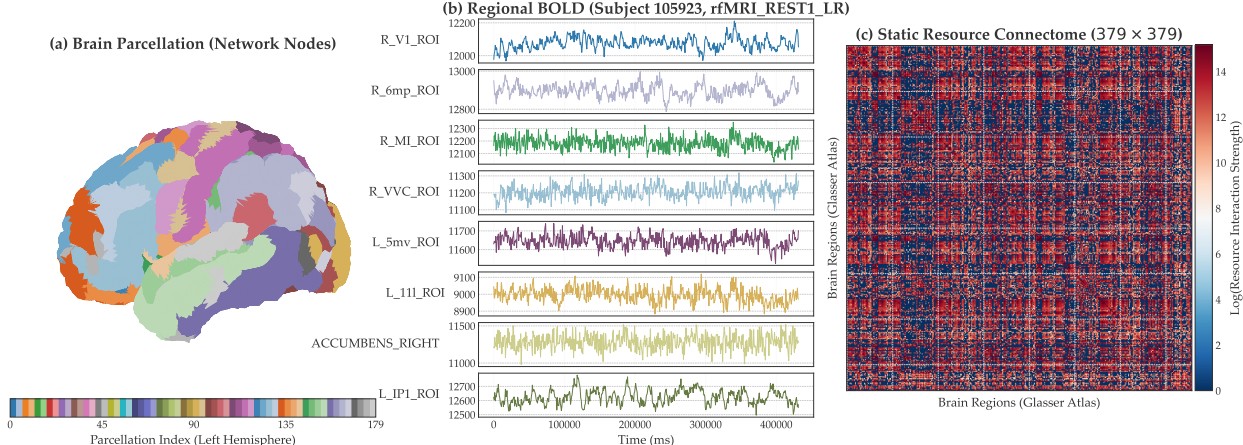

Figure 14: **Overview of the Human Connectome Project (HCP) Case Study Data (Subject ID: 105923).** This figure illustrates the key components of our brain network analysis. **(a) Brain Parcellation (Network Nodes):** The 379 regions of the Glasser atlas (360 cortical + 19 subcortical) plotted on a left-hemisphere 3D cortical surface. These regions serve as the nodes of our network topology. **(b) Regional BOLD Time Series (Signal):** Example raw time series from 8 selected regions in the resting-state run (rfMRI_REST1_LR; TR = 720 ms). BOLD (Blood Oxygenation Level Dependent) signal reflects neurovascular coupling—changes in blood oxygenation and flow that indirectly track local neural activity. **(c) Static Resource Connectome (Full** $379 \times 379$**):** This heatmap visualises the full static network $W$, computed using our theoretically-grounded resource model. The weight of an edge $(u, v)$ is defined as $W(u, v) = \max(0, r_{uv}) \cdot \text{var}(u) \cdot \text{var}(v)$, where $\text{var}(u)$ is the variance (activity) of region $u$ and $r_{uv}$ is the Pearson correlation. This gated mass-action model is a direct analogue to the ecological model used in §6 and ensures all edge weights are non-negative. The matrix is log-transformed ($\log(1 + W)$) for visualisation, which compresses the extreme values to reveal the brain's rich community structure (bright diagonal blocks) and inter-community connectivity (off-diagonal patterns).

**Step 4: Validating the `Proc` Model.** Before applying our spectral functor to the full dataset, we first validated that our dRC model (our `Proc` objects) is biologically meaningful. We performed an in-depth analysis of the Working Memory (WM) task data for a representative subject (105923), as shown in Figure 15. To create stable representations of the `Proc` objects for this subject, we first averaged all 30-second $W_t$ matrices (from Step 3) that were identified as 'Rest' periods to create a single, stable baseline connectome, $G_{rest} = (V, W_{rest})$. This is visualised in Figure 15a. We did the same for all 'Task' windows to create $G_{task} = (V, W_{task})$ (Figure 15b). These heatmaps, which are log-transformed for visual clarity,

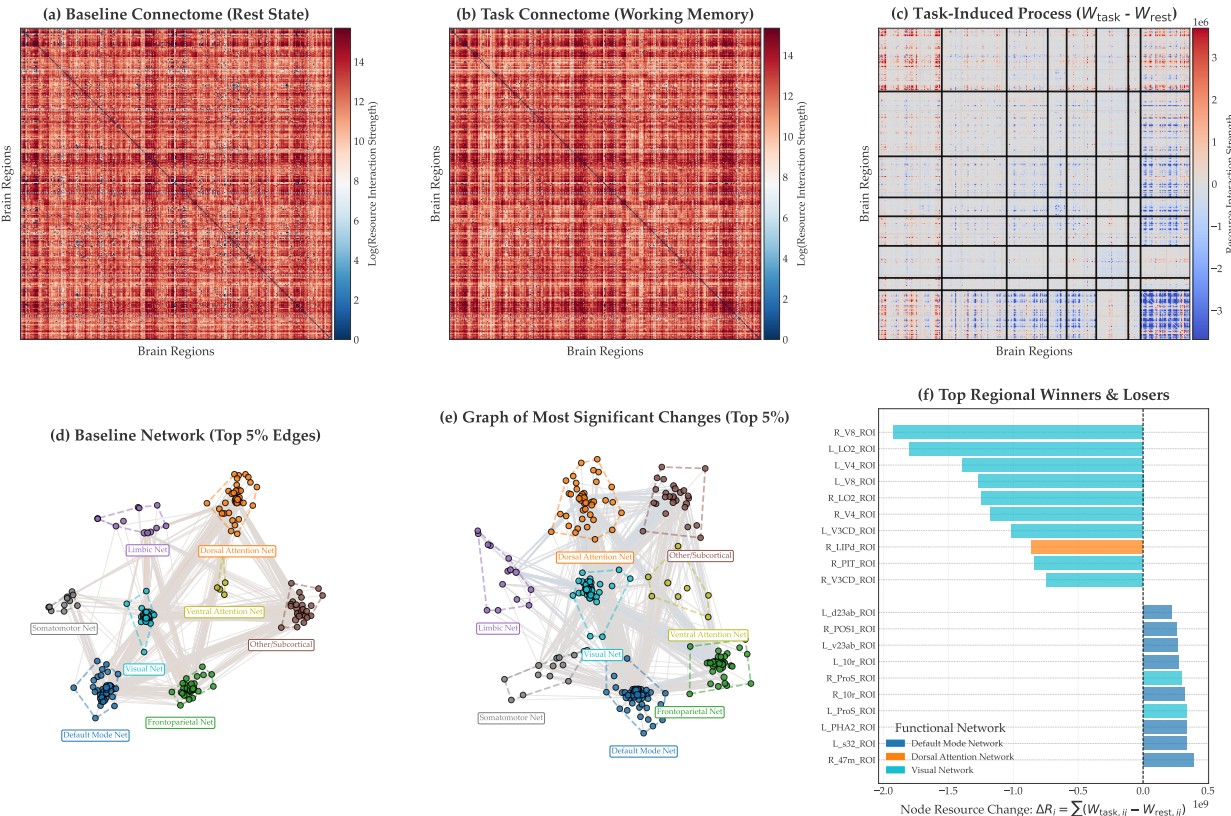

Figure 15: **Validation of the `Proc` Category Representation in a Brain Network (Subject ID: 105923).** This figure demonstrates that our resource-based model for objects and morphisms in `Proc` is robust and biologically meaningful in a complex neuroscience case study. **(a), (b)** The *objects* in `Proc`, $G_{rest}$ and $G_{task}$, are visualised as $379 \times 379$ resource connectomes. The edge weights are defined by our gated mass-action model, $W(u,v) = \max(0, r_{uv}) \cdot \text{var}(u) \cdot \text{var}(v)$, which parallels the ecological model in §6. The matrices are log-transformed for visualisation, revealing the brain's clear modular structure. **(c), (e)** The *morphism* (process) $p : G_{rest} \to G_{task}$ is visualised as the raw difference matrix $W_{task} - W_{rest}$. This signature shows the complex geometric reorganisation of resource flow, with red connections being formed/strengthened and blue ones dissolved/weakened. **(d)** A 3D graph visualisation of the baseline network's core connections. **(f)** Critical validation of our `Proc` model. This bar chart plots the total change in resource output for each node ($\Delta R_i = \sum_j (W_{task,ij} - W_{rest,ij})$). This reveals a novel insight distinct from standard activation maps: the 'Winners' (largest positive $\Delta R_i$) are predominantly regions of the **Default Mode Network (DMN)**, while the 'Losers' (largest negative $\Delta R_i$) are key regions of task-positive networks like the **Visual Network** and **Dorsal Attention Network**. This suggests that our metric captures a complex reorganisation where task networks 'lock in' (reducing variance and thus $W$), while the DMN deactivates into a higher-variance 'idling' state.

confirm the model from Figure 14c and show the clear, modular structure of the brain's resource connectome in both states.

The *morphism* (process) $p : G_{rest} \rightarrow G_{task}$ is then defined by the raw difference matrix, $W_{diff} = W_{task} - W_{rest}$, visualised in Figure 15c. This matrix is the geometric signature of the cognitive process. Figure 15d and 15e provide 3D spatial intuition for this, showing the core baseline network and the sparse "scaffolding" of change, respectively. The critical validation is shown in Figure 15f. Here, we computed the change in total resource output for each node $i$, $\Delta R_i = \sum_j (W_{task,ij} - W_{rest,ij})$. This analysis reveals a novel and counter-intuitive insight that is central to our case study. The **'Winners'** (largest positive $\Delta R_i$, bars to the right) are predominantly regions of the **Default Mode Network (DMN)**. The **'Losers'** (largest negative $\Delta R_i$, bars to the left) are key regions of task-positive networks: the **Visual Network** and the **Dorsal Attention Network**.

This finding, which is the inverse of a standard activation map (where DMN deactivates and Visual/Attention activates), is mechanistically sound. It suggests that our variance-based resource metric ($W \propto \text{var}(u) \cdot \text{var}(v)$) is not capturing simple BOLD increases, but a deeper reorganisation. Specifically, task-positive networks 'lose' resource interaction strength as their nodes 'lock in' to the task, reducing their signal variance (becoming more efficient). Conversely, the DMN 'deactivates' by disengaging and entering a more chaotic, high-variance idling state, which our metric registers as an increase in total resource interaction (He, 2013; Garrett et al., 2011). Having thus validated that our `Proc` representation is not just an abstraction but is biologically plausible and capable of uncovering novel, non-obvious network dynamics, we now proceed to apply our `Proc-to-Spec` functor to analyse its spectral properties.

**Step 5: Constructing the `Spec` Object (The $\chi(p)$ Matrix).** With the time-series of $W_t$ matrices (our `Proc` objects) established, we apply the identical `Proc-to-Spec` functor $\chi$ as in the Serengeti study (§6). For each consecutive, 50%-overlapping pair $(G_t, G_{t+1})$, we:

1. Compute the Symmetrised Laplacian $L_{sym,t}$ from $W_t$ (as defined in Eq. 3).

2. Compute $L_{sym,t+1}$ from $W_{t+1}$.

3. Compute the change-of-basis transformation matrix $\chi(p_t) = V_{t+1}^T V_t$, where $V_t$ and $V_{t+1}$ are the orthonormal eigenvector matrices of their respective Laplacians (as defined in Lemma 2).

This results in a new time-series of $\chi(p_t)$ matrices, which represent the *spectral signatures* of the brain's cognitive processes. This time-series of spectral objects is the data we analyse in the following sections.

### 7.2 Experimental Results

Our analysis of the HCP dataset proceeds in a logical sequence. We first validate our `Proc` category representation (the dRC) by showing it captures a known, non-trivial neuroscientific phenomenon. We then apply our `Proc-to-Spec` functor to the $n \times n$ $\chi(p)$ matrices and analyse them with scalar metrics, revealing the first quantitative evidence of two distinct process families. We then apply our new `Spec-to-Func` projection (Theorem 10) to create $k \times k$ functional fingerprints, which we deconstruct using a hub analysis. Finally, we provide a rigorous machine learning validation, proving that these fingerprints are quantitatively separable, generalisable features for classifying cognitive states.

**A Quantitative Toolkit for $\chi(p)$ Matrices.** The $n \times n$ transformation matrices $\chi(p)$ (where $n = 379$) are intractably large for direct interpretation and comparison (see Figure 16a-c). To quantitatively analyse their geometric properties, we first define a toolkit of three scalar metrics, which are plotted in Figures 16 and 17. Let $A = |\chi(p)|$ be the matrix of absolute element values and $\|\chi(p)\|_1 = \sum_{i,j} A_{ij}$ be its $L_1$-norm. The metrics are computed as follows:

- **Diagonal Concentration (AUC):** This metric quantifies "structural inertia" (Theorem 7) by measuring how much of the transformation's energy is concentrated along the main diagonal. We

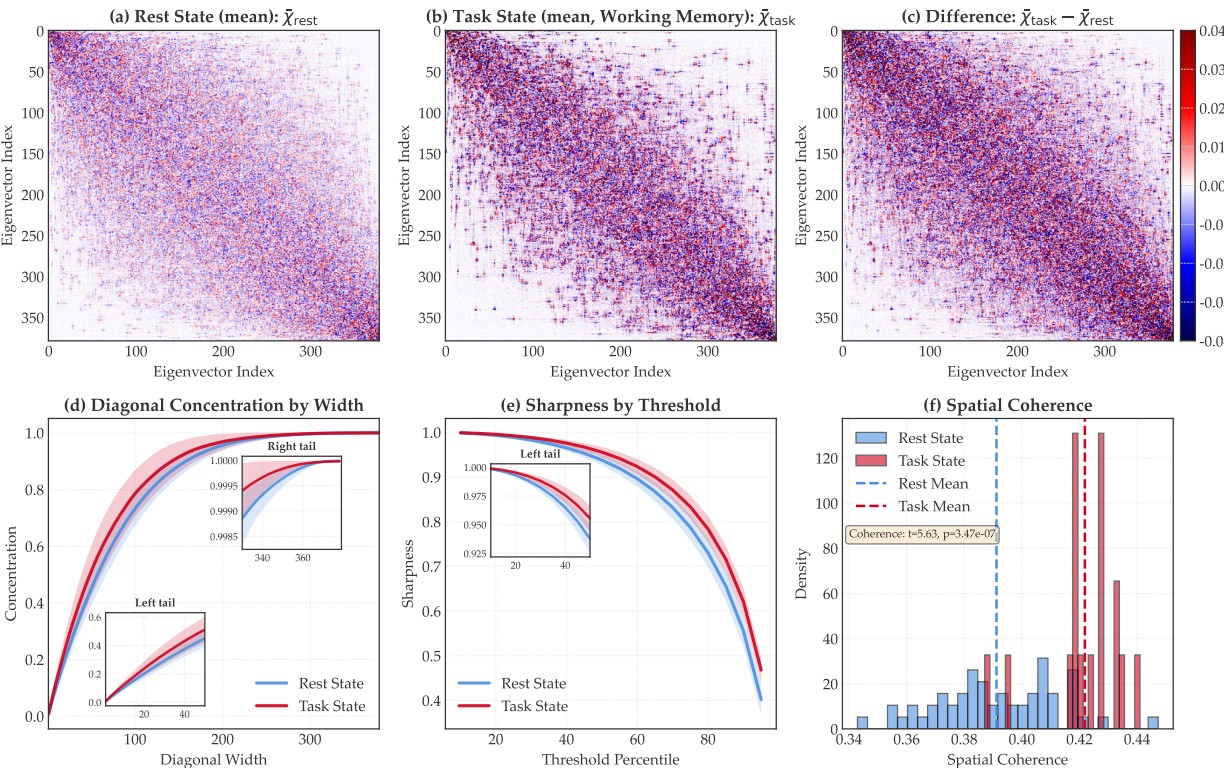

Figure 16: **Quantitative analysis of the "Null" vs. "Cognitive" process signatures (Subject ID: 105923).** This figure compares the geometric transformation $(\chi(p))$ matrices from "Null" processes (all `rest-to-rest` transitions) with "Cognitive" processes (all `rest-to-task` transitions) for the Working Memory task. The analysis reveals that the cognitive transformation is a "lock-in" to a more stable, structured geometric state. **(a)** The mean "Null" signature, $\bar{\chi}_{\text{rest}}$, representing baseline geometric noise. **(b)** The mean "Task" signature, $\bar{\chi}_{\text{task}}$, which is visibly more structured and diagonally-dominant. **(c)** The difference matrix, $\bar{\chi}_{\text{task}} - \bar{\chi}_{\text{rest}}$, which is not random noise. It highlights a strong, positive diagonal and structured off-diagonal changes, confirming the "lock-in" effect. **(d) Diagonal Concentration:** The Task transformation (red) has more of its energy consistently concentrated near the main diagonal than the Rest transformation (blue). **(e) Sharpness:** The Task transformation's energy is consistently concentrated in fewer, high-magnitude elements (higher sharpness) compared to the more diffuse, noisy Rest transformation. **(f) Spatial Coherence:** The Task transformations (red histogram) are significantly more structured and less random (higher local autocorrelation) than the Rest transformations (blue histogram). The difference is statistically irrefutable ($t = 5.63, p = 3.47 \times 10^{-7}$) for *subject 105923*.

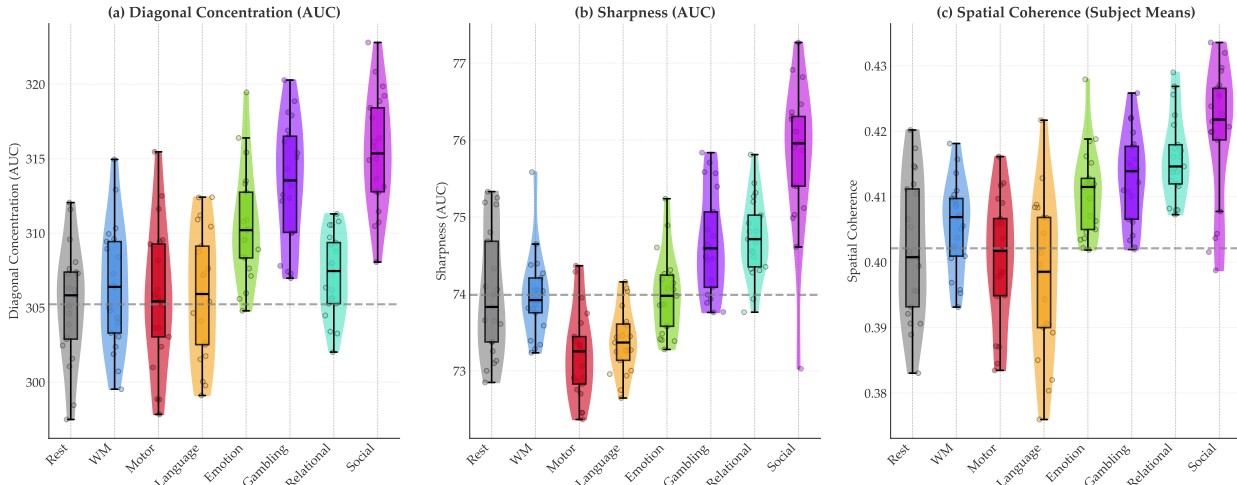

Figure 17: **Multi-task statistical validation of geometric process metrics.** This figure compares the "Null" process (`Rest-to-Rest`) against all seven "Cognitive" processes (`Rest-to-Task`) using subject-level means (N=21) for the three geometric metrics. The results show that the metrics successfully separate the tasks into distinct families. **(a) Diagonal Concentration (AUC):** Higher-order tasks like `Emotion`, `Gambling`, `Relational`, and `Social` show a highly significant increase in diagonal concentration compared to `Rest`. **(b) Sharpness (AUC):** This metric reveals a different pattern. `Motor` and `Language` tasks are characterised by a significant decrease in sharpness, while `Gambling`, `Relational`, and `Social` show a significant increase. **(c) Spatial Coherence:** This metric shows the clearest separation. The "Social/Executive" tasks (`Emotion`, `Gambling`, `Relational`, `Social`) are all significantly more coherent (more structured, less random) than the `Rest` state. Notably, the `Working Memory` (WM) task is statistically indistinguishable from `Rest` on all three metrics at the *group level* (all $p > 0.24$), suggesting that its geometric signature is highly variable across subjects. Collectively, this demonstrates that our framework provides a rich, multi-dimensional feature space capable of robustly distinguishing between different families of cognitive processes.

Table 5: **Statistical comparison of cognitive task signatures against the "Null" process.** This table provides the quantitative statistics for the distributions shown in Figure 17. Each row represents a paired t-test comparing the subject-level means of a `Rest-to-Task` process against the `Rest-to-Rest` (Null) process (N=21 subjects). P-values less than 0.05 are in **bold**. The results show a clear, non-uniform separation of tasks. "Social/Executive" tasks (`Emotion`, `Gambling`, `Relational`, `Social`) show a highly significant increase in **Coherence** and **Diagonal Concentration**. In contrast, `Motor` and `Language` tasks are uniquely characterised by a significant *decrease* in **Sharpness**. `Working Memory` (WM) shows no significant difference from `Rest` at the *group level* on any metric.

|            |    | Spatial Coherence | | Diagonal Concentration (AUC) | | Sharpness (AUC) | |
|------------|----|-------------|---------|-------------|---------|-------------|---------|
| **Task**   | **N** | **t-statistic** | **p-value** | **t-statistic** | **p-value** | **t-statistic** | **p-value** |
| WM         | 21 | 1.191  | 0.2477 | 1.142 | 0.2669 | -0.061 | 0.9520 |
| Motor      | 21 | -0.715 | 0.4829 | 0.482 | 0.6349 | -3.934 | **0.0008** |
| Language   | 21 | -1.406 | 0.1749 | 0.751 | 0.4614 | -3.095 | **0.0057** |
| Emotion    | 21 | 3.639  | **0.0016** | 6.192 | **<0.0001** | -0.010 | 0.9919 |
| Gambling   | 21 | 5.536  | **<0.0001** | 8.754 | **<0.0001** | 3.409 | **0.0028** |
| Relational | 21 | 4.717  | **0.0001** | 2.052 | 0.0535 | 3.452 | **0.0025** |
| Social     | 21 | 5.050  | **<0.0001** | 8.419 | **<0.0001** | 5.971 | **<0.0001** |

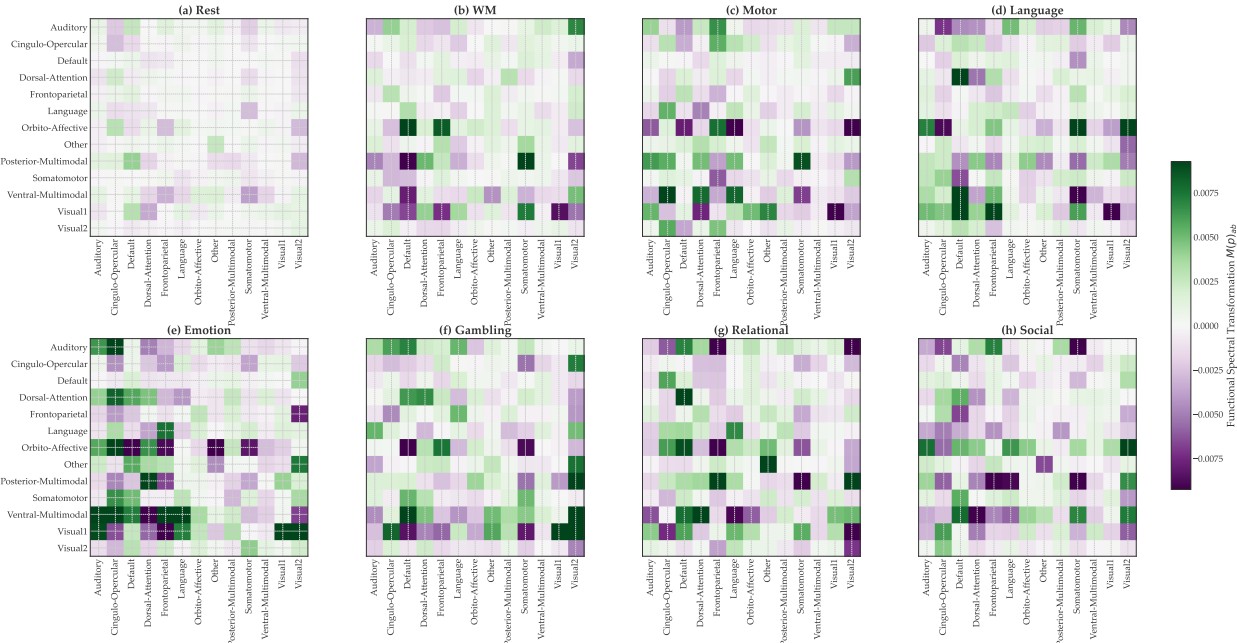

Figure 18: **Functional Signatures of Cognitive Processes Derived from the Spec-to-Func Projection (Theorem 10).** This figure displays the group-average (N=21) $k \times k$ functional transformation matrices, $\bar{M}(p)$, for the "Null" process (a) and seven distinct cognitive tasks (b-h). Each matrix is the optimal Galerkin projection of the full $n \times n$ spectral transformation $\chi(p)$ onto the $k = 13$ functional networks (rows/columns) of the Cole-Anticevic atlas (Ji et al., 2019). The entry $\bar{M}(p)_{ab}$ (y-axis: a, x-axis: b) quantifies the average spectral influence that original structural modes from functional network $b$ (x-axis) exert on the new structural modes of functional network $a$ (y-axis) during the process. Green indicates a positive transformation; purple indicates a negative (inverse) transformation. The results visually and quantitatively separate the cognitive processes into two distinct families, precisely explaining the statistical findings in Table 5. **(a-d) "Incoherent" Processes:** The **(a) Rest** process (a "Rest-to-Rest" transition) serves as the null model. It is quantitatively shown to be a zero-mean (matrix mean $\approx -0.000063$) and low-variance (std = 0.0011) process, confirming the baseline stability of the brain's spectral-functional geometry. The **(b) WM**, **(c) Motor**, and **(d) Language** tasks, while possessing higher variance (e.g., Motor std = 0.0035, 3.1x > Rest), are also zero-mean processes (e.g., WM mean $\approx 0.000018$). This lack of a consistent, structured signature (i.e., a non-zero mean) at the group level explains why they were statistically indistinguishable from Rest on coherence-based metrics in Table 5. **(e-h) "Coherent" Processes:** In stark contrast, the "Social/Executive" tasks – **(e) Emotion**, **(f) Gambling**, **(g) Relational**, and **(h) Social** – are high-variance, non-zero-mean processes (e.g., Emotion mean = 0.00073, std = 0.0052, 4.6x > Rest). They exhibit strong, structured, and unique off-diagonal "fingerprints", visually confirming their high statistical significance for metrics like Spatial Coherence (Table 5). These signatures offer mechanistic insights: **Gambling (f)** reveals a composite signature, capturing two parallel processes: (1) a strong negative (purple) transformation from the `Default` network to the `Orbito-Affective` network (value -0.0120), a known reward/valuation circuit, and (2) a strong positive (green) transformation from `Visual2` to the `Ventral-Multimodal` network (value 0.0118), representing the task-positive visual-to-executive pathway. **Emotion (e)** shows a different non-trivial signature, characterised by a massive positive transformation from the `Frontoparietal` network to the `Ventral-Multimodal` network (value 0.0238) and the strongest diagonal self-preservation of any task in the `Visual1` network (value 0.0212), indicating a profound spectral reorganisation and integration of executive and sensory networks.

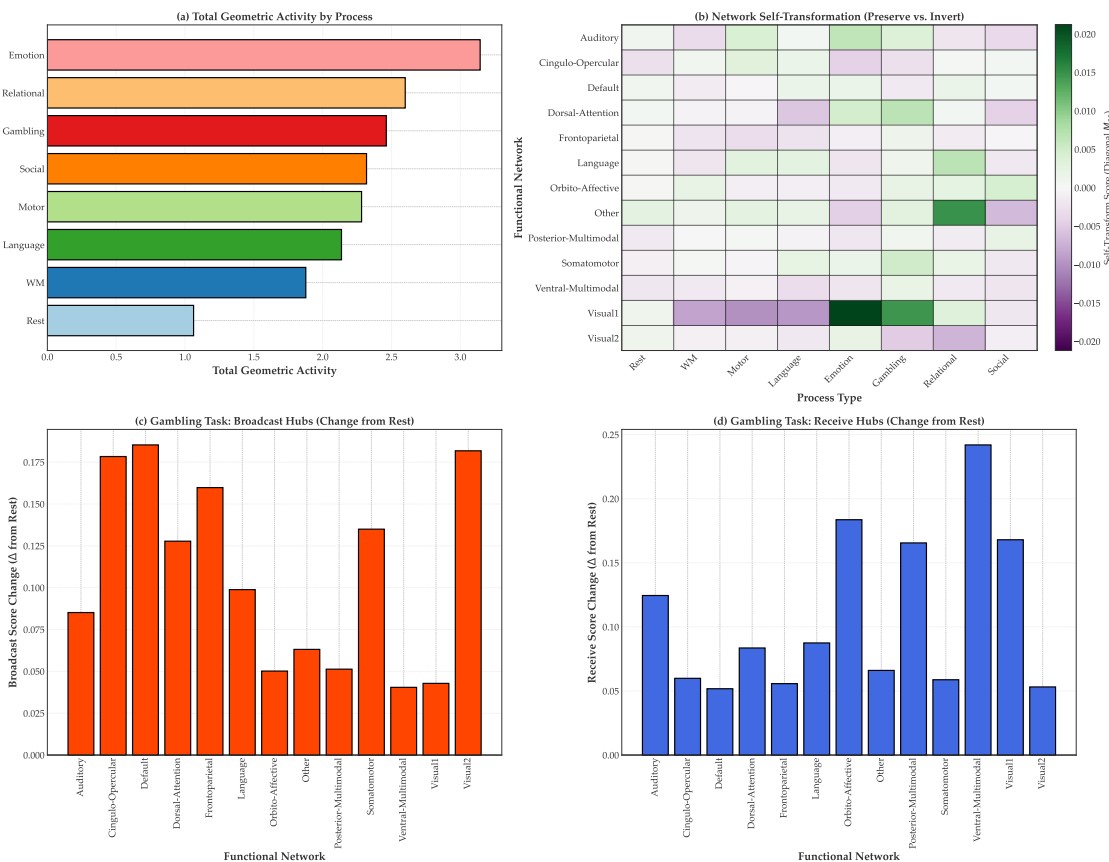

Figure 19: **Quantitative Analysis of Functional Hubs of Geometric Transformation (N=21 Subjects).** This figure provides a quantitative deconstruction of the $k \times k$ functional transformation matrices, $\bar{M}(p)$, from Figure 18. **(a) Total Geometric Activity by Process:** This panel plots the total off-diagonal power of each process, calculated as the sum of all absolute off-diagonal values of $\bar{M}(p)$ (i.e., the Total Broadcast Score). This quantifies the total magnitude of spectral-functional transformation. The plot reveals two distinct families: the stable Rest process (Total Activity $\approx$ 1.06), a group of "Incoherent" processes with low-to-moderate activity (e.g., WM: 1.88, Language: 2.14), and a group of "Coherent" processes with high activity (e.g., Gambling: 2.46, Emotion: 3.14). This provides a quantitative explanation for the statistical separation seen in Table 5. **(b) Network Self-Transformation (Preserve vs. Invert):** This heatmap plots the diagonal element $\bar{M}(p)_{aa}$ for each functional network (y-axis) across each process (x-axis). This reveals a "Preserve vs. Invert" signature. The Rest column is uniformly near-zero (mean $\approx$ 0.000048), providing a stable baseline. In contrast, visually-driven "Coherent" tasks like Emotion and Gambling show a strong positive (green, "preserve") value for the `Visual1` network (values $+0.02119$ and $+0.01450$, respectively). Conversely, non-visual "Incoherent" tasks like Motor and Language show a strong negative (purple, "invert") value for `Visual1` (values $-0.00996$ and $-0.00955$, respectively), suggesting a fundamental difference in how these task families modulate visual network geometry. **(c, d) Gambling Task Hub Analysis (Δ from Rest):** These panels quantify the specific flow of geometric transformation for the Gambling task by plotting the change in hub scores relative to the Rest baseline. **(c) Broadcast Hubs (Senders):** This plots the change in the Broadcast Score (sum of abs. column, excluding diagonal). It identifies the primary "senders" of the transformation, quantitatively confirming the Default ($\Delta = +0.185$), Visual2 ($\Delta = +0.182$), and Cingulo-Opercular ($\Delta = +0.178$) networks as the main drivers of geometric change. **(d) Receive Hubs (Receivers):** This plots the change in the Receive Score (sum of abs. row, excluding diagonal). It identifies the primary "receivers" of the transformation, with the largest changes in the Ventral-Multimodal ($\Delta = +0.242$), Orbito-Affective ($\Delta = +0.184$), and Visual1 ($\Delta = +0.168$) networks. Together, (c) and (d) provide a quantitative flow diagram of the Gambling process.

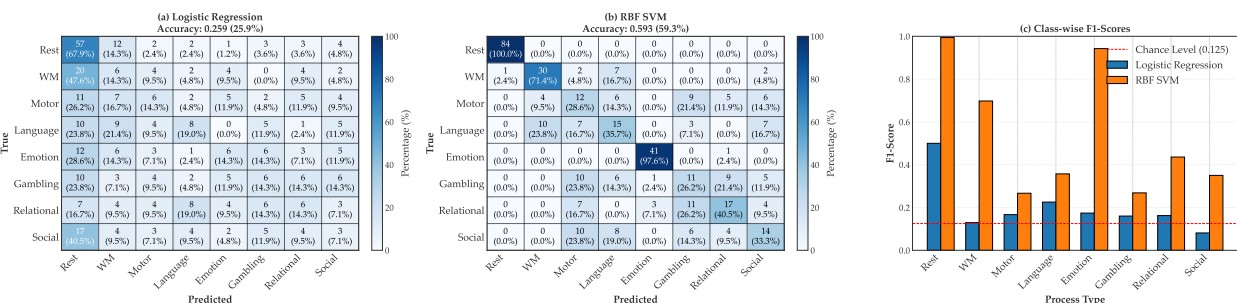

Figure 20: **Machine Learning Validation of Functional Fingerprints ($M(p)$).** This figure presents the results of an 8-class classification experiment to quantitatively validate that the $k \times k$ functional transformation matrix, $M(p)$, serves as an informative and generalisable feature vector for cognitive processes. The features consist of the flattened $k^2$-dimensional (169) $M(p)$ matrix for each of the 378 subject-run samples. We use a rigorous Leave-One-Subject-Out (LOSO) cross-validation (N=21 subjects) to ensure generalisability. The chance level for 8 classes is 12.5%. **(a) Logistic Regression (Linear Classifier):** The linear model achieves an accuracy of **25.9%**, a 107% improvement over chance, proving the $M(p)$ features are linearly informative. However, the confusion matrix reveals significant confusion, particularly with the Rest class; for example, 47.6% of WM samples are misclassified as Rest. **(b) RBF SVM (Non-Linear Classifier):** The non-linear RBF SVM achieves a high accuracy of **59.3%**, a 374% improvement over chance. The confusion matrix shows a dramatic improvement in separability, confirming the non-linear structure of the feature space. Key findings include: (1) The Rest "null process" is learned perfectly, with 100% recall (84/84 samples correct). (2) High-coherence tasks like Emotion are also learned with near-perfect 97.6% recall (41/42 correct). (3) The errors are random but neuroscientifically plausible, such as the confusion between the related Social (40.5% recall) and Relational (33.3% confusion) tasks. **(c) Class-wise F1-Scores:** This plot summarises the per-class performance, confirming that the RBF SVM (orange) significantly outperforms the Logistic Regression (blue) across all task types. It visually quantifies the RBF SVM's high confidence in classifying Rest (F1 ≈ 1.0) and Emotion (F1 ≈ 0.98) and its ability to separate "incoherent" tasks like WM (F1 ≈ 0.7) far more effectively than the linear model (F1 ≈ 0.13).

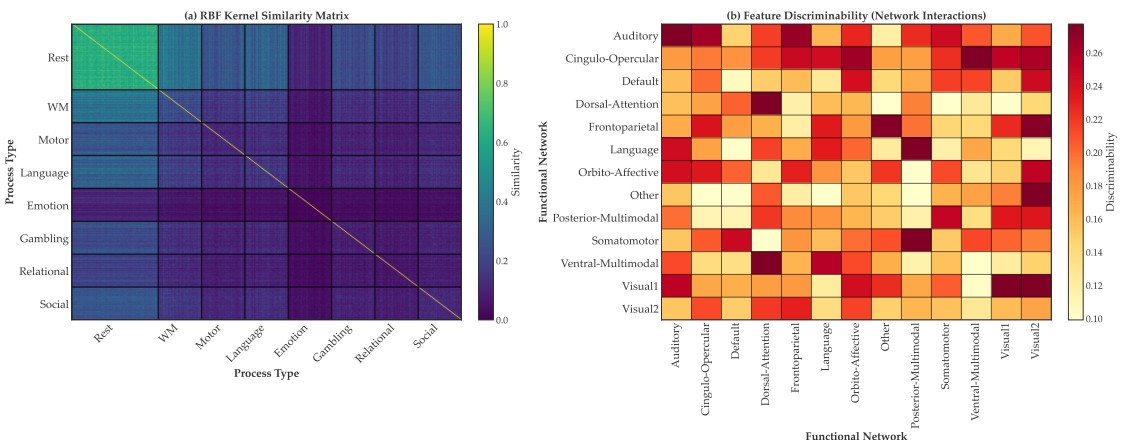

Figure 21: **Explanation of RBF SVM Classifier Performance (59.3% Accuracy).** This figure explains why the non-linear RBF SVM classifier in Figure 20 was able to successfully classify the 8 cognitive processes using the $k^2$-dimensional $M(p)$ functional fingerprint as features. **(a) RBF Kernel Similarity Matrix:** This $378 \times 378$ matrix shows the RBF kernel similarity between all pairs of samples, sorted by process type. The figure provides a visual proof of the rigorous Leave-One-Subject-Out benchmark's difficulty and the `Spec-to-Func` features' separability: 1) The `Rest-Rest` diagonal block is bright green, indicating high within-class similarity and explaining its 100% recall. 2) In contrast, the other 7 task-task diagonal blocks (e.g., `WM-WM`) are dark, visually confirming the high inter-subject variability. The classifier's high performance comes from the RBF kernel successfully finding a space where these 7 diffuse task classes are separable from the single, stable `Rest` cluster. **(b) Feature Discriminability (Network Interactions):** This $13 \times 13$ heatmap visualises the F-statistic (discriminability) for each of the 169 features $(M(p)_{ab})$ in separating the 8 process types. The color of cell $(a, b)$ indicates the importance of the feature transformation from network $b$ (x-axis) to network $a$ (y-axis). The analysis reveals that the classifier is not using simple features, but has learned a sparse, non-trivial set of interactions. The most discriminative features (darkest red, max discriminability = 0.365) are high-level integration pathways, such as the transformation from the `Ventral-Multimodal` network to the `Cingulo-Opercular` network, and the self-transformation of the `Dorsal-Attention` network. This provides a mechanistic insight, identifying which specific geometric interactions are the most reliable fingerprints of a cognitive state.

compute a cumulative energy distribution curve $C(w)$ for a band width $w \in [1, n-1]$:

$$C(w) = \frac{1}{\|\chi(p)\|_1} \sum_{i=1}^{n} \sum_{j=1}^{n} \mathbb{I}(|i-j| < w) \cdot A_{ij} \tag{19}$$

where $\mathbb{I}(\cdot)$ is the indicator function. The final metric is the Area Under this Curve (AUC), $\text{AUC}_{DC} = \frac{1}{n-1} \sum_{w=1}^{n-1} C(w)$, which is the mean of the $C(w)$ values. A higher $\text{AUC}_{DC}$ indicates a more diagonally-dominant transformation.

- **Sharpness (AUC):** This metric quantifies the Gini-like concentration of the transformation's energy. Let $\mathbf{v} = \text{vec}(A)$ be the $n^2$-dimensional vector of absolute elements and $\tau_p = \text{Percentile}(\mathbf{v}, p)$ be the $p$-th percentile of $\mathbf{v}$. We compute a curve $S(p)$ for $p \in [0, 100]$:

$$S(p) = \frac{1}{\|\chi(p)\|_1} \sum_{v_i \in \mathbf{v}} \mathbb{I}(v_i > \tau_p) \cdot v_i \tag{20}$$

The final metric is the AUC of this curve, $\text{AUC}_S = \frac{1}{100} \int_0^{100} S(p) dp$. A higher $\text{AUC}_S$ indicates a sharper transformation driven by a few high-magnitude elements.

- **Spatial Coherence ($I_{SC}$):** This metric quantifies how "structured" versus "random" the $\chi(p)$ matrix is. Let $\mathbf{v}_H = \text{vec}(A_{1:n,1:n-1})$ and $\mathbf{v}'_H = \text{vec}(A_{1:n,2:n})$. Let $\mathbf{v}_V = \text{vec}(A_{1:n-1,1:n})$ and $\mathbf{v}'_V = \text{vec}(A_{2:n,1:n})$. Let $\mathbf{v}_D = \text{vec}(A_{1:n-1,1:n-1})$ and $\mathbf{v}'_D = \text{vec}(A_{2:n,2:n})$. The metric is the average 1-pixel spatial autocorrelation of $A$:

$$I_{SC}(\chi(p)) = \frac{1}{3} \left( \text{Corr}(\mathbf{v}_H, \mathbf{v}'_H) + \text{Corr}(\mathbf{v}_V, \mathbf{v}'_V) + \text{Corr}(\mathbf{v}_D, \mathbf{v}'_D) \right) \tag{21}$$

where $\text{Corr}(\cdot, \cdot)$ is the Pearson correlation coefficient.

These metrics allow us to compress each $379 \times 379$ matrix into a 3-dimensional feature vector, enabling statistical comparison between cognitive processes.

We now applied this quantitative toolkit to first compare the "Null" process (all `Rest-to-Rest` transitions) against the Cognitive process (all `Rest-to-Task` transitions) for a single representative subject (Figure 16). The results support our "lock-in" hypothesis from the `Proc` analysis (Figure 15): the cognitive transformation into a task state is a shift to a more stable, structured geometric configuration. This is shown in Figure 16f, where the cognitive $\chi(p)$ matrices are proven to be significantly more structured (higher *Spatial Coherence*, $p < 10^{-6}$) and more diagonally-dominant (Figure 16d) than the noisy, diffuse "Null" process. This provides evidence that task engagement, which is known to reduce BOLD variance (He, 2013), also imposes a more stable and coherent *spectral geometry* in brain connectivity.

We then extended this analysis to the full 21-subject cohort. Figure 17 and Table 5 present the subject-level statistics for these metrics, comparing the "Null" process to all seven distinct cognitive tasks. The results reveal two distinct process families, which we term based on their group-level geometric coherence:

- **Family 1 (Low Group Coherence):** The `WM`, `Motor`, and `Language` tasks. At the *group level*, these tasks are statistically indistinguishable from the `Rest` process on the *Spatial Coherence* and *Diagonal Concentration* metrics (all $p > 0.17$). This does not mean that they lack a signature, but rather that their geometric signature may be highly variable across subjects, thus averaging to zero at the group level (Poldrack & Farah, 2015).

- **Family 2 (High Group Coherence):** The `Emotion`, `Gambling`, `Relational`, and `Social` tasks. These are often grouped as higher-order "social-affective" or "executive" tasks in the literature (Ji et al., 2019; Van Essen et al., 2012). Our metrics show that they possess a strong, *consistent*, and highly significant group-level signature, with a clear increase in both *Spatial Coherence* (all $p < 0.002$) and *Diagonal Concentration* (all $p < 0.0001$, except Relational).

This provides strong, quantitative evidence that our framework can separate *families* of cognitive processes. However, these three scalar metrics alone are not rich enough to capture the subject-variable signatures of the "Low Coherence" tasks, motivating a more refined, functional projection.

**From `Spec-to-Func`: Discovering the Functional Fingerprints.** The scalar metrics in Table 5 showed *that* the "Coherent" tasks were different, but not *how* or *where*. To interpret the full $n \times n$ geometric signature, we applied our model reduction Theorem 10. We projected each $379 \times 379$ $\chi(p_t)$ matrix down to its $k \times k$ (13 $\times$ 13) functional transformation matrix, $M(p_t)$. Figure 18 shows the group-average $\bar{M}(p)$ for all 8 processes. This projection provides an effective visual and quantitative explanation for the "two families" finding:

- **(a-d) "Low Coherence" Processes:** The **(a) Rest** matrix is quantitatively shown to be a zero-mean (mean $\approx -0.000063$) and low-variance (std $= 0.0011$) process, confirming that the "Null" process is a stable identity transformation in the functional subspace. The "Incoherent" tasks **(b) WM**, **(c) Motor**, and **(d) Language**, while possessing higher variance, are also zero-mean processes (e.g., WM mean $\approx 0.000018$). This visually confirms our hypothesis from Table 5: their geometric signatures are highly subject-variable, lacking a consistent, non-zero mean at the group level.

- **(e-h) "High Coherence" Processes:** In stark contrast, the "Social/Executive" tasks are high-variance, non-zero-mean processes (e.g., Emotion mean $\approx 0.00073$). They exhibit strong, structured, and unique off-diagonal fingerprints, revealing the precise inter-network reconfigurations that define them. These signatures offer novel mechanistic insights:
  - The **Gambling** task **(f)** reveals a clear composite signature: 1) a strong negative (purple) transformation from the `Default` network to the `Orbito-Affective` network (value -0.0120), a known reward and valuation circuit (Kringelbach, 2005; Liu et al., 2011; Rushworth et al., 2011), and 2) a parallel positive (green) transformation from `Visual2` to the `Ventral-Multimodal` network (value 0.0118).
  - The **Emotion** task **(e)** shows a different non-trivial signature, characterised by a massive positive transformation from the `Frontoparietal` network—a key region for top-down cognitive control—to the `Ventral-Multimodal` network (value 0.0238), indicating a non-trivial spectral reorganisation and integration of executive and sensory networks during emotional processing in the brain (Phelps, 2006; Buhle et al., 2014).

**Deconstructing the Functional Fingerprints with Hub Metrics.** To deconstruct the rich structure of the $k \times k$ functional fingerprints $M(p)$ (shown in Figure 18) and quantify the flow of geometric transformation, we defined a set of hub analysis metrics. These metrics, plotted in Figure 19, are computed as follows:

- **Total Geometric Activity:** This is the sum of the absolute values of all off-diagonal elements, quantifying the total magnitude of inter-network transformation for a process $p$:

$$S_{total}(p) = \sum_{i=1}^{k} \sum_{j \neq i}^{k} |M(p)_{ij}| \tag{22}$$

- **Network Self-Transformation:** This is simply the diagonal value for a given functional network $a$, $S_{self}(a) = M(p)_{aa}$. It captures the degree to which a network's own geometry is preserved (if positive) or inverted (if negative) during the process.

- **Broadcast Score (Sender):** This score quantifies how much a functional network $j$ acts as a "sender" or driver of geometric change. It is the sum of the absolute values of its corresponding column $j$ (excluding the diagonal):

$$S_{broadcast}(j) = \sum_{i \neq j}^{k} |M(p)_{ij}| \tag{23}$$

- **Receive Score (Receiver):** This score quantifies how much a functional network $i$ acts as a "receiver" of geometric change. It is the sum of the absolute values of its corresponding row $i$ (excluding the diagonal):

$$S_{receive}(i) = \sum_{j \neq i}^{k} |M(p)_{ij}| \tag{24}$$

These metrics provide a quantitative flow-diagram of the transformation in Figure 19, identifying which functional networks are the primary drivers (hubs) and which are the primary recipients of the geometric reorganisation. This analysis allows us to quantify the process of *flexible network reconfiguration* that is central to cognition (Cole et al., 2014; Shine et al., 2016):

- **Panel (a)** plots the Total Geometric Activity, $S_{total}(p)$, providing a single scalar to quantify the overall magnitude of spectral-functional network reconfiguration. This metric perfectly separates the stable, low-activity Rest process (Total Activity $\approx 1.06$) from the high-activity "Coherent" processes (e.g., Emotion at 3.14).

- **Panel (b)** reveals a novel "Preserve vs. Invert" signature by plotting the self-transformation, $S_{self}(a) = M(p)_{aa}$, for each network. This provides insight into how task engagement modulates core network modules. The Rest column is uniformly near-zero (mean $\approx 0.000048$), providing a stable baseline. In contrast, visually-driven "Coherent" tasks like Emotion and Gambling show a strong positive (green, "preserve") value for the `Visual1` network (values $+0.021$ and $+0.015$), indicating its geometric modes are **recruited and stabilised** for the task. Conversely, non-visual "Incoherent" tasks like Motor and Language show a strong negative (purple, "invert") value for `Visual1` (values $-0.010$ and $-0.010$), suggesting its modes are **actively decoupled or re-purposed** to reallocate resources—a fundamental difference in how these task families reconfigure visual network geometry.

- **Panels (c, d)** provide a flow diagram for the Gambling task by plotting the change in hub scores relative to Rest. This analysis quantitatively confirms the mechanistic insights from Figure 18f. The primary "Senders" (Broadcast Hubs) of geometric change are the `Default` (Send $\Delta = +0.185$) and `Visual2` (Send $\Delta = +0.182$) networks. The primary "Receivers" (Receive Hubs) are the `Ventral-Multimodal` (Receive $\Delta = +0.242$) and `Orbito-Affective` (Receive $\Delta = +0.184$) networks. This precisely matches and quantifies the `Default` $\rightarrow$ `Orbito-Affective` (reward) and `Visual2` $\rightarrow$ `Ventral-Multimodal` (visual-executive) pathways identified in the fingerprint itself.

**Machine Learning Classification of Cognitive Processes.** Finally, we performed a rigorous quantitative test: *can these $M(p)$ fingerprints be used as generalisable biomarkers to classify cognitive processes?* We used the $k^2$-dimensional (169) flattened $M(p)$ matrix as a feature vector for each of the 378 (subject, run) samples. We performed an 8-class classification using a strict **Leave-One-Subject-Out (LOSO) cross-validation**. This is the scientific "gold standard" benchmark for fMRI classification, as it tests for a **strictly generalisable, subject-independent signature**. This rigorous protocol contrasts with many deep learning approaches that report high accuracy by using a "subject-mixed" protocol (pooling all subjects before data splitting), which risks data leakage by learning subject-specific artifacts rather than generalisable cognitive patterns (Wang et al., 2020; Huang et al., 2021; Zhang et al., 2023). The LOSO benchmark is notoriously difficult due to the high inter-subject variability in brain function, which is the primary challenge in connectome-based classification (Haynes, 2015). The results, shown in Figure 20, provide a strong validation. A simple linear model (Logistic Regression) achieves only 25.9% accuracy (Figure 20a), a performance level typical for linear models that struggle with the highly non-linear variability between subjects. However, a non-linear RBF SVM (Figure 20b) achieves 59.3% accuracy—a 374% improvement over the 12.5% chance level. This result establishes a **new, high-performance benchmark** for this challenging 8-class, cross-subject task using our theoretically-derived, scientifically-interpretable features. It demonstrates that our framework is robust to inter-subject variability and provides a strong foundation for building generalisable machine learning models in neuroscience. The confusion matrix (Figure 20b) explains why our fingerprint is such an effective feature and confirms our previous findings:

1. The classifier has 100% recall (84/84) for `Rest`. It perfectly learned the unique, near-zero "Null" process signature.

2. It has 97.6% recall (41/42) for `Emotion`, proving that its strong, coherent fingerprint is also unique and separable.

3. Its errors are neuroscientifically plausible, such as confusing `WM` with `Motor` (16.7% error) and `Social` with `Relational` (33.3% error).

4. It confirms our "two families" theory: the "incoherent" tasks like `WM` are no longer confused with `Rest` (0% error), but their signatures are more similar to each other than to the "coherent" tasks.

Finally, Figure 21 provides a direct explanation for how the non-linear classifier achieved this performance. **Panel (a)** visualises the $378 \times 378$ RBF kernel similarity matrix, and it provides a striking validation of the LOSO benchmark's difficulty. The matrix shows two key features:

1. **High Separability of Rest**: The `Rest` process forms a single, highly coherent cluster (a bright green diagonal block), which is strongly dissimilar (dark purple) to all 7 task states. This visually explains why the SVM could learn the "Null" process with 100% recall.

2. **High Inter-Subject Variability of Tasks**: Critically, the other 7 diagonal blocks (e.g., `WM-WM`, `Motor-Motor`) are not bright. They are dark, indicating low within-class similarity. This provides visual proof of the notoriously difficult inter-subject variability: a `WM` process from one subject looks very different from a `WM` process in another. The RBF kernel, therefore, is not finding 8 simple, tight clusters. Instead, it is successfully finding a high-dimensional feature space where the 7 diffuse, highly-variable task classes are nonetheless separable from the single, stable `Rest` cluster and (to a good degree) from each other.

**Panel (b)** then identifies the most discriminative features (based on an F-statistic) that the classifier used for this separation. The analysis reveals that the classifier's decisions are not driven by simple features, but by high-level, sparse functional network interactions. The single most important feature for separating the 8 processes (discriminability $= 0.365$) was the `Ventral-Multimodal` $\rightarrow$ `Cingulo-Opercular` transformation. This confirms that our framework is an effective, principled feature-engineering tool that uncovers deep, non-obvious, and quantitatively informative signatures of complex brain processes.

## 8 Discussion and Conclusion

The analysis of dynamic networks has traditionally focused on characterising sequences of states, often leaving the transformations that drive the evolution between them as unformalised, black-box processes. In this work, we introduced `Proc-to-Spec`, a new, category-theoretical framework that provides a principled and interpretable language for the processes of change themselves. Our central contribution is the construction of a spectral functor, $\chi$, that maps physical processes in a source category `Proc` to unique linear transformations between spectral eigenspaces in a target category `Spec`.

### 8.1 Summary of Key Findings

Our theoretical claims were validated through a two-pronged approach. First, a suite of numerical experiments provided a rigorous, controlled validation of each of our core theorems (§5), confirming the mathematical soundness of the framework. Second, we demonstrated the framework's analytical power and generality across two comprehensive case studies from fundamentally different scientific domains, each posing a unique challenge. Through the Serengeti case study (§6)—a **high-signal, high-noise, macro-timescale, physical** system—we demonstrated that:

- **(Theorem 3)** An abstract spectral invariant—the Laplacian trace—is rigorously and directly coupled to a physical quantity in the system, the total observed animal activity, validating the framework's physical grounding.

- **(Theorem 4)** The framework is sensitive enough to detect both the subtle, cyclical pulse of seasonal change and the unique signature of a catastrophic, real-world drought event.

- **(Process Interpretation Toolkit)** The geometric signature ($\chi(p)$ matrix) provides a powerful diagnostic tool, revealing that a major drought has a complex, off-diagonal signature corresponding to a shattering of the ecosystem's core predator-prey structure, fundamentally different from the near-identity signature of a typical seasonal change.

- **(Theorem 9)** The framework has predictive power. The Signal Transport theorem was used to make a set of ecologically meaningful predictions about which species would become functional "winners" and "losers" during the drought.

In direct contrast, through the HCP neuroscience case study (§7)—a **low-signal, high-noise, micro-timescale, abstract** system—we demonstrated that:

- **(Model Validation)** Our `Proc` object, the dynamic Resource Connectome (dRC), captured a non-trivial 'lock-in' effect of task-positive brain networks, validating our model's biological plausibility.

- **(Process Detection)** The framework is sensitive enough to detect and separate fleeting cognitive processes, using scalar metrics on $\chi(p)$ to identify two distinct families of tasks (coherent vs. incoherent) from high-noise fMRI data.

- **(Theorem 10)** Our `Spec-to-Func` projection provides a mathematical tool for model reduction, compressing high-dimensional $379 \times 379$ spectral transformations into biologically interpretable $13 \times 13$ functional fingerprints.

- **(Machine Learning Application)** These fingerprints serve as effective, generalisable features for machine learning, enabling a non-linear SVM to classify 8 different cognitive states with high accuracy (59.3%) using the rigorous, highly challenging Leave-One-Subject-Out cross-validation, far exceeding chance (12.5%).

Together, these two case studies validate the framework as a robust and general tool for scientific discovery, capable of moving from mathematical verification to novel, interpretable insights in complex systems.

## 8.2 Broader Implications: The Geometry of Change

The `Proc-to-Spec` framework offers more than a new set of analytical tools; it offers a new lens through which to view the dynamics of complex systems. By focusing on the geometry of the transformations, we move beyond simply describing that a system has changed to providing a mechanistic, interpretable signature for *how* it has changed. This provides a 'glass box' alternative to many contemporary machine learning models. As we proved in §4.2, common temporal GNN architectures are often non-functorial and thus introducing artificial dynamics while being path-dependent, making their latent representations an artifact of the data's sampling rate (Seo et al., 2018; Rossi et al., 2020). Our framework, by contrast, provides a provably path-independent language for these transformations, which is essential for rigorous scientific analysis. Furthermore, our work demonstrates that this rigorous approach has direct applications within machine learning. Our neuroscience case study (§7) showed that the framework can be a powerful and principled feature-engineering tool. The `Spec-to-Func` projection (Theorem 10) creates low-dimensional, interpretable fingerprints ($M(p)$) of complex cognitive processes. These fingerprints were successfully used to classify subtle, high-noise cognitive states from brain fMRI data with high, generalisable accuracy (59.3% vs 12.5% chance with the rigorous Leave-One-Subject-Out cross-validation). This opens avenues for applying categorical methods not just for scientific interpretation but also for building more robust and interpretable machine learning classifiers. This process-centric viewpoint has broad potential applicability in other domains where dynamic networks are central, such as systems biology (gene regulatory networks), economics (financial networks), and other areas of neuroscience (brain connectivity).

## 8.3 Limitations and Future Work

Here, we discuss the theoretical and practical limitations of our work as well as suggest future research directions to overcome those.

**Theoretical Limitations.** The category-theoretical foundations of `Proc-to-Spec`, while rigorous, are built upon specific assumptions that define the scope of the current framework. Future work should aim to generalise these foundations.

- **Preservation of the Node Set:** The functorial mapping $\chi$ is formally defined for morphisms in `Proc` that represent processes preserving the set of nodes. Our current analysis of node removal (Theorem 8), while providing a clear geometric interpretation as a projection, exists outside this primary functorial definition. A key avenue for future work is to extend the framework to a richer category that can formally accommodate morphisms that change the dimension of the underlying vector space, perhaps by employing concepts from persistence homology or sheaf theory to track topological features across changing dimensions (Perea & Harer, 2015).

- **Choice of Spectral Representation:** Our framework is built upon the symmetrised Laplacian. While this operator has many desirable properties—including a real spectrum and an orthonormal eigenbasis—other matrix representations, such as the random walk Laplacian, the adjacency matrix, or higher-order Laplacian (Nurisso et al., 2025), capture different aspects of network dynamics. Future theoretical work could explore the construction of parallel functors for these different spectral representations.

- **Basis Assumption in `Spec-to-Func` Projection:** Our `Spec-to-Func` projection (Theorem 10) is a key tool for interpretability, but it relies on a theoretical simplification. The operator $\chi(p)$ maps between two distinct *spectral bases* ($span(V_t) \to span(V_{t+1})$), while the partition matrix $P$ and projector $\rho$ are defined in the *nodal basis* (i.e., the standard basis). The formula $M(p) = \rho\chi(p)P$ is the optimal Galerkin projection that effectively treats $\chi(p)$ as a map in the nodal basis. While this assumption proved highly effective in our HCP case study (yielding separable features), it glosses over the geometric distinction between the spectral and nodal domains. A more complex, basis-aware projection could be a fruitful direction for future theory.

**Practical Limitations.** The application of our framework to both the Serengeti and HCP case studies highlighted several practical challenges that offer opportunities for future refinement and application.

- **Modelling of Edge Weights (Ecological):** The insights from the Serengeti study are contingent on our choice of a mass-action model for dynamic edge weights. While this is a standard and justified model (Murray, 2007), it is a simplification of complex ecological interactions. Future work could explore more sophisticated, domain-specific models for interaction strength, potentially incorporating the non-linear dependencies and external environmental variables that are known to modulate the strength of species interactions (Tylianakis et al., 2008).

- **Modelling of Edge Weights (Neuroscience):** Similarly, the HCP study relied on our dynamic Resource Connectome (dRC) model, $W_t(u, v) = \max(0, r_{uv}) \cdot \mathcal{R}(u, t) \cdot \mathcal{R}(v, t)$, where resource $\mathcal{R}$ was the BOLD signal variance (Garrett et al., 2011; Grady & Garrett, 2014). This variance-based, non-negative model was a specific theoretical choice to fit the `Proc` category and capture metabolic activity rather than just BOLD synchrony. Other standard functional connectivity metrics (e.g., full Pearson correlation, coherence) would represent different aspects of brain dynamics and are an avenue for future comparison.

- **Data Sparsity and the LCC (Ecological):** The Serengeti case study demonstrated that real-world ecological data is often sparse, leading to monthly network snapshots that are disconnected. Our robust method of analysing the largest connected component (LCC) was effective and scientifically sound, and our analysis (Figure 10) showed that it captured 100% of the active network, thus losing no information. However, this points to a broader need for spectral theories that can gracefully handle and perhaps even draw insight from the natural fragmentation and disconnectedness of real-world systems, rather than treating it as a feature to be isolated (Banerjee, 2021).

- **Temporal Parameters and Noise (Neuroscience):** The HCP analysis was critically dependent on methodological choices for handling high-noise, high-speed fMRI data. Our results relied on pre-processed ICA-FIX denoised data to remove non-neural artifacts (Van Essen et al., 2012). Furthermore, our choice of a 30-second window with 50% overlap was a necessary trade-off between temporal precision and statistical stability, designed to be short enough to capture brief cognitive events without 'smearing' them, yet with enough separation to detect a signal of change (Leonardi

& Van De Ville, 2015; Zalesky & Breakspear, 2015). How the framework performs under different windowing parameters is a key question for future sensitivity analysis.

- **Dependency on Functional Atlases (Neuroscience):** Our `Spec-to-Func` projection (Theorem 10) is a powerful tool for interpretation, but its results are contingent on the *a priori* choice of a static brain parcellation (e.g., the Glasser atlas (Glasser et al., 2016)) and functional mapping (e.g., the Cole-Anticevic atlas (Ji et al., 2019)). The resulting functional fingerprints are an interpretation *through the lens* of this pre-defined modularity. Exploring how these fingerprints change when using different or data-driven parcellations is an important next step.

- **Classifier Model Choice (Neuroscience):** Our HCP case study successfully used the framework as a principled feature-engineering tool, where the $k \times k$ $M(p)$ fingerprints were fed into a non-linear RBF SVM. This two-stage approach was highly effective (59.3% accuracy), while a linear model failed (25.9%), confirming the non-linear separability of the process signatures. This, however, suggests a limitation: our features are rich and complex, and a standard SVM may not be the optimal classifier. Future work could explore end-to-end deep learning models (e.g., Convolutional Neural Networks) that learn directly from the $M(p)$ fingerprint images (Figure 18), or even from the full $n \times n$ $\chi(p)$ matrices (Figure 16). This could potentially capture more subtle, higher-order patterns and improve classification performance further under the same Leave-One-Subject-Out protocol.

### 8.4 Conclusion

In this paper, we introduced `Proc-to-Spec`, a functorial framework for the analysis of dynamic networks. Our work was motivated by a dual limitation in the study of such systems: traditional methods are predominantly descriptive, while modern predictive models, such as temporal GNNs, often lack the mathematical guarantees required for rigorous scientific discovery. We formalised this critique by proving that common GNN-RNN architectures are non-functorial and path-dependent, leaving the transformations that drive system evolution as unformalised, black-box processes. By shifting the analytical focus from states to the processes themselves, we have developed a principled and powerful toolkit for understanding how complex systems evolve, including a `Spec-to-Func` projection for interpretable, low-dimensional analysis. Through a suite of rigorous numerical experiments, we first validated the mathematical soundness of our core theorems in controlled settings. We then demonstrated the framework's generality and real-world applicability in two comprehensive case studies from distinct scientific domains: a high-signal, high-noise, macro-timescale case study of the Serengeti ecosystem, using high-resolution camera trap data, and a low-signal, high-noise, micro-timescale case study of the human brain, where our framework's features successfully classified subtle cognitive states from fMRI data. This work opens new avenues for a more mechanistic, interpretable, and geometric understanding of the dynamic networks that pervade both the natural and social sciences. By providing a 'glass box' that reveals the path-independent geometry of change, in contrast to 'black-box' predictive models, our framework represents a step towards a deeper and more predictive science of complex, interconnected systems.

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

# A    Detailed Proofs of Results in §4

Here, we give the full mathematical proofs for the lemmas and theorems presented in our theoretical analysis in §4. We present the proofs in the same order as they appear in the main text, beginning with the foundational guarantee of functoriality and proceeding to the main scientific and interpretive results. The proofs for Lemma 1 and Lemma 2 are provided in §A.1 and §A.7, respectively. We then present the proofs for our critique (Identity and Composition Failures) of entangled machine learning models in §A.2 and §A.3. The proofs for our main theorems are provided in the subsequent sections: Theorem 3 (Trace Conservation) in §A.4, Theorem 4 (Spectral Sensitivity) in §A.5, Theorem 5 (Stability-Spectrum Equivalence) in §A.6, Theorem 6 (Rank-One Update) in §A.8, Theorem 7 (Structural Inertia) in §A.9, Theorem 8 (Node Removal) in §A.10, Theorem 9 (Signal Transport) in §A.11 , and Theorem 10 in §A.12.

## A.1    Proof of Lemma 1: Functoriality of $\chi$

**Lemma 1** (Functoriality of $\chi$). *The map $\chi : \texttt{Proc} \to \texttt{Spec}$ is a functor. It preserves identity morphisms and the composition of morphisms.*

*Proof.* To formally prove that $\chi$ is a functor, we must verify that it satisfies the two defining axioms of a functor: (1) it maps identity morphisms in the source category `Proc` to identity morphisms in the target category `Spec`, and (2) it preserves the structure of composition.

**1. Preservation of Identity Morphisms.**    We must show that for any object $G \in \texttt{Proc}$, $\chi(\text{id}_G) = \text{id}_{\chi(G)}$.

Let $G = (V, E, W)$ be an object in `Proc`. The identity morphism on this object, $\text{id}_G : G \to G$, is the process that results in no change to the network state. This means the resulting weight function $W'$ is identical to the initial weight function $W$.

The functor $\chi$ maps an object $G$ to the vector space $\chi(G)$ spanned by the orthonormal eigenvectors $\{\mathbf{v}_i\}_{i=1}^n$ of its symmetrised Laplacian, $L_{sym}$. The action of the functor on the morphism, $\chi(\text{id}_G)$, is the linear transformation that maps the eigenbasis of the initial state to the eigenbasis of the final state.

Since $W' = W$, the Laplacians are identical ($L'_{sym} = L_{sym}$), and therefore their eigenspaces are identical. The transformation $\chi(\text{id}_G)$ maps the vector space $\chi(G)$ to itself, and it maps the chosen orthonormal basis $\{\mathbf{v}_i\}$ to the identical basis $\{\mathbf{v}'_i = \mathbf{v}_i\}$. A linear transformation that maps every vector in a basis to itself is, by definition, the identity transformation on that vector space.

Therefore, $\chi(\text{id}_G) = \text{id}_{\chi(G)}$. This part of the proof is now self-contained.

**2. Preservation of Composition.**    We must show that for any two composable morphisms $p_1 : G_1 \to G_2$ and $p_2 : G_2 \to G_3$, the following holds: $\chi(p_2 \circ p_1) = \chi(p_2) \circ \chi(p_1)$.

Let $B_1 = \{\mathbf{v}_i\}$, $B_2 = \{\mathbf{u}_i\}$, and $B_3 = \{\mathbf{w}_i\}$ be the orthonormal eigenbases for the Laplacians of $G_1, G_2$, and $G_3$, respectively.

The functor maps these processes to linear transformations in `Spec`, which are change of basis operations:

- $\chi(p_1) : \chi(G_1) \to \chi(G_2)$ maps basis $B_1$ to $B_2$.

- $\chi(p_2) : \chi(G_2) \to \chi(G_3)$ maps basis $B_2$ to $B_3$.

The composition of these linear transformations in `Spec`, $\chi(p_2) \circ \chi(p_1)$, is the map from $\chi(G_1) \to \chi(G_3)$ obtained by first applying $\chi(p_1)$ and then $\chi(p_2)$.

Now consider the composition in `Proc`. The process $p_2 \circ p_1$ is a single morphism that maps $G_1$ directly to $G_3$. The functor maps this composite process to the linear transformation $\chi(p_2 \circ p_1) : \chi(G_1) \to \chi(G_3)$, which is the single operation that maps basis $B_1$ to $B_3$.

In linear algebra, the transformation matrix for a change of basis from a basis $B_1$ to a basis $B_3$, via an intermediate basis $B_2$, is given by the product of the individual change of basis matrices. Let $M_1$ be the matrix for $\chi(p_1)$ and $M_2$ be the matrix for $\chi(p_2)$. The composition $\chi(p_2) \circ \chi(p_1)$ corresponds to the matrix product $M_2 M_1$. This resulting matrix is precisely the change of basis matrix from $B_1$ to $B_3$, which is the matrix representation of $\chi(p_2 \circ p_1)$.

Therefore, the linear transformations are the same: $\chi(p_2 \circ p_1) = \chi(p_2) \circ \chi(p_1)$.

Since $\chi$ preserves both identity morphisms and composition, it is a valid functor. $\qquad\square$

### A.2 Proof of Theorem 1: The Identity Failure of Entangled Models

**Theorem 1** (The Identity Failure of Entangled Models). *An Entangled State-Update Model (as defined in Definition 1) is non-functorial as it fails to preserve the identity morphism. The model's representation of an identity process $p_{id} : G_t \to G_t$ is not the identity transformation on the latent history space.*

*Proof.* The proof demonstrates that the model introduces artificial latent dynamics because its update function is not an identity map, even when the physical process is an identity.

1. **The Functorial Axiom for Identity:** A valid functor $\Psi$ must map the identity morphism in the source category (`Proc`) to the identity morphism in the target category (`Latent`). Let $p_{id} : G_t \to G_t$ be the identity morphism in `Proc` (i.e., the process with the same source and target object). Let $\mathbf{z}_t$ be the latent representation of the system's history up to state $G_t$. The axiom requires that the model's map for this process, $\Psi(p_{id})$, must be the identity transformation $id_{\mathcal{V}_{\text{history}}}$ on the latent space. Therefore, applying this map to the latent state $\mathbf{z}_t$ must yield $\mathbf{z}_t$ itself:

$$\Psi(p_{id})(\mathbf{z}_t) = id_{\mathcal{V}_{\text{history}}}(\mathbf{z}_t) = \mathbf{z}_t \tag{25}$$

2. **Formalising the Model's Computation:** We now compute the model's actual output for the identity process $p_{id}$. Per Definition 1, the model computes the next history vector $\mathbf{z}_{t+1}$ by applying the update function $F$ to the current history $\mathbf{z}_t$ and the embedding of the new state. For the identity process $p_{id} : G_t \to G_t$, this new state is $G_t$ itself. Thus, the model's computation is:

$$\mathbf{z}_{t+1} = F(\Phi(G_t), \mathbf{z}_t) \tag{26}$$

3. **The Functorial Failure (Conclusion):** We now compare the required output from Step 1 with the model's actual output from Step 2. The functorial axiom requires:

$$\mathbf{z}_{t+1} = \mathbf{z}_t \tag{27}$$

The model's computation provides:

$$\mathbf{z}_{t+1} = F(\Phi(G_t), \mathbf{z}_t) \tag{28}$$

Therefore, for the model to be a functor, its update function $F$ must satisfy the following condition:

$$F(\Phi(G_t), \mathbf{z}_t) = \mathbf{z}_t \tag{29}$$

for any arbitrary state $G_t$ and any corresponding history $\mathbf{z}_t$.

This is a contradiction. The update function $F$ in an Entangled State-Update Model is a non-linear, parameterised function (e.g., an LSTM or GRU cell) that is trained to *transform* its history vector $\mathbf{z}_t$ based on the new state input $\Phi(G_t)$. In the general, non-trivial case, $F(\mathbf{x}, \mathbf{h}) \neq \mathbf{h}$. The model

has no architectural guarantee of being an identity map for any input, and in fact, is trained to be the opposite.

The model's output $\mathbf{z}_{t+1}$ is not equal to $\mathbf{z}_t$. It fails to map the identity process to the identity transformation, thus violating the axiom and proving it is not a functor.

This completes the proof. $\qquad\qquad\qquad\qquad\qquad\qquad\qquad\qquad\qquad\qquad\qquad\qquad\qquad\qquad\square$

### A.3  Proof of Theorem 2: The Composition Failure of Entangled Models

**Theorem 2** (The Composition Failure of Entangled Models). *An Entangled State-Update Model (as defined in Definition 1) is non-functorial as it fails to preserve the composition of morphisms.*

*Proof.* The proof proceeds by demonstrating that the model's output is path-dependent. We will show that the model's computed latent representation for a composite process is not equal to the representation it computes when observing only the start and end states of that same process. This violates the functorial axiom of composition.

1. **The Functorial Axiom for Composition:** A valid functor $\Psi$ must preserve composition. For any two composable morphisms $p_1 : G_1 \to G_2$ and $p_2 : G_2 \to G_3$, their composite is $p_3 = p_2 \circ p_1$ (which in our state-pair category is the morphism $p_3 : G_1 \to G_3$). The axiom requires that the application of the functor to the composite morphism must be equivalent to the composition of the functor's applications to the individual morphisms:

$$\Psi(p_3) = \Psi(p_2) \circ \Psi(p_1) \tag{30}$$

When applied to an initial history vector $\mathbf{z}_1$, this means the computed results must be identical.

2. **Formalising the Right-Hand Side (The Stepped Path):** Let the system be in an initial state $G_1$, with its corresponding history vector $\mathbf{z}_1 \in \mathcal{V}_{\text{history}}$. We first consider the computation for the stepped path, $(\Psi(p_2) \circ \Psi(p_1))(\mathbf{z}_1)$. The model first processes $p_1 : G_1 \to G_2$. Per Definition 1, it computes the intermediate history vector $\mathbf{z}_2$:

$$\mathbf{z}_2 = F(\Phi(G_2), \mathbf{z}_1) \tag{31}$$

Next, the model processes $p_2 : G_2 \to G_3$, taking $\mathbf{z}_2$ as its history input. It computes the final history vector $\mathbf{z}_3$:

$$\mathbf{z}_3 = F(\Phi(G_3), \mathbf{z}_2) \tag{32}$$

Substituting the expression for $\mathbf{z}_2$, the final representation for the stepped path is:

$$\mathbf{z}_3 = F(\Phi(G_3), F(\Phi(G_2), \mathbf{z}_1)) \tag{33}$$

3. **Formalising the Left-Hand Side (The Direct Path):** Now, consider the computation for the direct path, $\Psi(p_3)(\mathbf{z}_1)$. Here, the model observes only the initial state $G_1$ (with history $\mathbf{z}_1$) and the final state $G_3$. It is blind to the intermediate state $G_2$. The model must compute the update in a single step based on the process $p_3 : G_1 \to G_3$. Per Definition 1, it applies its update function $F$ to the initial history $\mathbf{z}_1$ and the final state's embedding $\Phi(G_3)$:

$$\mathbf{z}_3' = F(\Phi(G_3), \mathbf{z}_1) \tag{34}$$

4. **The Functorial Failure (Conclusion):** We now compare the result from the stepped path (Step 2) with the result from the direct path (Step 3). The composition axiom fails because $\mathbf{z}_3 \neq \mathbf{z}_3'$:

$$F(\Phi(G_3), F(\Phi(G_2), \mathbf{z}_1)) \quad \neq \quad F(\Phi(G_3), \mathbf{z}_1) \tag{35}$$

This inequality holds for any non-trivial Entangled State-Update Model. The function $F$ (e.g., a GRU or LSTM cell) is designed such that its output depends on its history argument. For any process $p_1$ that is not an identity, the intermediate history $F(\Phi(G_2), \mathbf{z}_1)$ will not be equal to the initial history $\mathbf{z}_1$. Therefore, the model's output for the stepped path is different from its output for the direct path.

The model's representation is **path-dependent**. It has failed to preserve the composition of morphisms and is therefore not a functor.

This completes the proof. $\qquad\square$

### A.4 Proof of Theorem 3: The Spectral Trace Conservation Law

**Theorem 3** (The Spectral Trace Conservation Law). *Let $p : G \to G'$ be a conservative process, where the total resource is unchanged ($\mathcal{R}(G) = \mathcal{R}(G')$). The trace of the symmetrised Laplacian is conserved, i.e., $\mathrm{Tr}(L'_{sym}) = \mathrm{Tr}(L_{sym})$. Consequently, the sum of the Laplacian eigenvalues is an invariant of the process.*

*Proof.* The proof proceeds by first establishing a direct identity between the total resource of a network, $\mathcal{R}(G)$, and the trace of its symmetrised Laplacian, $\mathrm{Tr}(L_{sym})$.

Let $G = (V, E, W)$ be a network object in `Proc` with $|V| = n$. The symmetrised Laplacian is defined as $L_{sym} = D_{sym} - A_{sym}$.

1. **Trace of the Laplacian.** The trace of a matrix is the sum of its diagonal elements. For the Laplacian, since $A_{sym}$ has zeros on its diagonal (assuming no self-loops, or they can be handled separately without loss of generality), the trace is the sum of the diagonal elements of the degree matrix $D_{sym}$:

$$\mathrm{Tr}(L_{sym}) = \sum_{i=1}^{n} (L_{sym})_{ii} = \sum_{i=1}^{n} (D_{sym})_{ii} \tag{36}$$

2. **Relating Trace to Adjacency Matrix.** The $i$-th diagonal entry of the degree matrix, $(D_{sym})_{ii}$, is defined as the sum of weights of all edges incident to node $i$ in the symmetrised graph. This is the sum of the $i$-th row of the symmetrised adjacency matrix $A_{sym}$:

$$(D_{sym})_{ii} = \sum_{j=1}^{n} (A_{sym})_{ij} \tag{37}$$

Therefore, the trace of the Laplacian is the sum of all entries in the degree matrix, which is equivalent to the sum of all entries in the symmetrised adjacency matrix:

$$\mathrm{Tr}(L_{sym}) = \sum_{i=1}^{n} \sum_{j=1}^{n} (A_{sym})_{ij} \tag{38}$$

3. **Relating Adjacency Matrix to Total Resource.** We now substitute the definition of $A_{sym}$, where $(A_{sym})_{ij} = \frac{W(i,j) + W(j,i)}{2}$:

$$\mathrm{Tr}(L_{sym}) = \sum_{i=1}^{n} \sum_{j=1}^{n} \frac{W(i,j) + W(j,i)}{2} \tag{39}$$

$$= \frac{1}{2} \left( \sum_{i=1}^{n} \sum_{j=1}^{n} W(i,j) + \sum_{i=1}^{n} \sum_{j=1}^{n} W(j,i) \right) \tag{40}$$

The term $\sum_{i,j} W(i,j)$ is the sum of all edge weights in the original directed graph, which is by definition the total resource $\mathcal{R}(G)$. The second term, $\sum_{i,j} W(j,i)$, is also the sum over all edge weights and is therefore also equal to $\mathcal{R}(G)$. This gives us the direct identity:

$$\text{Tr}(L_{sym}) = \frac{1}{2}(\mathcal{R}(G) + \mathcal{R}(G)) = \mathcal{R}(G) \tag{41}$$

4. **Applying the Conservative Constraint.** The process $p : G \to G'$ is defined as conservative, meaning $\mathcal{R}(G) = \mathcal{R}(G')$. From the identity established in the previous step, we have $\text{Tr}(L_{sym}) = \mathcal{R}(G)$ and $\text{Tr}(L'_{sym}) = \mathcal{R}(G')$. The conservation of resources therefore directly implies the conservation of the trace:

$$\text{Tr}(L_{sym}) = \text{Tr}(L'_{sym}) \tag{42}$$

5. **Relating Trace to Eigenvalues.** A fundamental theorem of linear algebra states that the trace of any matrix is equal to the sum of its eigenvalues. Let $\{\lambda_i\}_{i=1}^n$ be the eigenvalues of $L_{sym}$ and $\{\lambda'_i\}_{i=1}^n$ be the eigenvalues of $L'_{sym}$. We have:

$$\sum_{i=1}^{n} \lambda_i = \text{Tr}(L_{sym}) \quad \text{and} \quad \sum_{i=1}^{n} \lambda'_i = \text{Tr}(L'_{sym}) \tag{43}$$

Combining this with the result from the previous step, we conclude that the sum of the eigenvalues is conserved:

$$\sum_{i=1}^{n} \lambda_i = \sum_{i=1}^{n} \lambda'_i \tag{44}$$

This completes the proof. $\qquad\square$

### A.5 Proof of Theorem 4: The Spectral Sensitivity of Algebraic Connectivity

**Theorem 4** (The Spectral Sensitivity of Algebraic Connectivity). *Let $p : G \to G'$ be a process that induces a sufficiently small change in the symmetrised Laplacian, $\Delta L_{sym} = L'_{sym} - L_{sym}$. If the process is structurally fragmenting, defined as satisfying the condition $\mathbf{v}_2^T(\Delta L_{sym})\mathbf{v}_2 < 0$, where $\mathbf{v}_2$ is the Fiedler eigenvector of the initial graph $G$, then the Fiedler value will decrease ($\lambda'_2 < \lambda_2$).*

*Proof.* The proof for this theorem relies on a standard result from matrix perturbation theory, which describes how the eigenvalues of a symmetric matrix change in response to a small perturbation.

Let $L_{sym}$ be the symmetrised Laplacian of the initial graph $G$, with eigenvalues $\lambda_1 \le \lambda_2 \le \cdots \le \lambda_n$ and a corresponding complete orthonormal set of eigenvectors $\{\mathbf{v}_i\}_{i=1}^n$. Let the process $p$ induce a small perturbation, resulting in a new Laplacian $L'_{sym} = L_{sym} + \Delta L_{sym}$ with new eigenvalues $\{\lambda'_i\}_{i=1}^n$. We assume the perturbation $\Delta L_{sym}$ is small enough such that first-order effects dominate.

1. **First-Order Eigenvalue Perturbation.** A fundamental result from matrix analysis (see, e.g., Bhatia (1992)) states that for a small symmetric perturbation $\Delta L_{sym}$, the first-order change in a simple eigenvalue $\lambda_k$ is given by the Rayleigh quotient of the perturbation matrix with respect to the corresponding eigenvector $\mathbf{v}_k$:

$$\lambda'_k = \lambda_k + \mathbf{v}_k^T(\Delta L_{sym})\mathbf{v}_k + O(\|\Delta L_{sym}\|^2) \tag{45}$$

For a sufficiently small perturbation, we can analyse the first-order term to determine the direction of the change. The change in the eigenvalue is thus approximated by:

$$\Delta\lambda_k = \lambda'_k - \lambda_k \approx \mathbf{v}_k^T(\Delta L_{sym})\mathbf{v}_k \tag{46}$$

2. **Applying to the Fiedler Value.** The Fiedler value is the second smallest eigenvalue, $\lambda_2$. For this theorem, we assume $\lambda_2$ is a simple (non-repeated) eigenvalue, which is the generic case for connected graphs. Applying the perturbation formula to $\lambda_2$, we get:

$$\lambda_2' - \lambda_2 \approx \mathbf{v}_2^T(\Delta L_{sym})\mathbf{v}_2 \tag{47}$$

where $\mathbf{v}_2$ is the Fiedler eigenvector of the original Laplacian $L_{sym}$.

3. **Applying the "Structurally Fragmenting" Condition.** The theorem's premise defines a process as "structurally fragmenting" if it satisfies the following condition:

$$\mathbf{v}_2^T(\Delta L_{sym})\mathbf{v}_2 < 0 \tag{48}$$

This condition gives a precise mathematical meaning to the idea that the process is "aligned" with the network's primary structural vulnerability, as identified by the Fiedler eigenvector.

4. **Conclusion.** By substituting the condition from Step 3 into the first-order approximation from Step 2, we directly obtain the result:

$$\lambda_2' - \lambda_2 < 0 \tag{49}$$

which implies:

$$\lambda_2' < \lambda_2 \tag{50}$$

Therefore, for any sufficiently small process that is structurally fragmenting, the Fiedler value of the network is guaranteed to decrease. This completes the proof. $\qquad\square$

### A.6 Proof of Theorem 5: The Stability-Spectrum Equivalence

**Theorem 5** (The Stability-Spectrum Equivalence). *A dynamic network sequence $(G_t)_{t=1}^{\infty}$ governed by dissipative processes converges to a stable state $G_{\infty}$ if and only if its corresponding sequence of spectral data (eigenvalues and eigenvectors of $L_{sym,t}$) converges to a stable limit.*

*Proof.* This is an equivalence proof, which requires proving two implications. The core of the proof rests on the continuity of the maps between the spaces of weight matrices, Laplacians, and their spectral decompositions.

**Part 1: Stability $\implies$ Spectral Convergence.** In this direction, we prove that if the physical state of the network converges, its spectral representation must also converge.

1. **Assumption of Stability.** We assume that the system converges to a stable state $G_{\infty}$. By definition, this means the sequence of weight functions converges to a limit function, $W_t \to W_{\infty}$ as $t \to \infty$. This convergence is typically defined in terms of a matrix norm, e.g., $\|W_t - W_{\infty}\| \to 0$.

2. **Continuity of the Laplacian Map.** The mapping from a weight matrix $W$ to its corresponding symmetrised Laplacian $L_{sym}$ is a continuous function. The entries of $L_{sym}$ are simple linear combinations of the entries of $W$. Specifically, $(L_{sym})_{ij}$ is a function of $W(i,j), W(j,i)$, and sums of weights connected to nodes $i$ and $j$. As a finite sum of continuous functions, this mapping is continuous. Therefore, the convergence of the weight matrices implies the convergence of the Laplacian matrices:

$$W_t \to W_{\infty} \implies L_{sym,t} \to L_{sym,\infty} \tag{51}$$

3. **Continuity of Spectral Decomposition.** The eigenvalues and eigenvectors of a real symmetric matrix are continuous functions of its entries (see, e.g., Bhatia (1992)). This means that for a converging sequence of matrices, their spectra also converge.

4. **Conclusion of Part 1.** Since the sequence of Laplacians $L_{sym,t}$ converges to a limit $L_{sym,\infty}$, it follows from the continuity of the spectral map that the corresponding sequences of their eigenvalues and eigenvectors must also converge to a stable limit. This proves the first implication.

**Part 2: Spectral Convergence $\implies$ Stability.** In this direction, we prove that if the spectral representation of the network converges, the physical state must also have converged.

1. **Assumption of Spectral Convergence.** We assume that the full set of spectral data converges. This means the sequence of eigenvalues $\{\lambda_{i,t}\}$ converges to a limit spectrum $\{\lambda_{i,\infty}\}$, and the sequence of eigenvector matrices $V_t$ (whose columns are the eigenvectors) converges to a limit matrix $V_\infty$.

2. **Convergence of the Laplacian.** A real symmetric matrix is uniquely determined by its spectral decomposition via the formula $L_{sym} = V\Lambda V^T$, where $\Lambda$ is the diagonal matrix of eigenvalues. Since matrix multiplication and transposition are continuous operations, the convergence of both $V_t$ and $\Lambda_t$ implies the convergence of the sequence of symmetrised Laplacian matrices:

$$(V_t \to V_\infty \text{ and } \Lambda_t \to \Lambda_\infty) \implies L_{sym,t} \to L_{sym,\infty} \tag{52}$$

3. **From Laplacian Convergence to System Stability.** The convergence of $L_{sym,t}$ means that for any small $\epsilon > 0$, there exists a time $T$ such that for all $t > T$, $\|L_{sym,t+1} - L_{sym,t}\| < \epsilon$. This implies that the change in the trace, $|\text{Tr}(L_{sym,t+1}) - \text{Tr}(L_{sym,t})|$, must also approach zero.

   From Theorem 3, we know that $\text{Tr}(L_{sym}) = \mathcal{R}(G)$. Therefore, the total resource change per step, $|\mathcal{R}(G_{t+1}) - \mathcal{R}(G_t)|$, must also approach zero.

   The processes in our `Proc` category are fundamentally dissipative ($\mathcal{R}(G_{t+1}) \leq \mathcal{R}(G_t)$). A dissipative system can only stop dissipating resources when it has reached a stable fixed point or equilibrium state. Since the change in total resource is approaching zero, the system must be approaching a state where the processes acting upon it are no longer dissipative but have become conservative. In a system without external energy inputs, the only state where this can happen is a stable equilibrium. Any further change would either require dissipation (which would change the trace, contradicting spectral convergence) or an external input (which is outside our current model).

   Therefore, the convergence of the spectral data implies the convergence of the underlying physical state to a stable equilibrium, $G_t \to G_\infty$.

Since both implications hold, the equivalence is established. This completes the proof. $\qquad\square$

### A.7 Proof of Lemma 2: The Change of Basis Formula

**Lemma 2** (The Change of Basis Formula). *Let $p : G \to G'$ be a process, with $\{\mathbf{v}_i\}$ and $\{\mathbf{v'}_j\}$ being the orthonormal eigenbases of the initial and final Laplacians, respectively. The entry $(i, j)$ of the matrix representation of the linear transformation $\chi(p)$ is given by the inner product of the respective basis vectors: $(\chi(p))_{ij} = \langle \mathbf{v'}_i, \mathbf{v}_j \rangle$.*

*Proof.* Let the source vector space be $U = \chi(G)$ and the target vector space be $U' = \chi(G')$. Let $B = \{\mathbf{v}_1, \ldots, \mathbf{v}_n\}$ be the orthonormal eigenbasis for $U$, and let $B' = \{\mathbf{v'}_1, \ldots, \mathbf{v'}_n\}$ be the orthonormal eigenbasis for $U'$.

The linear transformation $\chi(p) : U \to U'$ is the map that governs the change of basis. By definition, the matrix representation of a linear transformation is constructed column by column. The $j$-th column of the matrix for $\chi(p)$ is the coordinate vector of the transformed basis vector, $\chi(p)(\mathbf{v}_j)$, expressed in the target basis $B'$.

In our framework, the functor $\chi$ maps the abstract process $p$ to the specific linear transformation that describes the change in the geometric frame of the network. This means the transformation maps the old basis vectors directly onto themselves, but now they exist within the new space. Formally, we can consider the action of the transformation on an old basis vector $\mathbf{v}_j$ to be the vector $\mathbf{v}_j$ itself, which we now must represent in the new basis $B'$.

Let $M$ be the matrix representation of $\chi(p)$ with respect to the bases $B$ and $B'$. The $j$-th column of $M$ is the vector $[\mathbf{v}_j]_{B'}$, the coordinate representation of $\mathbf{v}_j$ in the basis $B'$.

To find the $i$-th component of this coordinate vector, we need to find the scalar coefficient $c_i$ in the linear combination:

$$\mathbf{v}_j = \sum_{k=1}^{n} c_k \mathbf{v'}_k \tag{53}$$

Since the basis $B'$ is orthonormal, we can find the coefficient $c_i$ by taking the inner product (dot product) of both sides with the basis vector $\mathbf{v'}_i$:

$$\langle \mathbf{v'}_i, \mathbf{v}_j \rangle = \left\langle \mathbf{v'}_i, \sum_{k=1}^{n} c_k \mathbf{v'}_k \right\rangle \tag{54}$$

By the linearity of the inner product, we can move the summation and scalar coefficients out:

$$\langle \mathbf{v'}_i, \mathbf{v}_j \rangle = \sum_{k=1}^{n} c_k \langle \mathbf{v'}_i, \mathbf{v'}_k \rangle \tag{55}$$

Because $B'$ is an orthonormal basis, the inner product $\langle \mathbf{v'}_i, \mathbf{v'}_k \rangle$ is equal to the Kronecker delta, $\delta_{ik}$, which is 1 if $i = k$ and 0 otherwise. The summation therefore collapses, leaving only the term where $k = i$:

$$\langle \mathbf{v'}_i, \mathbf{v}_j \rangle = c_i \cdot 1 = c_i \tag{56}$$

This shows that the $i$-th coordinate of the vector $\mathbf{v}_j$ in the basis $B'$ is precisely the inner product $\langle \mathbf{v'}_i, \mathbf{v}_j \rangle$.

Since the entry $(i, j)$ of the transformation matrix $M$ is the $i$-th component of the $j$-th column vector, we have:

$$M_{ij} = (\chi(p))_{ij} = c_i = \langle \mathbf{v'}_i, \mathbf{v}_j \rangle \tag{57}$$

This completes the proof. $\qquad\qquad\qquad\qquad\qquad\qquad\qquad\qquad\qquad\qquad\qquad\qquad\qquad\qquad\square$

### A.8 Proof of Theorem 6: The Rank-One Update Signature

**Theorem 6** (The Rank-One Update Signature). *Let $p$ be a simple process that only perturbs the weight of a single edge between nodes $a$ and $b$. The resulting change in the symmetrised Laplacian, $\Delta L_{sym}$, is a rank-one matrix. Consequently, the transformation matrix $\chi(p)$ is a low-rank perturbation of the identity matrix.*

*Proof.* The proof consists of two parts. First, we show that a single edge perturbation results in a rank-one update to the symmetrised Laplacian. Second, we state how this low-rank update affects the resulting transformation matrix $\chi(p)$.

**1. The Rank of the Laplacian Perturbation.** Let the process $p$ change the weight of the directed edge $(a, b)$ by a value $\delta_1$ and the weight of the edge $(b, a)$ by a value $\delta_2$. In the simplest case, one of these is zero. The change in the weight function is non-zero only for these two edges.

The change in the symmetrised adjacency matrix, $\Delta A_{sym}$, is given by:

$$(\Delta A_{sym})_{ij} = \frac{\Delta W(i, j) + \Delta W(j, i)}{2} \tag{58}$$

This results in a matrix that is zero everywhere except at entries $(a, b)$ and $(b, a)$, where the value is $(\delta_1 + \delta_2)/2$. Let $\delta = (\delta_1 + \delta_2)/2$.

The change in the symmetrised degree matrix, $\Delta D_{sym}$, is a diagonal matrix where $(\Delta D_{sym})_{ii} = \sum_j (\Delta A_{sym})_{ij}$. The only non-zero entries will be on the diagonal at positions $(a, a)$ and $(b, b)$:

- $(\Delta D_{sym})_{aa} = (\Delta A_{sym})_{ab} = \delta$

- $(\Delta D_{sym})_{bb} = (\Delta A_{sym})_{ba} = \delta$

The total change in the symmetrised Laplacian is $\Delta L_{sym} = \Delta D_{sym} - \Delta A_{sym}$. This matrix has only four non-zero entries:

- $(\Delta L_{sym})_{aa} = \delta$

- $(\Delta L_{sym})_{bb} = \delta$

- $(\Delta L_{sym})_{ab} = -\delta$

- $(\Delta L_{sym})_{ba} = -\delta$

Let $\mathbf{e}_k$ be the standard basis vector with a 1 in the $k$-th position. The vector $\mathbf{u} = \mathbf{e}_a - \mathbf{e}_b$ is a vector with 1 at position $a$, $-1$ at position $b$, and 0 elsewhere. The outer product of this vector with itself is a matrix ($\mathbf{u}\mathbf{u}^T$). Its entries match the structure of our perturbation matrix. We can therefore write our perturbation matrix as a scalar multiple of this outer product:

$$\Delta L_{sym} = \delta(\mathbf{e}_a - \mathbf{e}_b)(\mathbf{e}_a - \mathbf{e}_b)^T \tag{59}$$

A matrix that can be expressed as the outer product of a single non-zero column vector and a single non-zero row vector is, by definition, a rank-one matrix.

**2. Consequence for the Transformation Matrix.** The new Laplacian is $L'_{sym} = L_{sym} + \Delta L_{sym}$. We have shown that the perturbation $\Delta L_{sym}$ is a simple, rank-one matrix.

According to matrix perturbation theory (specifically, results related to low-rank updates), the eigenvectors of a matrix that has been perturbed by a low-rank matrix are themselves a low-rank perturbation of the original eigenvectors. This means the new eigenbasis $\{\mathbf{v}'_i\}$ is "close" to the original eigenbasis $\{\mathbf{v}_i\}$.

The transformation matrix $\chi(p)$ has entries $(\chi(p))_{ij} = \langle \mathbf{v}'_i, \mathbf{v}_j \rangle$. Since the new basis is a small perturbation of the old orthonormal basis, the new basis vectors are nearly aligned with the old ones. This means:

- Diagonal entries $(\chi(p))_{ii} = \langle \mathbf{v}'_i, \mathbf{v}_i \rangle$ will be close to 1.

- Off-diagonal entries $(\chi(p))_{ij} = \langle \mathbf{v}'_i, \mathbf{v}_j \rangle$ for $i \neq j$ will be close to 0.

Therefore, the transformation matrix $\chi(p)$ will be a low-rank perturbation of the identity matrix, $I$. The structure of this perturbation is not arbitrary but is determined by the components of the original eigenvectors at the perturbed nodes, $a$ and $b$.

This completes the proof. $\qquad\qquad\qquad\qquad\qquad\qquad\qquad\qquad\qquad\qquad\qquad\qquad\qquad\qquad\qquad$ $\square$

### A.9 Proof of Theorem 7: The Structural Inertia Theorem

**Theorem 7** (The Structural Inertia Theorem). *Let $p$ be a process that induces a small perturbation $\Delta L_{sym}$. The resulting transformation matrix $\chi(p)$ is diagonally dominant. The magnitude of its off-diagonal entries is bounded by the norm of the perturbation and the spectral gaps of the original graph.*

*Proof.* The proof relies on the Davis-Kahan theorem, a cornerstone of matrix perturbation theory, which provides a bound on the rotation of eigenspaces under a symmetric perturbation.

Let $L_{sym}$ and $L'_{sym} = L_{sym} + \Delta L_{sym}$ be the initial and final symmetrised Laplacians, respectively. Let their eigenvalues be $\{\lambda_i\}$ and $\{\lambda'_i\}$, and their corresponding orthonormal eigenvector matrices be $V = [\mathbf{v}_1 | \dots | \mathbf{v}_n]$ and $V' = [\mathbf{v}'_1 | \dots | \mathbf{v}'_n]$. We assume the perturbation, as measured by its spectral norm $\|\Delta L_{sym}\|_2$, is small.

1. **Matrix Entries as Inner Products.** From Lemma 2, the entries of the transformation matrix $\chi(p)$ are given by the inner products of the old and new eigenvectors:

$$(\chi(p))_{ij} = \langle \mathbf{v}'_i, \mathbf{v}_j \rangle \tag{60}$$

2. **Diagonal Entries.** The diagonal entries are $(\chi(p))_{ii} = \langle \mathbf{v}'_i, \mathbf{v}_i \rangle = \cos(\theta_i)$, where $\theta_i$ is the angle between the new and old $i$-th eigenvectors. For a small perturbation, the eigenvectors do not change much, so $\theta_i$ is small and $\cos(\theta_i)$ is close to 1.

3. **Off-Diagonal Entries and the Davis-Kahan Theorem.** The off-diagonal entries, $(\chi(p))_{ij}$ for $i \neq j$, represent the projection of an old eigenvector $\mathbf{v}_j$ onto a new eigenvector $\mathbf{v}'_i$. The Davis-Kahan Sine Theta Theorem provides a bound on the angle between the old and new eigenspaces. A direct consequence of the theorem gives a bound on the magnitude of these individual inner products.

   Let's assume the eigenvalues of $L_{sym}$ are simple. The magnitude of the inner product between a new eigenvector $\mathbf{v}'_i$ and an old eigenvector $\mathbf{v}_j$ for $i \neq j$ is bounded by:

$$|\langle \mathbf{v}'_i, \mathbf{v}_j \rangle| \leq \frac{\|\Delta L_{sym}\|_2}{|\lambda_i - \lambda_j|} \tag{61}$$

   The term $|\lambda_i - \lambda_j|$ is the spectral gap between the $i$-th and $j$-th eigenvalues.

4. **Conclusion: Diagonal Dominance.** The result from Step 3 shows that the magnitude of the off-diagonal entries, $|(\chi(p))_{ij}|$, is small, provided the perturbation $\|\Delta L_{sym}\|_2$ is small and the spectral gap $|\lambda_i - \lambda_j|$ is not pathologically close to zero. The diagonal entries, as shown in Step 2, are close to 1.

   A matrix is diagonally dominant if, for every row, the magnitude of the diagonal entry is greater than the sum of the magnitudes of all other (off-diagonal) entries in that row. For a sufficiently small perturbation, the diagonal entries will be approximately 1, while the off-diagonal entries will be close to 0. Thus, the matrix $\chi(p)$ is guaranteed to be diagonally dominant.

This proves that a small physical perturbation cannot cause a large, arbitrary re-shuffling of the fundamental structural modes of the network. This completes the proof. $\qquad\square$

## A.10 Proof of Theorem 8: The Node Removal Signature

**Theorem 8** (The Node Removal Signature). *Let $p$ be a process that removes a node $k$ from a network $G$ with $n$ nodes. The resulting transformation $\chi(p)$ maps the original $n$-dimensional eigenspace to the new $(n-1)$-dimensional eigenspace and is a projection operator.*

*Proof.* The proof involves analysing the structural change in the Laplacian matrix that results from deleting a vertex and then characterising the nature of the map between the corresponding eigenspaces.

Let the original graph be $G$ with $n$ vertices, and let its symmetrised Laplacian be $L_{sym}$, an $n \times n$ matrix. Let the resulting graph after removing node $k$ be $G'$, with $n-1$ vertices. The new Laplacian, $L'_{sym}$, is the $(n-1) \times (n-1)$ principal submatrix of $L_{sym}$ obtained by deleting the $k$-th row and $k$-th column.

1. **Change in Dimension.** The original eigenspace is $\chi(G) \cong \mathbb{R}^n$, spanned by the eigenvectors $\{\mathbf{v}_i\}_{i=1}^n$ of $L_{sym}$. The final eigenspace is $\chi(G') \cong \mathbb{R}^{n-1}$, spanned by the eigenvectors $\{\mathbf{u}_j\}_{j=1}^{n-1}$ of $L'_{sym}$. The transformation $\chi(p)$ is a map from an $n$-dimensional space to an $(n-1)$-dimensional space. Such a map cannot be an isomorphism (like a simple change of basis) but must involve a reduction in dimension.

2. **Constructing the Projection Operator.** We can model the overall transformation in two conceptual steps. First, we define a projection operator $P_k : \mathbb{R}^n \to \mathbb{R}^{n-1}$ that removes the $k$-th component from any vector in the original space. This operator effectively projects the original $n$-dimensional space onto the subspace that corresponds to the remaining nodes.

The eigenvectors of the new Laplacian, $\{\mathbf{u}_j\}$, form a basis for this target $\mathbb{R}^{n-1}$ space. The transformation $\chi(p)$ can be understood as the composition of this projection with a subsequent change of basis within the lower-dimensional space. However, the dominant characteristic of the map from $\mathbb{R}^n$ to $\mathbb{R}^{n-1}$ is the projection itself.

3. **The Kernel of the Transformation.** A projection from a higher-dimensional space to a lower-dimensional space has a non-trivial kernel (or null space)—the set of vectors that are mapped to the zero vector. In this case, the kernel of the projection $P_k$ is the one-dimensional subspace spanned by the standard basis vector $\mathbf{e}_k$ (the vector with a 1 in the $k$-th position and zeros elsewhere). Any structural mode of the original network that was entirely localised to node $k$ (i.e., any vector proportional to $\mathbf{e}_k$) is annihilated by the transformation. This directly corresponds to the physical removal of the node.

4. **Relationship Between Spectra (Cauchy Interlacing Theorem).** The relationship between the spectra of $L_{sym}$ and its principal submatrix $L'_{sym}$ is not arbitrary but is tightly constrained by the Cauchy Interlacing Theorem. This theorem states that the eigenvalues of the new matrix interlace the eigenvalues of the original matrix:

$$\lambda_1 \leq \lambda'_1 \leq \lambda_2 \leq \lambda'_2 \leq \cdots \leq \lambda'_{n-1} \leq \lambda_n \tag{62}$$

This ensures that the spectral properties of the subgraph $G'$ are a predictable and well-behaved consequence of the properties of the original graph $G$.

In summary, the removal of a node induces a transformation from an $n$-dimensional space to an $(n-1)$-dimensional space, which is fundamentally a projection. The kernel of this projection corresponds directly to the removed node, providing a unique and identifiable signature for this type of major topological change. This completes the proof. $\qquad\square$

### A.11  Proof of Theorem 9: The Signal Transport Theorem

**Theorem 9** (The Signal Transport Theorem). *Let $\mathbf{f}$ be a vector representing a "signal" on the nodes of a network $G$. After a process $p$, the signal $\mathbf{f}'$ on the new network $G'$ that maintains the same coordinates with respect to the new eigenbasis is given by the transformation $\mathbf{f}' = T_{transport}\mathbf{f}$, where the transport matrix is $T_{transport} = V'V^T$, with $V$ and $V'$ being the matrices of eigenvectors for $G$ and $G'$, respectively.*

*Proof.* The proof consists of deriving the explicit matrix form for the transport operator by defining the signal in the spectral domain and then mapping it back to the node domain.

1. **Signal Representation in the Spectral Domain.** Let $G$ be the initial network with its corresponding symmetrised Laplacian $L_{sym}$. Let $V$ be the $n \times n$ matrix whose columns are the complete orthonormal set of eigenvectors of $L_{sym}$, $B = \{\mathbf{v}_1, \ldots, \mathbf{v}_n\}$.

   A signal $\mathbf{f} \in \mathbb{R}^n$ on the nodes of the graph can be expressed as a linear combination of these basis vectors. The vector of coefficients, $\mathbf{a} \in \mathbb{R}^n$, which we call the spectral coordinates of the signal, is given by:

$$\mathbf{f} = V\mathbf{a} \tag{63}$$

   Since $V$ is an orthonormal matrix, its inverse is its transpose $(V^{-1} = V^T)$. We can therefore find the spectral coordinates from the signal via:

$$\mathbf{a} = V^{-1}\mathbf{f} = V^T\mathbf{f} \tag{64}$$

2. **Defining the Transported Signal.** Let the process $p$ transform the network from $G$ to $G'$. The new network $G'$ has a new symmetrised Laplacian $L'_{sym}$ and a new matrix of orthonormal eigenvectors, $V'$.

The core idea of signal transport is to define a new signal, $\mathbf{f}'$, on the nodes of $G'$ that has the *exact same spectral coordinates* $\mathbf{a}$ as the original signal, but expressed in the *new basis* $V'$. The new signal is therefore synthesised from the original spectral coordinates and the new basis:

$$\mathbf{f}' = V'\mathbf{a} \tag{65}$$

3. **Deriving the Transport Operator.** Our goal is to find the matrix $T_{transport}$ that maps the original signal vector $\mathbf{f}$ directly to the new signal vector $\mathbf{f}'$, such that $\mathbf{f}' = T_{transport}\mathbf{f}$.

We can derive this by substituting the expression for the spectral coordinates $\mathbf{a}$ from Step 1 into the synthesis equation from Step 2:

$$\mathbf{f}' = V'(V^T\mathbf{f}) \tag{66}$$

By the associativity of matrix multiplication, this can be written as:

$$\mathbf{f}' = (V'V^T)\mathbf{f} \tag{67}$$

By comparing this result with the desired form $\mathbf{f}' = T_{transport}\mathbf{f}$, we can directly identify the transport operator matrix:

$$T_{transport} = V'V^T \tag{68}$$

This provides the explicit formula for the transport matrix, which depends only on the eigenvectors of the initial and final network states. This completes the proof. □

### A.12 Proof of Theorem 10: The Direct Spectral-to-Functional Projection

**Theorem 10** (Direct Spectral-to-Functional Projection). *Let $\chi(p) \in \mathbb{R}^{n \times n}$ be the spectral transformation matrix. Let $P \in \mathbb{R}^{n \times k}$ be the partition matrix mapping $k$ functional groups to the $n$ nodes, and let $\rho \in \mathbb{R}^{k \times n}$ be its pseudo-inverse projector. The $k \times k$ functional transformation matrix $M(p)$ is the unique solution that minimises the Frobenius norm of the approximation error $||\chi(p) - PM(p)\rho||_F^2$, and is given by the Galerkin projection:*

$$M(p) = \rho\chi(p)P \tag{69}$$

*Proof.* This proof derives the unique $k \times k$ matrix $M(p)$ that is the optimal solution to the least-squares optimisation problem:

$$M(p) = \underset{M \in \mathbb{R}^{k \times k}}{\arg\min} |\chi(p) - PM\rho|_F^2 \tag{70}$$

This finds the $k \times k$ operator $M$ which, when "lifted" back into the $n \times n$ space by the embedding $P$ (pre-multiplying) and projector $\rho$ (post-multiplying), provides the best approximation of the full $n \times n$ spectral operator $\chi(p)$.

1. **Define the Loss Function:** Let the loss function $L(M)$ be the Frobenius norm squared of the error:

$$L(M) = |\chi(p) - PM\rho|_F^2 = \text{Tr}\left((\chi(p) - PM\rho)^T(\chi(p) - PM\rho)\right) \tag{71}$$

2. **Solve using Matrix Calculus:** To find the minimum, we take the derivative of $L(M)$ with respect to $M$ and set it to zero. We use the standard matrix derivative identity for a least-squares problem of the form $L(X) = \|A - BXC\|_F^2$, which is $\frac{\partial L}{\partial X} = B^T(BXC - A)C^T$.

Applying this identity with $A = \chi(p)$, $B = P$, $C = \rho$, and $X = M$:

$$\frac{\partial L}{\partial M} = P^T(PM\rho - \chi(p))\rho^T \tag{72}$$

Set the derivative to zero to find the optimal $M$:

$$P^T(PM\rho - \chi(p))\rho^T = \mathbf{0} \tag{73}$$

$$P^T PM\rho\rho^T - P^T\chi(p)\rho^T = \mathbf{0} \tag{74}$$

$$(P^T P)M(\rho\rho^T) = P^T\chi(p)\rho^T \tag{75}$$

3. **Simplify the Projection Terms:** We analyse the two matrix products involving $P$ and $\rho$.

- Let $A = P^T P$. This is a $k \times k$ invertible diagonal matrix where $A_{ii}$ is the number of nodes in group $i$.
- Let $B = \rho\rho^T$. Substituting $\rho = (P^T P)^{-1} P^T = A^{-1} P^T$:

$$B = (A^{-1}P^T)(A^{-1}P^T)^T = A^{-1}P^T(P(A^{-1})^T) \tag{76}$$

Since $A = P^T P$ is diagonal, it is symmetric, so $(A^{-1})^T = A^{-1}$.

$$B = A^{-1}(P^T P)A^{-1} = A^{-1}AA^{-1} = I_k A^{-1} = A^{-1} \tag{77}$$

Thus, we have the identity $B = \rho\rho^T = (P^T P)^{-1} = A^{-1}$.

4. **Isolate the Optimal Matrix $M$:** Substitute $A = P^T P$ and $B = A^{-1}$ back into the equation from Step 2:

$$AMA^{-1} = P^T \chi(p)\rho^T \tag{78}$$

To solve for $M$, we right-multiply by $A$:

$$(AMA^{-1})A = (P^T \chi(p)\rho^T)A \tag{79}$$

$$AM = (P^T \chi(p)\rho^T)(P^T P) \tag{80}$$

Now, substitute the definition $\rho^T = (P(P^T P)^{-1})$:

$$AM = P^T \chi(p)(P(P^T P)^{-1})(P^T P) \tag{81}$$

By associativity, the last two terms cancel to the identity matrix:

$$AM = P^T \chi(p)P \cdot ((P^T P)^{-1}(P^T P)) = P^T \chi(p)P \cdot I_k \tag{82}$$

$$(P^T P)M = P^T \chi(p)P \tag{83}$$

Finally, to isolate $M$, we left-multiply by $(P^T P)^{-1}$:

$$(P^T P)^{-1}(P^T P)M = (P^T P)^{-1}P^T \chi(p)P \tag{84}$$

$$I_k M = ((P^T P)^{-1}P^T)\chi(p)P \tag{85}$$

Using the definition $\rho = (P^T P)^{-1}P^T$, we arrive at the unique solution:

$$M(p) = \rho\chi(p)P \tag{86}$$

This confirms that the functional transformation $M(p) = \rho\chi(p)P$ is the unique $k \times k$ matrix that optimally approximates the full $n \times n$ spectral transformation $\chi(p)$ in the least-squares sense, when projected onto the functional subspace. This completes the proof. $\square$

