# OpenReview forum: "Proc-to-Spec: A Functorial Map of Network Processes"
_TMLR — Accepted by TMLR_

### Review · Reviewer_N8T4 · 2025-10-11

**Summary Of Contributions:**

This paper proposes a category-theoretical framework named Proc-to-Spec, which provides a novel way of thinking about dynamic systems. The framework establishes a set of rigorous theorems, including those on physical conservation laws, spectral sensitivity, and stability-spectrum equivalence. Built upon the core idea of a functor that translates physical system dynamics into the language of linear algebra, Proc-to-Spec demonstrates its effectiveness and reliability through tests on both synthetic data and real-world examples.

Strengths：
1.This article is coherent and logically fluent. The use of bold headings in sections such as the methodology and theoretical analysis makes the proposed framework easier to understand.
2.The theoretical analysis section includes extensive Lemma and Theorem analysis, providing rich theoretical support for the proposed framework.
3.The experimental section thoroughly tests the framework using synthetic data and real-world examples, demonstrating the effectiveness of the approach.

Weaknesses：
1.There is a lot of noise, observation bias and missing values ​​in real data. It is recommended to add a discussion on how to improve the robustness of the model.
2.This paper only analyzes monthly LCC, which may result in the loss of some information. It is recommended to add experiments to illustrate the proportion of LCC information in the overall information.

**Audience:**

Yes

**Audience Explanation:**

Different from the predominantly descriptive methods commonly used, this framwork focusing on the geometric nature of the transformations between discrete network states provides a different way of thinking about dynamic systems.

**Claims And Evidence:**

Yes

**Claims Explanation:**

Extensive experiments on synthetic data and real-world examples demonstrate the effectiveness of the category-theoretic framework Proc-to-Spec.

**Requested Changes:**

Please refer to the previous Weakness for details.

---

> ### Author Response · Authors · 2025-11-07
>
> We are very grateful to Reviewer N8T4 for their positive assessment. The reviewer's two "Weakness" points were excellent, practical criticisms that identified key areas where our analysis needed to be more rigorous and transparent. We have performed significant revisions to add the exact discussion and experiments the reviewer recommended.
>
> **All changes in the paper are highlighted in blue**.
>
> ### Response to Weaknesses / Requested Changes
>
> **1. Weakness 1: "add a discussion on how to improve the robustness of the model" (to noise, bias, and missing values)**
>
> We agree that this is a critical issue for any real-world application. We have addressed this in two major ways:
>
> -   **1A. New Case Study (HCP) as a "High-Noise" Test (Section 7)**: Our new, 16+-page neuroscience case study on the Human Connectome Project (HCP) was designed specifically to test the framework's robustness in a **"low-signal, high-noise, micro-timescale"** domain, in sharp contrast to the existing **"high-signal, high-noise, macro-timescale"** ecological study in Section 6.
>
>     -   As detailed in **Section 7.1**, fMRI data is an indirect proxy for neural activity and is exceptionally noisy, contaminated by non-neural artifacts from subject motion, breathing, and heartbeat.
>
>     -   We detail our specific procedures to handle this noise: 1) We used the HCP's **ICA-FIX denoised dataset** to regress out these known noise components. 2) We carefully selected our temporal parameters. Our initial experiments with a 99% window overlap (1-TR step) failed to find a signal, as the overlap mathematically "averaged out" the signal of change. Our choice of a **30-second window with 50% overlap** was a necessary methodological choice to ensure consecutive networks were substantially distinct, allowing a clear signal of change to be detected above the noise floor.
>     - We add a new theoretical section (Section 4.6) to establish the **`Spec-to-Func` model reduction theorem (Theorem 10)** that allows us to compress high-dimensional, potentially noisy spectral functor maps to low-dimensional functional subsapces that are more robust for scientific analysis and interpretation.
>
>     -   We then use our new `Spec-to-Func` model reduction (Theorem 10) as a **theoretically-pprincipled, scientifically-interpretable feature engineering tool** for highly noisy, subject-variable cognitive process identification. The success of our 8-class ML classifier (59.3% accuracy, **Fig. 20**) serves as a strong validation that the framework _is_ robust enough to extract meaningful and generalizable signatures from extremely high-dimensional, noisy data.
>
> -   **1B. New Discussion in Section 8.3**: As requested, we have added a new, explicit discussion on robustness to the **"Practical Limitations" (Section 8.3)**  . This section now contains specific bullet points detailing the robustness measures taken for _both_ case studies:
>
>     -   For the **HCP study**, we explicitly discuss the critical importance of using "pre-processed ICA-FIX denoised data" and our specific windowing parameters ("30-second window with 50% overlap") as the key methodological choices to ensure robustness to noise.
>
>     -   For the **Serengeti study**, we discuss our "robust method of analysing the largest connected component (LCC)" and our pipeline for handling "sparse, non-numeric camera trap data" as the primary methods for managing data sparsity and observation bias.
>
>
> **2. Weakness 2: "add experiments to illustrate the proportion of LCC information in the overall information" (due to "loss of some information")**
>
> This is an excellent point. Our original paper did not prove that our LCC-based analysis was not discarding important information. We have now run this experiment and added it to the paper.
>
> -   The results are presented in our new **Figure 10 ("LCC Coverage and Stability Analysis")** and discussed in **Section 6.2**  .
>
> -   This new analysis provides a quantitative and definitive answer. **Figure 10a** shows that, for the Serengeti dataset, the Largest Connected Component (LCC) **consistently captured 100% of all active nodes and 100% of all active resource (total interaction weight)** in the network for every single month.
>
> -   This result proves that our LCC-based method, while being robust to the transient drop-out of isolated, non-interacting species, **resulted in zero loss of information** from the active, interacting network.
>
> -   Furthermore, **Figure 10b/c** shows that this LCC was not only complete but also highly stable, with a mean Jaccard Index of 0.939 and mean turnover rates of <4%. This confirms the LCC was a robust analytical core.
>
>
> We thank the reviewer for these two suggestions. Addressing them has added a new, rigorous noise-handling validation (the HCP study) and a critical data-loss validation (Figure 10) to the paper, making our claims much more robust and well-supported.

---

### Review · Reviewer_dQse · 2025-10-13

**Summary Of Contributions:**

The paper considers complex networks described as (medium sized) weighted graphs.  For the most part of the node set and their representation is fixed (small changes in nodes might be ok).  The data of relevance is the weights on the edges.  This should describe the amount of interaction between the nodes.

The technical result of the paper is a stability analysis of the spectral representation of the graph, through the symmetrized Laplacian.  It shows that various changes to the weights of the graphs (or perhaps in the node set), have bounded, and perhaps understandable, changes in the spectral representation.  This proven theoretically pulling from nice linear algebra properties, and also demonstrated on a few empirical examples.  The paper has a large case study of a relationship between major entities (predators, prey, plants) in the Serengeti that is captured over a long observation period.

The technical results are presented in a category-theory perspective.  I did not understand why this was needed, and if or what it added.

The writing is mostly clear, with proof sketches in the main body, and technical proofs in the appendix.  There are nice graphs illustrating the many claims and examples.  My two pieces of constructive feedback are:

  - the introduction and related work was too long and felt disconnected from the actual results.  I would have an appreciated a more technical background section that introduced how similar results are presented through category theory, and why they were useful paradigms.

  - the figures were very large, and often several pages away from where they were discussed.  I would encourage the authors to move the figure closer to where they are discussed, even if it meant sacrificing their size or composition.

**Audience:**

Yes

**Audience Explanation:**

I am mixed on this.  As a long-time reviewer for TMLR this is not a paper out of scope of my realm of interest.  But if I were to write a paper like this, TMLR is not the venue for the audience I would target.  It feels much more like an applied math paper (a very nice one for such a venue).  But I do not really understand the tie to machine learning.  There are some nice results about the stability of the symmetrized unnormalized Laplacian that does show up in ML applications, but these are not the emphasized aspects of the work.

To this point, most of the references are not to traditional ML venues.  The few that are mostly to semi-related topics in manifold learning or graph neural networks which the related work section mostly dismisses as not that related.

I am open to this paper being accepted at TMLR.  But I would appreciate a rebuttal from the authors to better make that case.  Even better would be a section (say in the Discussion) explaining the potential implications in machine learning.

[I have updated this question after the author revision and rebuttal]

**Claims And Evidence:**

Yes

**Claims Explanation:**

The theorems as stated all seem to be correct.  They are also mostly verified numerically and through examples.

There are some sort of sensational claims in the introduction about potentially inferring causality, which I found a little dubious.  But they are written in a speculative enough way that I think it's not out of bounds.

**Requested Changes:**

I would like understand better the potential implications in machine learning.  Adding this discussion to the paper is requested.

See also constructive suggestions above. These are not blocking however.

---

> ### Author Response · Authors · 2025-11-07
>
> We thank Reviewer dQse for their insightful and thorough review. We are particularly grateful for the reviewer's very perceptive comments regarding the paper's connection to machine learning and its fit for the TMLR audience.
>
> To address this point, we have fundamentally re-written the paper's core argument to be ML-centric. This involved adding **new theoretical sections (Section 4.2, 4.6)** formally critiquing GNNs and solving model reduction via least-squares, and a **completely new, 16+-page neuroscience case study (Section 7)** demonstrating a concrete ML application for brain analysis.
>
> **All changes in the paper are highlighted in blue**.
>
> We address the reviewer's points below.
>
> **1. Requested Change: "I would like understand better the potential implications in machine learning. Adding this discussion to the paper is requested."**
>
> We have addressed this in three significant ways:
>
> -   **1A. New Theoretical Critique of ML Models (Section 4.2):** The reviewer noted they "did not understand why category theory was needed." We have added a new theoretical section, **Section 4.2 ("Non-Functoriality of Entangled State-Update Models")**, to provide this exact motivation.
>
>     -   We define a broad class of common temporal ML models (e.g., GNN-RNNs) as "Entangled State-Update Models".
>
>     -   We then use the axioms of category theory to _formally prove_ that these models are **non-functorial**. Specifically, they fail to preserve identity (Theorem 1), introducing artificial latent dynamics, and fail to preserve composition (Theorem 2), making them **path-dependent**.
>
>     -   This section now provides the core theoretical justification for our work. We frame category theory as a **benchmark for model soundness**, which common ML models fail, and we propose our "glass-box" framework as a provably sound alternative essential for rigorous scientific analysis.
>
> -   **1B. New Case Study with ML Application (Section 7)**: To prove the practical implications for ML, we have added a comprehensive, 16+-page second case study (Section 7) on a low-signal, high-noise neuroscience dataset (the **Human Connectome Project**).
>
>     -   We explicitly frame this analysis as a **theoretically principled, scientifically interpretable feature-engineering** task for human brain analysis from highly fMRI data.
>
>     -   We use our **new `Spec-to-Func` projection (Theorem 10)** to create low-dimensional, interpretable functional fingerprints ($M(p)$) of 8 different cognitive processes captured in the Human Connectome Project.
>
>     -   As shown in **Figure 20**, we use these $M(p)$ fingerprints as features to train a non-linear SVM. With strict Leave-One-Subject-Out cross-validation, our features achieve **59.3% accuracy** on this highly challenging 8-class problem (vs. 12.5% chance).
>
>     -   This provides a concrete, novel, and successful machine learning application, demonstrating our framework's utility as a "glass-box" tool for **feature extraction in high-noise, high-dimensional network data**.
>
> -   **1C. Rewritten Discussion (Section 8)**: As requested, we have rewritten **Section 8.2 ("Broader Implications")** to be almost entirely focused on the ML implications, summarising the arguments from the new Sec 4.2 (ML Critique), Sec 4.6 (Model Reduction), and Sec 7 (Neuroscience ML Study).
>
> **2. Constructive Feedback: Intro/Related Work "too long and felt disconnected from the actual results."**
>
> We thank the reviewer for this accurate criticism. The original introduction was disconnected because the results it needed to connect to (Sec 4.2, Sec 4.6, and Sec 7) did not exist yet.
>
> -   We have **rewritten the Introduction (Section 1) and Related Work (Section 2.5)** to fix this.
>
> -   The Introduction now opens with the dual gaps of "**descriptive**" traditional models and "**non-functorial**" ML models. The section now also clearly contrasts the Serengeti and Neuroscience studies and emphasizes the **ML focus** of the latter.
>
> -   The Related Work (Sec 2.5) no longer "dismisses" GNNs, but instead sets up the problem, introducing them as "Entangled State-Update Models" to create a direct narrative link to the new theoretical critique in Section 4.2.
>
> **3. Constructive Feedback: Figure Placement**
>
> The reviewer noted that "figures were very large, and often several pages away."
>
> -   We have gone through the entire manuscript and **adjusted the size and placement of all figures and tables** in the `.tex` file to be as close as possible to their first textual reference. Due to the **complexity of both case studies (Sec 6, Sec 7)**, we intended to maintain the rich quantitative and qualitative results in those figures and tables for **maximum scientific rigor**.
>
> By providing both a formal critique of existing ML models and a new, practical ML application, we are confident the paper is now an excellent fit for the TMLR audience.

---

> > ### Comment · Reviewer_dQse · 2025-11-15
> >
> > Thanks to the authors for the very thorough revision and additions to the paper.  I believe the paper is better for it, and now a clear good fit on topic for TMLR's audience.

---

### Review · Reviewer_owpE · 2025-10-21

**Summary Of Contributions:**

The authors introduce a framework that maps processes in ressource-constrained systems to the eigenspace in order to analyse them there. The framework is tested on a network coming from a food web.

The authors aim at a mathematical formulation and formal language for processes with a focus on the transformations between processes. The analysis in the eigenspace is interesting and the foodweb is a motivating real-world example.

However, if the authors aim at introducing the framework as a generally useful tool, they'd need to evaluate it on more datasets from diverse sources.
If they rather aim at the analysis of this specific foodweb and its development I'd expect new findings in this dataset and more biological analysis -- that would, however, not really fit TMLR.


**Strong points**

S1) Analysis of processes in the eigenspace seems like a good idea

S2) Use-case is motivating

S3) Well written, intuitions are explained

**Weak points**

W1) Only one use case (even if it is based on a large amount of data) is not enough to show the usefulness of an entire new framework

W2) Some details in the experiments could improve the paper, e.g. while Table 2 gives some information about the data that is not really necessary (e.g., how many square km were regarded in the study), I'm missing exact information of the actual graph that is used after all the preprocessing. While there is a description how it is obtained, I'd be interested in how large that largest connected component actually is.

W3) Adding some quantitative analysis would be great: e.g., in Fig. 10, instead of just regarding heat maps and their off-diagonal values, one could use an actual metric to capture this (e.g., measure bandwidth of the matrix)

**Audience:**

Yes

**Audience Explanation:**

Analysing processes in the eigenspace is interesting and I can imagine that it indeed works for systems other than the food network.

**Broader Impact Concerns:**

I do not have concerns regarding ethical implications.

**Claims And Evidence:**

No

**Claims Explanation:**

For this it would need more experiments on data coming from other sources to prove that the framework also works for more general use cases and not only this specific data.

A lot of the analysis is qualitative, which is nice and intuitive, but a quantitative evaluation would be more objective here, especially for readers coming from computer science.

**Requested Changes:**

In order to be acceptable, I'd suggest to realize the following:

A) Add at least two more datasets for the analysis

B) Add quantitative analysis of the claims supported by qualitative analysis only

C) Increase reproducibility by being more precise:
 - give information about the data after preprocessing (e.g., size of largest connected component)
 - how did you chose the "top ten" in Figure 11? Also in the caption of Fig.11, which exactly are the "ecological literature" supporting your claim?

---

> ### Author Response · Authors · 2025-11-07
>
> We thank Reviewer owpE for their constructive feedback. The reviewer's suggestions—that the framework's generality must be proven with more datasets and its claims strengthened with quantitative metrics—were clear and invaluable.
>
> We undertook a significant revision to address every point. **All changes in the paper are highlighted in blue**. Key changes include:
>
> 1.  A **new, comprehensive case study** (Section 7) in a completely different scientific domain (neuroscience).
>
> 2.  **New theoretical sections** (Section 4.2, 4.6) providing a formal critique of GNNs and a strong connection to machine learning as well as a new `Spec-to-Func` projection.
>
> 3.  **New quantitative metrics, figures, and analysis** (e.g., Section 6.2, Figure 5, 6, 9, 10, 12, 13) to make our original analysis more rigorous.
>
> Here, we address the reviewer's requested changes point-by-point.
>
> ### Response to Requested Changes
>
> **A) "Add at least two more datasets for the analysis" (and W1: "Only one use case")**
>
> We added a comprehensive second case study (Section 7) analysing dynamic brain connectomes from the **Human Connectome Project (HCP)**.
>
> This new case study is a direct methodological contrast to the Serengeti study, proving its generality:
>
> -   **Serengeti (Sec 6):** A **high-signal, high-noise, macro-timescale, sparse, physical** system. Challenge: interpreting a known, massive external shock (a drought).
>
> -   **HCP (Sec 7):** A **low-signal, high-noise, micro-timescale, dense, abstract** system. Challenge: detecting fleeting, internal cognitive state shifts from high-noise fMRI data.
>
>
> This new case study also demonstrates a new machine learning application. We use our `Spec-to-Func` projection (Theorem 10) as a principled feature-engineering tool to create "functional fingerprints" ($M(p)$) of cognitive processes. As shown in **Figure 20**, these fingerprints are used to train a non-linear SVM to classify 8 different cognitive states with **59.3% accuracy** (vs. 12.5% chance) using strict Leave-One-Subject-Out cross-validation.
>
> By demonstrating success in two **fundamentally different** scientific domains, we provide the strong evidence of generality the reviewer requested.
>
> **B) "Add quantitative analysis of the claims supported by qualitative analysis only" (and W3)**
>
> We agree. The prior qualitative analysis was insufficient.
>
> We replaced this with a rigorous quantitative analysis. As detailed in **Section 6.3** and **Figure 12**'s caption, we now use two specific metrics to quantify the "shattering" of the network's geometry:
>
> 1.  **Diagonal Dominance Ratio ($R_{DD}$):** (Defined in Eq. 15) Quantifies how "diagonal" (stable) a transformation is.
>
> 2.  **Shannon Entropy ($H$):** We compute entropy on the Fiedler vector's squared components after transformation (Fig 12, middle row). Low entropy means preservation; high entropy proves it was "shattered".
>
>
> This analysis lets us _prove_ our claim:
>
> -   A **Typical Season** (Fig 12f) was a minor perturbation, preserving the Fiedler vector ($H = 0.0353$ bits).
>
> -   The **Drought Collapse** (Fig 12d) was a catastrophic reorganisation, shattering the Fiedler vector ($H = 0.9582$ bits).
>
>
> **C) "Increase reproducibility by being more precise" (and W2)**
>
> We added a new section and figures to address reproducibility concerns.
>
> -   **C1. LCC Information (W2):** Reviewer asked for "exact information of the actual graph... how large that largest connected component actually is." We added a new data processing subsection (**Section 6.2**) and a new validation figure (**Figure 10**) to address this.
>
>     -   **Figure 10a** proves the LCC **consistently captured 100% of all active nodes and resources** in the network each month. Thus, our LCC-based analysis lost no information from the active network.
>
>     -   **Figure 10b/c** quantifies the LCC's stability (e.g., mean Jaccard Index = 0.939), showing it was a robust core.
>
> -   **C2. "Top ten" in Figure 11 (now 13):** We clarified in the caption of **Figure 13** that the "top ten" refers to the ten most central species _before_ the drought (June 2011). This provides a clear, interpretable visualisation of how the ranks of these key species were disrupted.
>
> -   **C3. "ecological literature" citation:** We clarified this in the caption of **Figure 13** and the main text. Our findings—that large grazers (_Eudorcas_, _Alcelaphus_) and their specialist predators (_Panthera_) are "losers" while generalists (_Crocuta_) are "winners"—are supported by **Sinclair et al. (2007)**. This is a seminal review paper on long-term Serengeti dynamics that explicitly confirms these ecological shifts.
>
>
> We also revised the **Abstract, Introduction (Section 1), Related Work (Section 2), and Discussion (Section 8)** to integrate these new contributions.
>
> We thank the reviewer again for their feedback, which substantially improved the generality, quantitative rigour, and ML-relevance of our work.

---

> > ### Comment · Reviewer_owpE · 2025-12-04
> >
> > Thanks to the authors for putting so much effort into the revision. Especially including another use case and adding a meaningful quantitative analysis improved the work.

---

### Decision · Action_Editor_6dJg · 2026-02-06

**Recommendation:** Accept as is

**Audience:**

Yes

**Audience Explanation:**

There has now been an effort to better link the methods in the manuscript with the standard approaches used in the ML community. The section on Machine Learning on Graphs is a welcome addition.

**Claims And Evidence:**

Yes

**Claims Explanation:**

After the revision, the authors have added new experiments that better support the claims. The authors have also downweighted a few strong sentences present in their previous draft.